# Unlocking the Potential of Model Calibration in Federated Learning

**Yun-Wei Chu**[1], **Dong-Jun Han**[2], **Seyyedali Hosseinalipour**[3], **Christopher G. Brinton**[1]
[1]Purdue University, [2]Yonsei University, [3]University at Buffalo-SUNY
`chu198@purdue.edu, djh@yonsei.ac.kr,`
`alipour@buffalo.edu, cgb@purdue.edu`

## Abstract

Over the past several years, various federated learning (FL) methodologies have been developed to improve model accuracy, a primary performance metric in machine learning. However, to utilize FL in practical decision-making scenarios, beyond considering accuracy, the trained model must also have a reliable confidence in each of its predictions, an aspect that has been largely overlooked in existing FL research. Motivated by this gap, we propose Non-Uniform Calibration for Federated Learning (`NUCFL`), a generic framework that integrates FL with the concept of *model calibration*. The inherent data heterogeneity in FL environments makes model calibration particularly difficult, as it must ensure reliability across diverse data distributions and client conditions. Our `NUCFL` addresses this challenge by dynamically adjusting the model calibration objectives based on statistical relationships between each client's local model and the global model in FL. In particular, `NUCFL` assesses the similarity between local and global model relationships, and controls the penalty term for the calibration loss during client-side local training. By doing so, `NUCFL` effectively aligns calibration needs for the global model in heterogeneous FL settings while not sacrificing accuracy. Extensive experiments show that `NUCFL` offers flexibility and effectiveness across various FL algorithms, enhancing accuracy as well as model calibration.

## 1 Introduction

Federated learning (FL) has rapidly gained traction as a prominent distributed machine learning approach, enabling collaborative model training across a network of clients by periodically aggregating their local models at a central server (Konecný et al., 2016; McMahan et al., 2017). Its significance extends to various critical applications, including medical diagnostics, self-driving cars, and multilingual systems, where data privacy and decentralized training are paramount. FL has been widely studied with a focus on improving local model updates (Reddi et al., 2021; Sahu et al., 2018), refining global aggregation methods (Ji et al., 2019; Wang et al., 2020a), and enhancing communication efficiency (Sattler et al., 2019; Diao et al., 2020; Parasnis et al., 2023). Most of these works in FL consider accuracy as the main performance metric.

**Motivation.** However, beyond accuracy, in various decision-making scenarios where an incorrect prediction may result in high risk (e.g., medical applications or autonomous driving), it is also crucial for the users to determine whether to rely on the FL model's prediction or not for each decision. In particular, users should rely on the neural network's decision only when the prediction is likely to be correct. Otherwise, they may need to consider alternatives such as human decision. To achieve this, the trained FL model should have a *reliable confidence* in each of its predictions, meaning that the confidence of the neural network matches well with its actual accuracy. In centralized settings, recent studies including Guo et al. (2017) have uncovered that neural networks are often miscalibrated, indicating that the prediction confidence of the model does not accurately reflect the probability of correctness. Handling this miscalibration issue is even more important in many FL use-cases, where an overconfident global model could lead to misinformed decisions with potentially severe consequences for each client.

In centralized training settings, research generally follows two paths to address this miscalibration issue. First, train-time calibration methods (Hebbaluguppe et al., 2022b; Liang et al., 2020; Liu et al.,

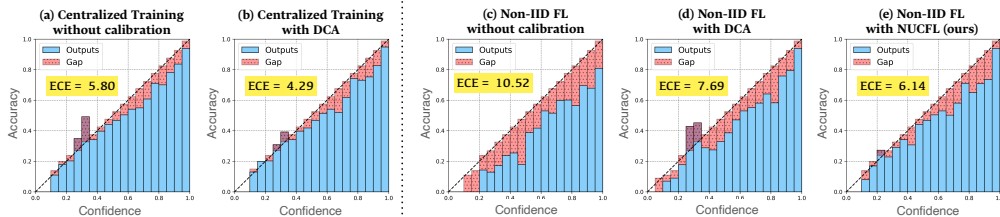

Figure 1: Reliability diagrams and calibration errors for centralized training and non-IID FL (using FedAvg) trained with various calibration methods on CIFAR-100 dataset. Our method ensures well-calibrated FL, evidenced by a notably smaller calibration error and a smaller gap (red region) between confidence and accuracy.

2021; Mukhoti et al., 2020; Kumar et al., 2018; Müller et al., 2019; Lin et al., 2017) incorporate explicit regularizers during the training process to adjust neural networks, scaling back over/under-confident predictions. On the other hand, post-hoc calibration (Guo et al., 2017; Hendrycks & Gimpel, 2016; Hinton et al., 2015; Kull et al., 2017) transforms the network's output vector to align the confidence of the predicted label with the actual likelihood of that label for the sample. Post-hoc calibration methods are applied to the already trained model to improve calibration using an additional holdout dataset. However, an appropriate holdout dataset may not be available in many privacy-sensitive applications, making post-hoc calibration strategies often impractical for FL.

**Goals and observations.** In this paper, we aim to incorporate model calibration into the FL training process, a problem that has been largely underexplored in existing research. This setting introduces new challenges to improving model calibration, as the heterogeneous data distributions across different clients must be carefully considered. We first observe whether existing train-time calibration methods applied during client-side local training can effectively address calibration needs in FL. To gain insights, in Fig. 1, we report accuracy, confidence, and the expected calibration error (ECE) of different methods in different settings. Figs. 1(a) to 1(b) illustrate the capability of train-time calibration methods to narrow the gap between model confidence and accuracy in centralized learning. A comparison between Figs. 1(a) and 1(c) reveals that FL experiences more significant model miscalibration than centralized learning. This disparity is likely due to data heterogeneity across distributed FL clients, causing each client's local data to have a different impact on calibration performance. When applying auxiliary-based calibration methods, such as DCA (Liang et al., 2020) and MDCA (Hebbalaguppe et al., 2022b) – which add a penalty term to the original classification loss – to FL's local training (Fig. 1(d)), a better ECE is achieved compared to FL without calibration (Fig. 1(c)). However, we observe that the model in Fig. 1(d) is still not well-calibrated by neglecting global calibration needs in heterogeneous FL settings, resulting in overconfident predictions. Given the fact that train-time calibration methods directly modify confidence during training, potentially affecting the model's accuracy—which is also a critical factor—we thus pose the following question:

*How can we unlock the potential of model calibration in FL while not sacrificing accuracy?*

**Contributions.** In this paper, we propose Non-Uniform Calibration for Federated Learning (`NUCFL`), a versatile and generic model calibration framework for FL that integrates seamlessly with any existing FL algorithms, such as FedProx (Sahu et al., 2018), Scaffold (Karimireddy et al., 2019), FedDyn (Acar et al., 2021), and FedNova (Wang et al., 2020b). Designed to work with any train-time calibration auxiliary loss, `NUCFL` is a straightforward yet effective solution that dynamically adjusts calibration penalties based on the relationship between global and local distribution. Specifically, our intuition is that if each client's local model closely resembles the global model, it is likely that the local model and data represent the global calibration needs well. Therefore, our idea is to choose a large penalty term to improve model calibration for the clients that have a high similarity, while imposing a small penalty for other clients to prioritize accuracy. This non-uniform calibration approach considering data heterogeneity across FL clients significantly minimizes calibration errors without accuracy degradation, as seen in Figure 1(e). Importantly, the flexibility of `NUCFL` allows for continuous improvement as new FL algorithms or calibration losses are developed. Our key contributions are summarized as follows:

- Our work pioneers a systematic study on model calibration in FL. We analyze whether existing calibration methods can be applied to FL algorithms and study the following question: *How does model calibration impact the behavior of federated optimization methods?*

- We propose `NUCFL`, a non-uniform model calibration approach tailored to FL. Our `NUCFL` dynamically adjusts calibration penalties based on the nuanced relationship between each local model and the global model, effectively capturing data heterogeneity.

- Extensive experiments demonstrate that `NUCFL` is highly flexible, seamlessly integrating with 5 different FL algorithms (e.g., FedProx, Scaffold, FedDyn) and various auxiliary-based train-time calibration methods to improve both calibration performance and accuracy.

## 2 RELATED WORK

**Federated learning.** FedAvg (McMahan et al., 2017) is the pioneering FL algorithm, with numerous adaptations proposed to mitigate the accuracy drop experienced with non-IID data. Much of the research has focused on designing better global aggregation methods to enhance model convergence and performance. Some researchers (He et al., 2020; Lin et al., 2020) replaced weight averaging with model ensemble and distillation, others (Yurochkin et al., 2019; Wang et al., 2020a) matched local model weights before averaging, and another group (Ji et al., 2019) introduced an attention mechanism to optimize global aggregation. Another significant area has concentrated on optimizing local training protocols to optimize computation and data usage on clients' devices. Some researchers (Sahu et al., 2018; Acar et al., 2021) employed regularization toward the global model, while Karimireddy et al. (2019) utilized server statistics to correct local gradients. Additionally, various studies have addressed challenges specific to FL, such as ensuring fairness across all participants (Li et al., 2020; Chu et al., 2024b; 2022a;b), enhancing communication efficiency to minimize bandwidth usage (Ko et al., 2023; Chu et al., 2024a; Chang et al., 2024; Lan et al., 2023a;b; Yuan et al., 2024), and addressing privacy concerns inherent in distributed computations (Truong et al., 2020). However, the main focus of existing FL research is on improving accuracy. The mismatch between confidence and actual accuracy in these works leads to overconfident FL models, making FL inapplicable in high-risk decision-making scenarios. Our work bridges this gap by investigating an overlooked aspect of federated learning: *model calibration*.

**Model calibration.** Model calibration techniques can be categorized into post-hoc calibration and train-time calibration methods. Post-hoc methods adjust the model using a hold-out dataset after training. Common examples include Temperature Scaling (Guo et al., 2017), which adjusts logits by a scalar learned from a hold-out set before softmax, and Dirichlet Calibration (Kull et al., 2017), which modifies log-transformed class probabilities with an extra network layer. However, in FL applications, the lack of an additional validation set due to privacy concerns often renders post-hoc calibration methods inapplicable. On the other hand, train-time calibration integrates calibration directly within the training process. Müller et al. (2019) enhance calibration using label smoothing on soft targets, while Lin et al. (2017) use focal loss to minimize the KL divergence between predicted and target distributions. Recently, augmenting standard cross-entropy loss with additional penalty loss terms has become a popular train-time calibration method (Liang et al., 2020; Hebbalaguppe et al., 2022b; Kumar et al., 2018). This auxiliary loss penalizes the model when there is a reduction in cross-entropy loss without a corresponding increase in accuracy, often indicative of overfitting. However, as seen in Fig. 1, a direct application of these methods still faces limitations, as they are not able to meet the calibration needs considering data heterogeneity across FL clients. Our work fills this gap by taking a non-uniform calibration approach considering the characteristics of FL.

**Calibration and FL.** Peng et al. (2024) proposed FedCal, a calibration scaler aggregated solely from local clients, and, to the best of our knowledge, is the only prior work explicitly addressing calibration error in FL. However, FedCal does not consider the interaction between global and local calibration needs, which can lead to a calibration bias towards local heterogeneity. Our calibration framework addresses this issue by considering the interaction between client and server, thereby better accommodating broader global calibration requirements. We empirically confirm these advantages of our approach compared to the existing work in Section 5.2. It is also worth mentioning that Zhang et al. (2022); Luo et al. (2021) also use the term "calibration" in the FL setup. However, we stress that the context of calibration in their work is different from the one in the model calibration literature that considers confidence of the prediction. Zhang et al. (2022); Luo et al. (2021) focuses on calibrating the classifier to improve the *accuracy* itself, which can be categorized into conventional FL works that do not consider the error between model confidence and accuracy.

---

**Algorithm 1** General FL Framework

---
1: **Input:** global model $w$, local model $w_m$ for client $m$, local epochs $E$, and rounds $T$.
2: **for** each round $t = 1, 2, ..., T$ **do**
3:     Server sends $w^{(t-1)}$ to all clients.
4:     **for** each client $m \in M$ **do**
5:         Initialize local model $w_m^{(t,0)} \leftarrow w^{(t-1)}$
6:         **for** each epoch $e = 1, 2, ..., E$ **do**
7:             Each client performs local updates via: $w_m^{(t,e)} \leftarrow \texttt{ClientOPT}(w_m^{(t,e-1)}, \mathcal{L}_m)$
8:         **end for**
9:         $w_m^{(t,E)}$ denotes the result after performing $E$ epochs of local updates.
10:       Client sends $\delta_m^{(t)} = w^{(t-1)} - w_m^{(t,E)}$ to the server after local training.
11:     **end for**
12:     Server computes aggregate update $\delta^{(t)} = \sum_{m \in M} \frac{|\mathcal{D}_m|}{|\mathcal{D}|} \delta_m^{(t)}$
13:     Server updates global model $w^{(t)} \leftarrow \texttt{ServerOPT}(w^{(t-1)}, \delta^{(t)})$
14: **end for**

---

# 3 PRELIMINARIES

## 3.1 FEDERATED LEARNING

In FL, the training data is gathered from a set of clients $M$ and each client $m$ possesses a training set $\mathcal{D}_m = \{x_i, y_i\}_{i=1}^{|\mathcal{D}_m|}$, where $x$ represents the input data and $y$ denotes the corresponding true label. We consider the following standard optimization formulation of federated training and seek to find model parameters $w$ that solve the problem:

$$\min_w \mathcal{L}(w) = \sum_{m \in M} \frac{|\mathcal{D}_m|}{|\mathcal{D}|} \mathcal{L}_m(w), \tag{1}$$

where $w$ represents the model parameter, $\mathcal{D} = \cup_{m \in M} \mathcal{D}_m$ denotes the aggregated training set from all clients, and $\mathcal{L}_m$ measures the average loss of a model on the training data of the $m$-th client. The objective is to find a model that fits all clients' data well on (weighted) average.

FL methods involve *parallel local training at clients* and *global aggregation at a server* over multiple communication rounds to address the aforementioned problem. Most federated optimization methods fit within a general framework outlined by Reddi et al. (2020), as shown in Algorithm 1. At round $t$, the server sends its previous global model $w^{(t-1)}$ to all clients as initialization. Each client $m$ then performs $E$ epochs of local training using $\texttt{ClientOPT}$ depending on the FL algorithm, and subsequently produces a local model $w_m^{(t,E)}$. Then each client communicates the difference $\delta_m^{(t)}$ between their learned local model and the server model, denoted as $\delta_m^{(t)} = w^{(t-1)} - w_m^{(t,E)}$. The server computes a weighted average of the client updates, denoted as $\delta^{(t)}$, and then updates the global model $w^{(t)}$ using $\texttt{ServerOPT}$, which also depends on FL designs. For example, FedAvg employs standard stochastic gradient descent updates for $\texttt{ClientOPT}$, and its $\texttt{ServerOPT}$ is formulated as $w^{(t)} = w^{(t-1)} - \delta^{(t)}$.

## 3.2 MODEL CALIBRATION

In supervised multi-class classification, the input $x \in \mathcal{X}$ and label $y \in \mathcal{Y} = \{1, ..., K\}$ are random variables following a ground truth distribution $\pi(x, y) = \pi(y|x)\pi(x)$. Let $s$ be the confidence score vector in $\mathbb{R}^K$, with $s[y] = f(x)$ representing the confidence that the model $f(\cdot)$ predicts for a class $y$ given input $x$. The predicted class $\hat{y}$ is computed as: $\hat{y} = \arg\max_y s[y]$, with the corresponding confidence $\hat{s} = \max_y s[y]$. A model is considered perfectly calibrated (Guo et al., 2017) if:

$$\mathbb{P}(\hat{y} = y | \hat{s} = p) = p, \forall p \in [0, 1]. \tag{2}$$

That is to say, we expect the confidence estimate of the prediction to reflect the true probability of the prediction being accurate. For example, given 100 predictions, each with a confidence of $0.8$, we expect that 80 should be correctly classified. One concept of miscalibration involves the difference between confidence and accuracy, which can be described as:

$$\mathbb{E}_{\hat{s}}[|(\hat{y} = y|\hat{s} = p) - p|]. \tag{3}$$

Note that "increasing confidence" refers to adjusting the model's predicted probabilities to higher values, but it does not necessarily imply "better model calibration." Excessively high confidence can lead to overconfident predictions, potentially misleading users to over-rely on the model even when its predictions are inaccurate. The primary goal of model calibration, whether via train-time or post-hoc methods, is to minimize discrepancy (3) to ensure the model's confidence accurately reflects its prediction accuracy. This ensures that users can directly know the probability of correctness based on the model confidence, allowing the model to be applied in various decision-making scenarios.

# 4 PROPOSED MODEL CALIBRATION FOR FEDERATED LEARNING

## 4.1 BRIDGING FEDERATED LEARNING AND MODEL CALIBRATION

Our approach aims to implement calibration on each FL client's local training. Note that in conventional FL algorithms, the loss $\mathcal{L}_m$ of client $m$ in (1) can be expressed as follows:

$$\mathcal{L}_m(w_m^{(t,e)}) = \frac{1}{|\mathcal{D}_m|} \sum_{i=1}^{|\mathcal{D}_m|} \ell(w_m^{(t,e)}; x_i, y_i), \tag{4}$$

where $w_m$ is the local model, $t$ the current global round, $e$ the local iteration index, and $\ell(\cdot)$ the loss function (e.g., cross-entropy) applied to a data instance. One can also modify the loss in (4), i.e., by adding a proximal term as in FedProx (Sahu et al., 2018). Recent studies on calibration (Liang et al., 2020; Hebbalaguppe et al., 2022b) suggest that incorporating an auxiliary loss, which captures the disparity between accuracy and confidence (as in equation (3)), and penalizing the model accordingly, leads to superior performance compared to direct loss type adjustments (Müller et al., 2019; Brier, 1950; Lin et al., 2017). These auxiliary loss methods maintain the original classification loss and seamlessly add a penalty, minimizing interference with FL performance, especially since some FL algorithms are specifically designed around their classification loss strategies. This makes the auxiliary loss methods easily applicable and effective across various FL settings.

**Proposed framework.** Therefore, for client $m$, we propose to integrate the calibration loss with the FL loss as follows:

$$\mathcal{L}_m^{cal}(w_m^{(t,e)}) = \frac{1}{|\mathcal{D}_m|} \sum_{i=1}^{|\mathcal{D}_m|} \left[ \ell(w_m^{(t,e)}; x_i, y_i) + \beta_m \ell_{cal}(w_m^{(t,e)}; x_i, y_i) \right], \tag{5}$$

where $\ell_{cal}(\cdot)$ is the auxiliary calibration loss function, and $\beta_m$ is a scalar weight for each client $m$. In the centralized calibration case, this scalar weight is manually selected to refine the auxiliary calibration loss. However, in our design, we automatically determine $\beta_m$ based on the relationship between the server and clients in a FL setup (without any hyperparameter tuning), taking into account the heterogeneity as detailed in Section 4.2. The auxiliary calibration loss $\ell_{cal}$ can be derived from various calibration designs. For instance, Liang et al. (2020) directly adds the difference between confidence and accuracy (DCA) as an auxiliary loss, which can be represented as:

$$\ell_{cal}\left(\{c_i\}_{1 \le i \le N}, \{\hat{s}_i\}_{1 \le i \le N}\right) = \ell_{dca} = \left| \frac{1}{N} \sum_{i=1}^{N} c_i - \frac{1}{N} \sum_{i=1}^{N} \hat{s}_i \right|, \tag{6}$$

where $N$ denotes the total number of training samples, $\hat{s}_i$ represents the confidence score of the model's predicted label $\hat{y}_i$ for sample $i$, and $c_i = 1$ if $\hat{s}_i$ equals the true label $y_i$; otherwise, $c_i = 0$. On the other hand, Hebbalaguppe et al. (2022b) provide a more detailed analysis of (6) and propose a Multi-class DCA (MDCA), which offers a more granular examination by considering all classes instead of focusing solely on the predicted label. The proposed auxiliary loss can be written as:

$$\ell_{cal}\left(\{c_i\}_{1 \le i \le N}, \{s_i\}_{1 \le i \le N}\right) = \ell_{mdca} = \frac{1}{K} \sum_{j=1}^{K} \left| \frac{1}{N} \sum_{i=1}^{N} c_i[j] - \frac{1}{N} \sum_{i=1}^{N} s_i[j] \right|, \tag{7}$$

where $K$ is the number of classes and $N$ the number of samples. $c_i[j] = 1$ shows that label $j$ is the ground truth for sample $i$, and $c_i[j]$ gives the confidence score for the $j$-th class of sample $i$.

**Advantages and challenges.** The differentiability of these auxiliary loss terms ((6) and (7)) facilitates their integration with other application-specific loss functions, allowing them to enhance

calibration without significantly impacting the primary classification loss, $\ell$. This minimizes any adverse effects on performance while correcting calibration errors. For instance, FL algorithms like FedProx (Sahu et al., 2018) and FedNova (Wang et al., 2020b) incorporate custom loss functions within their local modeling function, `ClientOPT` (outlined in Algorithm 1), to address deviations from the global model due to non-IID data. Despite these enhancements, it is critical to note that these advanced FL algorithms still fundamentally require a standard classification loss, which they fine-tune for specific objectives. When integrating calibration into the `ClientOPT` of FL algorithms, we will exclusively calibrate the standard classification loss, maintaining the integrity of other design elements, such as global aggregation, to ensure their original objectives are not compromised. Choosing an appropriate $\beta_m$ tailored to each client $m$ is the remaining challenge.

## 4.2 Non-Uniform Calibration for Federated Learning (NUCFL)

**Motivation: Limitation of uniform penalty.** In Equation (5), we promote calibration during federated local training by targeting the confidence-accuracy gap in individual local models via auxiliary calibration loss. The main aim of FL is to train a global model capable of effectively handling the global data distribution. Therefore, we aim for calibration to account for the heterogeneity inherent in FL setups, marking a departure from calibration approaches used in centralized learning. Direct application of auxiliary loss calibration methods without adapting to FL characteristics would result in uniform calibration weights, such that $\beta_1 = \beta_2 = ... = \beta_m$. This leads to local models being calibrated solely to their respective datasets $D_m$, with uniformly small weights biasing calibration toward local heterogeneity, and uniformly large weights potentially neglecting accuracy improvements from classification. Such discrepancies can result in a global model that, while comprising well-calibrated local models, does not achieve optimal performance across the broader distribution. This fixed calibration penalty approach, therefore, risks neglecting the broader global calibration needs necessary for optimal performance across the entire global distribution, e.g., as in Fig. 1(d).

**Non-uniform penalty design (NUCFL).** Given that each client receives the global model from the previous round as their starting point for each new federated round, this offers an opportunity to indirectly adjust local calibration to mirror global model characteristics. To leverage this, we propose Non-Uniform Calibration for Federated Learning (NUCFL), which provides a dynamic calibration penalty to local models to better reflect global calibration needs. Drawing inspiration from the concept of similarity characterization in FL techniques (Sahu et al., 2018; Tan et al., 2023), we hypothesize

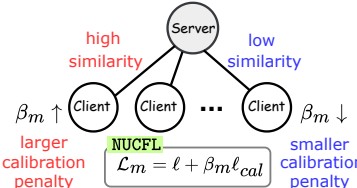

Figure 2: Idea of proposed NUCFL.

that a local model closely resembling the global model is likely to represent global characteristics well, suggesting that the penalty appropriately reflects the calibration needs of the global model. Conversely, a dissimilar/heterogeneous local model suggests a focus on local objectives (e.g., to improve accuracy) at the expense of global alignment. By adjusting penalties at each local epoch based on the similarity between local and global models, NUCFL ensures that local training enhances *both the calibration and accuracy* of the global model. As illustrated in Fig. 2, we specifically set

$$\beta_m = \texttt{sim}(\delta^{(t-1)}, \delta_m^{(t,e)}),\tag{8}$$

where $\delta^{(t-1)}$ is the accumulated gradient from the previous round $t-1$, representing the received global model, while $\delta_m^{(t,e)} = w^{(t-1)} - w_m^{(t,e)}$ calculates the gradient for client $m$ at local epoch $e$ during round $t$. The function $\texttt{sim}(\cdot)$ measures similarity between local and global models, generating values between 0 and 1. Specifically, it can use general cosine similarity or more advanced measures like Centered Kernel Alignment (CKA) (Kornblith et al., 2019).

For instance, if the local model is similar to the global model, indicating alignment in their characteristics, a higher calibration penalty is applied to the local model to address global calibration needs. Otherwise, if the similarity score is low, a smaller penalty is applied, focusing on accuracy improvement instead. Our non-uniform penalty method tailored to FL, strategically improving model calibration and accuracy. Additionally, our NUCFL requires no extra hyperparameter tuning, as $\beta_m$ is automatically determined by (8) based on the given similarity function.

**Remark 1.** NUCFL can integrate any auxiliary-based calibration loss and adapt to any FL algorithm, independent of aggregation method or local loss function. To our knowledge, this is the early work to specifically address model calibration across various FL algorithms. Similar to most calibration

research (Lin et al., 2017; Müller et al., 2019; Brier, 1950; Kumar et al., 2018; Mukhoti et al., 2020; Liu et al., 2021; Liang et al., 2020; Hebbalaguppe et al., 2022b;a), our work is an empirical study that explores the practical impacts of our methods. This research area remains challenging, particularly in providing theoretical guarantees for calibration methods, even in centralized settings; we instead substantiate the effectiveness of our method through extensive experiments. We will demonstrate how our strategic approach effectively calibrates FL in Section 5.2.

**Remark 2 (Compatibility with post-hoc calibration).** Although our NUCFL is a train-time calibration approach that is applied during FL training, the proposed technique complements post-hoc methods whenever a holdout dataset is available. For example, temperature scaling can be applied to the trained global FL model for further refinement, as empirically demonstrated in Section 5.2.

## 5 EXPERIMENTS

### 5.1 EXPERIMENTAL SETUP

**Dataset and model.** We conduct experiments using four image classification datasets commonly utilized in FL research (Caldas et al., 2018; McMahan et al., 2017; Mohri et al., 2019): MNIST (LeCun et al., 1998), FEMNIST (Cohen et al., 2017), CIFAR-10 (Krizhevsky, 2009), and CIFAR-100 (Krizhevsky, 2009). For the MNIST and FEMNIST datasets, we use a CNN (LeCun & Bengio, 1998) with two 5x5 convolution layers, each followed by 2x2 max pooling, and fully connected layers with ReLU activation. We use AlexNet (Krizhevsky et al., 2012) with five convolutional layers and ResNet-34 (He et al., 2015) for the CIFAR-10 and CIFAR-100 datasets, respectively.

**FL data distribution.** In the IID setup, data samples from each class are distributed equally to $M = 50$ clients. To simulate non-IID conditions across clients, we follow (Hsu et al., 2019; Nguyen et al., 2023; Chen et al., 2023) to partition the training set into $M = 50$ clients using a Dirichlet distribution with $\alpha = 0.5$. Results with different $\alpha$ are reported in the Appendix B.2.

**Calibration baselines.** We compare our method against models trained using Cross-Entropy (Uncal.), representing training without calibration. We adapt centralized train-time calibration methods, such as Focal Loss (FL) (Lin et al., 2017), Label Smoothing (LS) (Müller et al., 2019), Brier Score (BS) (Brier, 1950), MMCE (Kumar et al., 2018), FLSD (Mukhoti et al., 2020), MbLS (Liu et al., 2021), DCA (Liang et al., 2020), and MDCA (Hebbalaguppe et al., 2022b), to the FL setting based on (5) and (6) to evaluate their performance. We also compare our method with FedCal (Peng et al., 2024), a state-of-the-art method designed for post-hoc calibration in FL. Detailed parameters for each calibration baseline are provided in Appendix A.1.

**FL algorithms and setups.** We conduct model calibration on five well-known FL algorithms, including FedAvg (McMahan et al., 2017), FedProx (Sahu et al., 2018), Scaffold (Karimireddy et al., 2019), FedDyn (Acar et al., 2021), and FedNova (Wang et al., 2020b). We run each FL algorithm for 100 rounds, evaluating the final global model, with 5 epochs for each local training. We use the SGD optimizer with a learning rate of $10^{-3}$, weight decay of $10^{-4}$, and momentum of 0.9. For additional details on the training specifics of each algorithm, please see Appendix A.2.

**Auxiliary loss and similarity measures for NUCFL.** We implement our framework using two auxiliary-based calibration losses, $\ell_{dca}$ and $\ell_{mdca}$. We utilize three similarity measurements–cosine similarity (COS), linear centered kernel alignment (L-CKA), and RBF-CKA (Kornblith et al., 2019)–which output a similarity score from 0 (not similar at all) to 1 (identical) for evaluating the similarity between the learned local model and the global model. We refer to configurations using our technique as "NUCFL ($* + \odot$)," where $*$ denotes auxiliary calibration losses $\ell_{dca}/\ell_{mdca}$ and $\odot$ denotes similarity measurements COS/L-CKA/RBF-CKA, yielding six settings for our methods.

**Evaluation metrics.** To evaluate performance, we measure the accuracy (%) of the final global model on the test set and assess calibration using commonly used metrics such as Expected Calibration Error (ECE%) and Static Calibration Error (SCE%). ECE (Naeini et al., 2015) approximates calibration error (3) by dividing predictions into $I$ bins and calculating a weighted average of accuracy-confidence discrepancies: ECE $= \sum_{i=1}^{I} \frac{B_i}{N} |A_i - C_i|$, where $N$ represents the total number of samples, with weighting based on the proportion of samples within each confidence bin/interval. Each $i$-th bin covers the interval $(\frac{i-1}{I}, \frac{i}{I}]$ within the confidence range, with $B_i$ indicating the number of samples in the $i$-th bin. $A_i = \frac{1}{|B_i|} \sum_{j \in B_i} \mathbb{I}(\hat{y}_j = y_j)$ calculates the accuracy within bin $B_i$, and $C_i = \frac{1}{|B_i|} \sum_{j:\hat{s}_j \in B_i} \hat{s}_j$ gives the average predicted confidence

| Calibration Method | FedAvg | | | FedProx | | | Scaffold | | | FedDyn | | | FedNova | | |
|---|---|---|---|---|---|---|---|---|---|---|---|---|---|---|---|
| | Acc ↑ | ECE ↓ | SCE ↓ | Acc ↑ | ECE ↓ | SCE ↓ | Acc ↑ | ECE ↓ | SCE ↓ | Acc ↑ | ECE ↓ | SCE ↓ | Acc ↑ | ECE ↓ | SCE ↓ |
| Uncal. | 61.34 | 10.52 | 3.61 | 61.88 | 9.37 | 3.58 | 62.08 | 11.42 | 3.82 | 62.39 | 12.53 | 3.84 | 63.01 | 11.45 | 3.82 |
| Focal (Lin et al., 2017) | 60.59 | 12.88 | 3.74 | 62.07 | 11.45 | 3.79 | 61.39 | 13.28 | 4.88 | 60.21 | 11.22 | 3.81 | 61.15 | 13.21 | 4.71 |
| LS (Müller et al., 2019) | 59.35 | 15.61 | 6.28 | 62.23 | 14.37 | 6.14 | 62.15 | 11.69 | 4.05 | 61.34 | 15.19 | 6.58 | 63.05 | 16.03 | 6.89 |
| BS (Brier, 1950) | 61.20 | 11.32 | 3.80 | 60.43 | 10.15 | 3.61 | 60.39 | 10.92 | 3.80 | 62.17 | 13.81 | 4.91 | 60.44 | 11.39 | 4.06 |
| MMCE (Kumar et al., 2018) | 60.00 | 13.41 | 4.93 | 60.11 | 11.32 | 3.77 | 61.03 | 11.37 | 3.83 | 61.35 | 12.93 | 3.95 | 61.28 | 12.55 | 3.90 |
| FLSD (Mukhoti et al., 2020) | 58.71 | 11.51 | 3.92 | 59.28 | 10.66 | 3.81 | 60.49 | 13.61 | 5.15 | 60.94 | 11.83 | 4.04 | 60.49 | 13.81 | 4.88 |
| MbLS (Liu et al., 2021) | 59.62 | 9.42 | 3.55 | 60.39 | 11.37 | 3.84 | 61.38 | 11.99 | 4.07 | 62.30 | 14.95 | 6.50 | 62.93 | 13.66 | 4.69 |
| DCA (Liang et al., 2020) | 61.24 | 7.69 | 3.24 | 61.93 | 8.74 | 3.40 | 62.11 | 9.25 | 3.55 | 62.93 | 10.17 | 3.75 | 63.15 | 9.27 | 3.55 |
| **NUCFL** (DCA+COS) | 61.88 | 6.21 | 3.11 | 62.38 | 8.15 | 3.35 | 62.17 | 8.88 | 3.50 | 62.81 | 9.29 | 3.54 | 63.24 | 8.85 | 3.51 |
| **NUCFL** (DCA+L-CKA) | 62.05 | **6.14** | **3.07** | 62.31 | **8.04** | **3.30** | 62.25 | **8.41** | **3.45** | 62.94 | **9.14** | **3.52** | 63.27 | 8.52 | 3.43 |
| **NUCFL** (DCA+RBF-CKA) | 61.59 | 6.19 | 3.11 | 61.89 | 8.17 | 3.35 | 61.94 | 8.52 | 3.45 | 62.84 | 9.21 | 3.55 | 63.17 | **8.01** | **3.30** |
| MDCA (Hebbalaguppe et al., 2022b) | 61.03 | 7.71 | 3.29 | 62.00 | 8.21 | 3.37 | 62.23 | 9.04 | 3.51 | 62.84 | 10.24 | 3.72 | 63.29 | 10.00 | 3.71 |
| **NUCFL** (MDCA+COS) | 62.00 | 6.38 | 3.14 | 61.93 | 7.94 | 3.29 | 62.17 | 8.31 | **3.40** | 62.91 | 9.33 | 3.56 | 63.14 | 9.16 | 3.58 |
| **NUCFL** (MDCA+L-CKA) | 62.17 | 6.25 | 3.11 | 62.03 | **7.88** | **3.25** | 62.22 | **8.30** | **3.40** | 62.88 | 9.19 | 3.53 | 63.14 | 9.03 | 3.51 |
| **NUCFL** (MDCA+RBF-CKA) | 61.54 | **6.20** | **3.09** | 62.15 | 8.02 | 3.29 | 62.15 | 8.42 | 3.45 | 62.65 | 9.24 | 3.55 | 63.22 | **8.59** | **3.47** |

Table 1: Accuracy (%), calibration measures ECE (%), and SCE (%) of various federated optimization methods with different calibration methods under non-IID scenario on the CIFAR-100 dataset. Values in boldface represent the best calibration provided by our method for the auxiliary calibration method, and underlined values indicate the best calibration across all methods. Averaged performances are reported; *complete results with standard deviation are included in the Appendix.*

| Calibration Method | FedAvg | | | FedProx | | | Scaffold | | | FedDyn | | | FedNova | | |
|---|---|---|---|---|---|---|---|---|---|---|---|---|---|---|---|
| | Acc ↑ | ECE ↓ | SCE ↓ | Acc ↑ | ECE ↓ | SCE ↓ | Acc ↑ | ECE ↓ | SCE ↓ | Acc ↑ | ECE ↓ | SCE ↓ | Acc ↑ | ECE ↓ | SCE ↓ |
| Uncal. | 90.95 | 4.17 | 1.66 | 91.45 | 4.61 | 1.95 | 91.28 | 5.02 | 2.51 | 92.75 | 5.24 | 2.60 | 92.22 | 4.92 | 2.44 |
| Focal (Lin et al., 2017) | 90.63 | 4.77 | 1.96 | 90.00 | 5.39 | 2.77 | 91.13 | 5.38 | 2.72 | 91.15 | 5.84 | 2.91 | 91.13 | 5.39 | 2.63 |
| LS (Müller et al., 2019) | 91.07 | 3.75 | 1.62 | 90.37 | 4.52 | 1.90 | 91.33 | 5.18 | 2.64 | 93.08 | 5.19 | 2.63 | 92.04 | 4.95 | 2.46 |
| BS (Brier, 1950) | 91.48 | 5.19 | 2.65 | 90.98 | 5.42 | 2.80 | 88.72 | 4.19 | 1.81 | 91.39 | 5.37 | 2.62 | 90.38 | 4.65 | 1.92 |
| MMCE (Kumar et al., 2018) | 90.22 | 4.85 | 2.30 | 90.12 | 4.01 | 1.79 | 91.44 | 5.07 | 2.55 | 92.11 | 4.93 | 2.49 | 91.35 | 5.08 | 2.49 |
| FLSD (Mukhoti et al., 2020) | 90.02 | 4.93 | 2.95 | 90.39 | 4.99 | 2.04 | 92.88 | 5.19 | 2.58 | 91.17 | 5.61 | 2.84 | 91.22 | 5.23 | 2.63 |
| MbLS (Liu et al., 2021) | 90.62 | 4.27 | 1.99 | 91.49 | 4.79 | 1.94 | 92.87 | 5.39 | 2.60 | 92.06 | 5.44 | 2.75 | 90.39 | 4.61 | 1.92 |
| DCA (Liang et al., 2020) | 91.84 | 3.61 | 1.52 | 92.03 | 4.25 | 1.61 | 92.04 | 4.44 | 1.82 | 93.08 | 4.61 | 1.92 | 92.37 | 4.31 | 1.71 |
| **NUCFL** (DCA+COS) | 91.77 | 3.52 | 1.49 | 91.95 | 3.61 | 1.35 | 92.42 | 4.39 | 1.77 | 93.22 | **4.20** | **1.65** | 92.25 | 4.13 | 1.68 |
| **NUCFL** (DCA+L-CKA) | 91.63 | 3.52 | 1.47 | 92.10 | **3.60** | **1.30** | 92.19 | **4.09** | **1.60** | 93.45 | 4.22 | **1.65** | 92.33 | **4.02** | **1.54** |
| **NUCFL** (DCA+RBF-CKA) | 91.74 | **3.49** | **1.40** | 91.99 | 3.77 | 1.35 | 91.74 | 4.15 | 1.62 | 92.95 | 4.34 | 1.75 | 92.41 | 4.11 | 1.58 |
| MDCA (Hebbalaguppe et al., 2022b) | 91.64 | 3.75 | 1.53 | 92.17 | 4.42 | 2.05 | 92.95 | 4.61 | 1.90 | 93.19 | 4.71 | 1.93 | 92.05 | 4.62 | 1.82 |
| **NUCFL** (MDCA+COS) | 91.29 | 3.61 | 1.44 | 91.95 | 3.95 | 1.77 | 93.07 | **4.14** | **1.62** | 92.88 | 4.41 | 1.80 | 92.45 | 4.32 | 1.70 |
| **NUCFL** (MDCA+L-CKA) | 91.97 | **3.28** | **1.28** | 92.20 | **3.88** | **1.51** | 92.77 | 4.20 | 1.80 | 93.11 | **4.31** | **1.75** | 91.99 | 4.28 | **1.67** |
| **NUCFL** (MDCA+RBF-CKA) | 91.35 | 3.56 | 1.40 | 92.21 | 4.00 | 1.77 | 93.04 | 4.19 | 1.79 | 93.04 | 4.40 | 1.80 | 92.39 | **4.17** | **1.67** |

Table 2: Average performance of each algorithm under non-IID scenario on the FEMNIST dataset. Complete results with standard deviation are included in the Appendix.

for samples where $\hat{s}_j \in B_i$. The recently proposed SCE (Nixon et al., 2019), extending ECE for multi-class settings, groups predictions into bins by class probability, computes calibration errors per bin, and averages these across all bins: $\text{SCE} = \frac{1}{K} \sum_{i=1}^{I} \sum_{j=1}^{K} \frac{B_{i,j}}{N} |A_{i,j} - C_{i,j}|$, where $K$ represents the number of classes, and $B_{i,j}$ denotes the count of $j$-th class samples in the $i$-th bin. $A_{i,j} = \frac{1}{|B_{i,j}|} \sum_{k \in B_{i,j}} \mathbb{I}(j = y_k)$ measures accuracy, while $C_{i,j} = \frac{1}{|B_{i,j}|} \sum_{k \in B_{i,j}} s_k[j]$ gives the average confidence for the $j$-th class in the $i$-th bin. We set the number of bins, $I = 20$, for ECE and SCE, and evaluate the final global model. We performed all the experiments on four random seeds and reported the average performance along with the standard deviation.

## 5.2 EXPERIMENTAL RESULTS

**Can train-time calibration apply to federated learning?** We first compare train-time calibration in traditional centralized training and non-IID FL. Using the entire CIFAR-100 dataset, we train models for 100 epochs with various calibration methods, presenting the results in Table 6 of Appendix B.1. Most methods effectively maintain accuracy and reduce calibration errors in centralized settings. Yet, as shown in Table 1, these methods often underperform in calibration and accuracy when applying to FL. Notably, methods like DCA and MDCA, which include an auxiliary loss, show robustness in federated settings. They balance classification accuracy and calibration, preserving confidence while penalizing miscalibration, making them stand out in FL environments.

**The effects of confidence calibration on FL.** Tables 1 and 2 show accuracy and calibration error for various calibration methods applied to FL algorithms under non-IID scenarios using CIFAR-100 and FEMNIST datasets. Among these baselines, auxiliary-based methods like DCA and MDCA show superior performance, with comparable accuracy and lower error. Our method outperforms them, as evidenced by the superior performance of NUCFL (DCA/MDCA + ◎) over DCA/MDCA, highlighting the importance of our non-uniform penalty, which is specifically tailored to address global calibration needs during local training. NUCFL's superior performance, utilizing various similarity measures and auxiliary losses, demonstrates its flexibility and effectiveness. Our consistent improvements across various FL algorithms also highlight its adaptability. See Appendix B.2 for complete results with standard deviations, as well as other datasets and data distributions. Further discussion on the trade-off between accuracy and reliability can be found in Appendix B.6.

**Mitigating over/under-confidence.** The results in Tables 1 and 2 show that our method improves SCE and ECE over the baselines, but they highlight whether they correct for over-confidence or

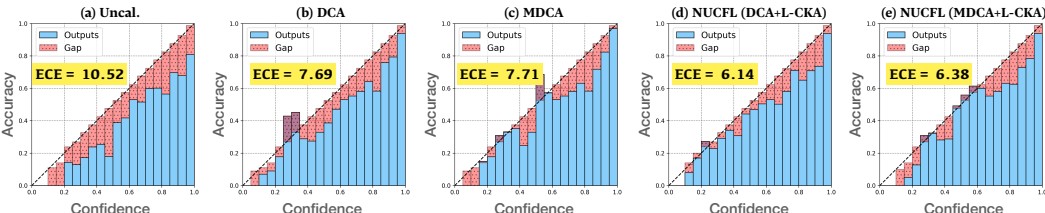

Figure 3: Reliability diagrams for non-IID FedAvg with different calibration methods using the CIFAR-100 dataset. The lower ECE and smaller gap (red region) show the effectiveness of our method.

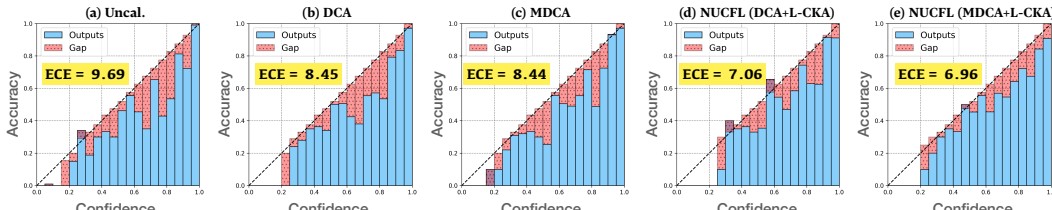

Figure 4: Reliability diagrams for non-IID FedAvg using the CIFAR-10 dataset.

under-confidence. To clarify this, we present reliability diagrams (Degroot & Fienberg, 1983; Niculescu-Mizil & Caruana, 2005), which visually compare predicted probabilities to actual outcomes, plotting expected accuracy against confidence. If the model is perfectly calibrated (i.e. if $\mathbb{P}(\hat{y} = y^* | s[\hat{y}]) = s[\hat{y}]$), then the diagram should plot the identity function. Any deviation from a perfect diagonal represents miscalibration. In Fig. 3 and 4, we present reliability histograms for non-IID FL using various calibration methods, where each FL method yields over-confident predictions. For example, if a model predicts the probability of a class $k$ from $\{1, ..., K\}$ as 0.7, it is expected to be correct 70 out of 100 cases; if the actual number of correct predictions is less than 70, the model is considered over-confident. Such over-confidence can be problematic in critical applications like medical diagnostics, potentially leading to under-treatment and serious health risks. Our method shows a smaller gap (red region) and lower ECE, effectively mitigating over-confidence. For details on how our method also tackles under-confidence, please refer to Appendix B.3.

**Effectiveness of our design.** We explore model calibration on various FL settings, analyzing the effects of different non-IID degrees $\alpha$, numbers of clients $M$, and percentages of participating clients per round. Fig. 5 shows the accuracy and calibration results of FedAvg across various configurations on the CIFAR-100 dataset, where we adjust one variable at a time from the default setting: $M = 50$, $\alpha = 0.5$, 100% participation. Across all settings, our method consistently shows robust calibration. Notably, NUCFL remains effective under challenging conditions, such as with lower $\alpha$ values (higher non-IID disparity) or fewer participant per round. Additional results under diverse conditions, including an even smaller number of partial participants under various data distributions, are provided in Appendix B.4, highlighting the adaptability of our approach in complex FL environments.

**Calibration strategy validation.** Our calibration strategy assigns higher auxiliary penalty loss to local models that deviate more from the global model, addressing global calibration needs during local training. To validate our design, we test an opposite logic, applying higher penalties to clients more similar to the global model. We

| Calibration Method | Non-IID FedAvg | | |
|---|---|---|---|
| | Acc ↑ | ECE ↓ | SCE ↓ |
| NUCFL (DCA+L-CKA) | 62.05 | **6.14** | **3.07** |
| NUCFL (MDCA+L-CKA) | 62.17 | 6.25 | 3.11 |
| Reversed NUCFL (DCA+L-CKA) | 60.36 | 13.89 | 4.98 |
| Reversed NUCFL (MDCA+L-CKA) | 61.77 | 16.11 | 6.87 |

Table 3: Destructive experiment for our method.

adjust the weight $\beta_m$ for each client $m$ to $\texttt{sim}(\delta^{(t-1)}, \delta_m^{(t,e)})^{-1}$. We conducted this experiment to assess whether reversing our calibration logic affects FL performance and to validate our original design. In Table 3, we applied a reverse penalty strategy to optimal configurations of our method (i.e., NUCFL (DCA/MDCA + L-CKA)) on the CIFAR-100 dataset. The results show that reversed logic degrades performance by biasing calibration towards local heterogeneity while neglecting global needs, even performing worse than uncalibrated FL (Table 1). This confirms the soundness of our design, which strategically prioritizes global calibration needs during local training.

**Comparison with FedCal** (Peng et al., 2024). We now conduct comparisons with the only existing model calibration framework for FL, FedCal (Peng et al., 2024). FedCal bases its algorithm on

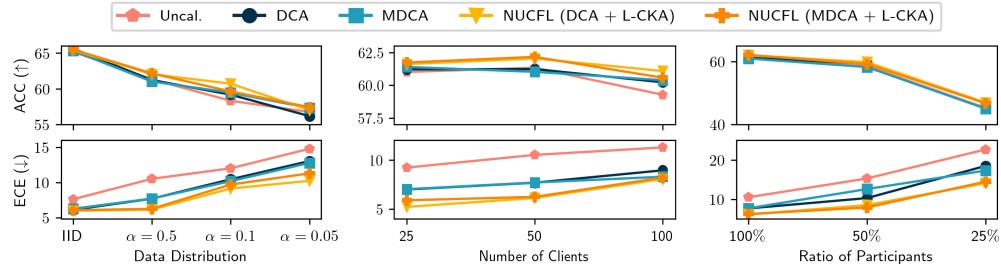

Figure 5: Comparison of confidence calibration across different FL settings using the CIFAR-100 dataset.

FedAvg; thus, we display the calibration results of our method alongside the baseline using FedAvg under non-IID conditions ($\alpha = 0.5$) on CIFAR-100 datatset in Table 4.

For the scaler used in their work, we adhered to their design, employing an MLP with layer sizes of 100-64-64-100. We see that our proposed NUCFL shows superior performance compared to FedCal. While using a scaler aggregated from local clients can reduce global ECE, it may neglects the interactions between global and local

| Calibration Method | Non-IID FedAvg | | |
|---|---|---|---|
| | Acc ↑ | ECE ↓ | SCE ↓ |
| Uncal. | 61.34 | 10.52 | 3.61 |
| FedCal (Peng et al., 2024) | 61.34 | 8.80 | 3.49 |
| NUCFL (DCA+L-CKA) | **62.05** | **6.14** | **3.07** |
| NUCFL (MDCA+RBF-CKA) | **61.54** | **6.20** | **3.09** |

Table 4: Comparison with FedCal.

calibration needs. Our method considers these interactions, minimizing local biases and better meeting global calibration requirements. More comparisons using other datasets and data distribution are available in Appendix B.4. The superiority of our method in both accuracy and calibration error shows NUCFL's effectiveness in complex FL environments.

**Compatibility with post-hoc calibration.** Temperature scaling (TS) (Guo et al., 2017) is one of the most effective post-hoc calibration methods, which adjusts a model's logits by a scaling factor to calibrate confidence scores without affecting accuracy. Like all post-hoc methods, TS requires a hold-off dataset to tune the model and learn the calibration parameter (i.e., temperature). However, this need conflicts with the data privacy policies of FL, making TS impractical in such settings. For this analysis, we set aside these privacy concerns to compare with TS, assuming that data from each client is available at the server. Under these assumptions, we conduct non-IID ($\alpha = 0.5$) FedAvg and sample five images

| Calibration Method | Non-IID FedAvg | | |
|---|---|---|---|
| | Acc ↑ | ECE ↓ | SCE ↓ |
| Uncal. | 61.34 | 10.52 | 3.61 |
| DCA | 61.24 | 7.69 | 3.24 |
| MDCA | 61.03 | 7.71 | 3.29 |
| NUCFL (DCA+L-CKA) | **62.05** | **6.14** | **3.07** |
| NUCFL (MDCA+L-CKA) | **62.17** | **6.25** | **3.11** |
| TS (Guo et al., 2017) | 61.34 | 8.22 | 3.43 |
| DCA+TS | 61.24 | 7.00 | 3.03 |
| MDCA+TS | 61.03 | 7.29 | 3.11 |
| NUCFL (DCA+L-CKA)+TS | **62.05** | **5.91** | **2.97** |
| NUCFL (MDCA+L-CKA)+TS | **62.17** | **6.03** | **3.00** |

Table 5: Comparison and integration with post-hoc calibration. The results demonstrate our method's compatibility with post-hoc calibration.

from each client to create a server dataset for post-hoc calibration. Table 5 compares train-time and post-hoc calibration methods using the CIFAR-100 dataset. Results show TS outperforming some train-time methods, like MDCA, while our method still provides superiority in both accuracy and calibration error. Recall from **Remark 2** that our NUCFL can also be complemented by post-hoc methods when a holdout dataset is available for refining the global model. We apply post-hoc calibration on the global model using TS after train-time calibration during local training. This combined approach shows improved performance over train-time calibration alone. Notably, integrating TS with NUCFL yields the best results, significantly reducing error.

## 6 CONCLUSION

We conducted one of the earliest systematic study on model calibration for FL and introduced a novel framework, NUCFL, designed to ensure a well-calibrated global model. NUCFL measures the similarity between client and server models and dynamically adjusts calibration penalties for each client, addressing both heterogeneity and global calibration needs effectively. Through extensive experiments, NUCFL demonstrated significant improvements over existing calibration methods, proving its adaptability with various types of auxiliary calibration penalties and compatibility across different FL algorithms. Our research seeks to deepen understanding of the underlying effects of model calibration on FL, enhancing the trustworthiness of standard FL systems. Future work could explore calibration for personalized FL settings with personalized layers, unsupervised FL settings with dynamically adjusted calibration weighting for domain adaptation, and the use of similarity-based weighting for other auxiliary losses to enhance FL.

ACKNOWLEDGMENTS

This work was supported in part by the National Science Foundation (NSF) under grants CNS-2146171 and CPS-2313109, by the Office of Naval Research (ONR) under grant N00014-22-1-2305, and by the Air Force Office of Scientific Research (AFOSR) under grant FA9550-24-1-0083.

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

# A  DETAILED SETTINGS FOR CALIBRATION BASELINES AND FEDERATED ALGORITHMS

## A.1  CALIBRATION BASELINES

We offer an overview of each compared calibration method, including the hyperparameter settings utilized during training. Focal loss (Lin et al., 2017) is defined as $\text{FL} = -\sum_{i=1}^{N}(1 - p_{i,y_i})^{\gamma} \cdot \log(p_{i,y_i})$, where $p_{i,y_i}$ is the predicted confidence score for sample $i$ and $\gamma$ is a hyper-parameter. We trained models with $\gamma \in \{1, 2\}$. Label Smoothing (LS) (Müller et al., 2019) is defined as $\text{LS} = -\sum_{i=1}^{N}\sum_{j=1}^{K} q_{i,j} \cdot \log(p_{i,j})$, where $p_{i,j}$ is the predicted confidence score for class $j$ of sample $i$, and $q_i$ is the soft target vector for sample $i$, such that $q_{i,j} = \frac{\alpha}{K-1}$ if $j \neq y_i$, else $q_{i,j} = (1 - \alpha)$. $\alpha$ is a hyperparameter set at 0.1, following established literature settings (Hebbalaguppe et al., 2022b). For the Brier Score (BS) (Brier, 1950), we train using a loss defined as the squared difference between the predicted probability vector and the one-hot target vector. For Maximum Mean Calibration Error (MMCE) (Kumar et al., 2018), we use the weighted MMCE, with $\lambda$ selected from $\{2, 4\}$, as a regularization term along with cross-entropy. For FLSD (Mukhoti et al., 2020), we use $\gamma = 3$, reported as the best configuration in their paper. For Margin-based Label Smoothing (MbLS) (Liu et al., 2021), we set the margin at 10 and $\mu = 0.1$, configurations reported as optimal in their paper. For DCA (Liang et al., 2020), we train using a loss composed of cross-entropy plus $\beta \cdot \ell_{dca}$, where $\beta$ is a hyperparameter selected from $\{1, 5, 10\}$ as demonstrated in their paper. Similarly, for MDCA (Hebbalaguppe et al., 2022b), we apply the cross-entropy loss enhanced by $\beta \cdot \ell_{mdca}$, with $\beta$ also chosen from $\{1, 5, 10\}$ as well. For all baselines, we report results that show the lowest ECE among their hyperparameter settings in our paper.

## A.2  FEDERATED ALGORITHMS

For all FL algorithms, we distributed the training data across $M = 50$ clients and performed 100 rounds of training. Each client conducted 5 local training epochs per round. The batch size was set at 64, and we explored learning rates within the range [1e-2, 5e-3, 1e-3, 5e-4]. For FedAvg (McMahan et al., 2017), clients perform standard stochastic gradient descent (SGD) updates as `ClientOPT` and upload their gradients to the central server. On the server side, the global model is updated using SGD as `ServerOPT` with the received gradients from the clients. FedProx (Sahu et al., 2018) introduces a proximal term during local client training as their `ClientOPT`, to address data heterogeneity issues by preventing significant drifts in client model updates. It employs a weighted average for aggregation and updates the global model using SGD, similar to `ServerOPT` in FedAvg. We set the proximal term coefficient parameter $\mu = 1$ for the proximal term coefficient in FedProx. Scaffold (Karimireddy et al., 2019) employs client-variance reduction as its `ClientOPT` and corrects model drifts in local updates by introducing control variates. For `ServerOPT`, Scaffold incorporates the clients' gradients and control variates for aggregation, and updates the global model using SGD. FedDyn (Acar et al., 2021) dynamically adjusts the local objectives by incorporating two regularizers as `ClientOPT` to ensure that the local optimum aligns asymptotically with the stationary points of the global objective. We set the parameter $\alpha = 0.1$ as commonly used in the literature, and the local model is updated using SGD. Additionally, they introduce their server update as `ServerOPT` through the use of server state vectors. FedNova (Wang et al., 2020b) uses normalized stochastic gradients as `ClientOPT` to compensate for data imbalances across clients, and then introduces a weighted rescaling process as their `ServerOPT`. For applying `NUCFL` to FL algorithms, we provide a detail framework of our `NUCFL` in Algorithm 2. We run all experiments on a 3-GPU cluster of Tesla V100 GPUs, with each GPU having 32GB of memory.

# B  ADDITIONAL EXPERIMENTS AND ANALYSES

## B.1  CALIBRATION ON CENTRALIZED LEARNING

We present train-time calibration methods for traditional centralized training in Table 6. We train ResNet-34 using the SGD optimizer on the entire CIFAR-100 training dataset for 100 epochs with various train-time calibration methods. We found that most of these methods effectively calibrate in centralized settings, maintaining accuracy and exhibiting lower calibration errors.

| Calibration Method | MNIST (CNN) | | | FEMNIST (CNN) | | | CIFAR-10 (AlexNet) | | | CIFAR-100 (ResNet-34) | | |
|---|---|---|---|---|---|---|---|---|---|---|---|---|
| | Acc ↑ | ECE ↓ | SCE ↓ | Acc ↑ | ECE ↓ | SCE ↓ | Acc ↑ | ECE ↓ | SCE ↓ | Acc ↑ | ECE ↓ | SCE ↓ |
| Uncal. | 97.13 | 0.74 | 0.26 | 94.29 | 3.07 | 1.58 | 84.19 | 5.53 | 2.55 | 68.33 | 5.80 | 3.00 |
| Focal (Lin et al., 2017) | 97.09 | 0.70 | 0.25 | 94.33 | 2.98 | 1.55 | 84.17 | 5.01 | 2.43 | 68.42 | 5.66 | 2.88 |
| LS (Müller et al., 2019) | 97.22 | 0.61 | 0.23 | 94.51 | 2.89 | 1.53 | 84.69 | 5.27 | 2.46 | 68.19 | 5.59 | 2.85 |
| BS (Brier, 1950) | 96.99 | 0.70 | 0.24 | 95.02 | 3.00 | 1.59 | 84.23 | 5.44 | 2.53 | 68.59 | 5.60 | 2.87 |
| MMCE (Kumar et al., 2018) | 97.23 | 0.55 | 0.20 | 94.31 | 2.71 | 1.50 | 84.33 | 4.87 | 2.35 | 68.69 | 5.44 | 2.55 |
| FLSD (Mukhoti et al., 2020) | 97.11 | 0.70 | 0.24 | 94.11 | 2.95 | 1.54 | 84.11 | 2.88 | 1.54 | 68.44 | 5.69 | 2.88 |
| MbLS (Liu et al., 2021) | 97.55 | 0.54 | 0.20 | 94.30 | 2.66 | 1.41 | 84.29 | 3.66 | 2.04 | 68.71 | 4.14 | 1.96 |
| DCA (Liang et al., 2020) | 97.29 | 0.52 | 0.20 | 94.33 | 2.59 | 1.39 | 84.20 | 3.71 | 2.05 | 68.45 | 4.29 | 1.98 |
| MDCA (Hebbalaguppe et al., 2022b) | 94.22 | 0.55 | 0.20 | 94.37 | 2.70 | 1.43 | 84.22 | 3.82 | 2.08 | 68.44 | 4.19 | 1.98 |

Table 6: Performance of centralized training with various calibration methods across different datasets. Train-time calibration methods demonstrate improvements in calibration error compared to the uncalibrated (Uncal.) model.

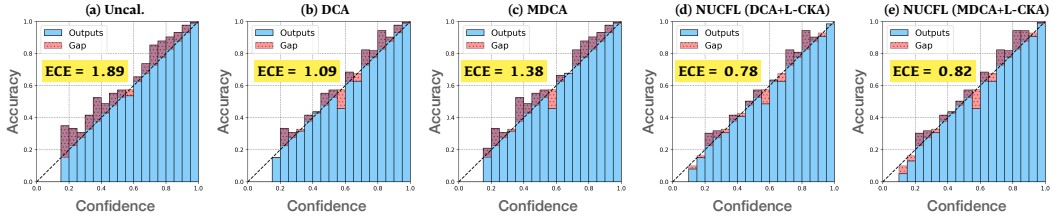

Figure 6: Reliability diagrams for non-IID FedAvg ($\alpha = 0.5$) with different calibration methods using the MNIST dataset. The lower ECE and smaller gap show the effectiveness of our method.

## B.2 ADDITIONAL RESULTS OF CALIBRATION ON FL

This section explores the performance of each calibration method on different datasets and data distributions for various FL algorithms. To ensure consistency in our experiments, most of the configuration remains unchanged. Specifically, we set the number of clients ($M$) to 50 and the total number of federated rounds ($T$) to 100, with each client performing $E = 5$ epochs of local training. Additionally, the hyperparameter search space and the details for each FL algorithm are consistent, as detailed in Appendix A. Tables 7, 8, 9, and 10 display the performance and calibration metrics of various FL algorithms on the MNIST dataset across IID and non-IID conditions, with $\alpha$ values of 0.5, 0.1, and 0.05, respectively. Tables 11, 12, 13, and 14 show the performance using FEMNIST dataset across IID and non-IID conditions. Similarly, Tables 15, 16, 17, and 18 present the performance and calibration metrics of different FL algorithms on the CIFAR-10 dataset under IID and non-IID conditions, with corresponding $\alpha$ values of 0.5, 0.1, and 0.05, respectively. Finally, Tables 19, 20, 21, and 22 show the performance using CIFAR-100 dataset across IID and non-IID conditions. Table 26 shows the calibration results with different local epoch numbers for non-IID ($\alpha = 0.5$) FedAvg using CIFAR-100 dataset. Across these experimental results, which consider various data distribution setups and different datasets, our generic framework NUCFL consistently demonstrates superiority over the baseline by achieving lower calibration error. The performance improvements of NUCFL with different configurations, such as various similarity measurements and different auxiliary calibration losses, highlight the effectiveness and flexibility of our method.

In addition to the widely used FL algorithms introduced in Section 5, we conducted additional experiments using FedSpeed (Sun et al., 2023), FedSAM (Qu et al., 2022), FedMR (Hu et al., 2023), and FedCross (Hu et al., 2022) which are considered as the SOTA FL optimization methods. We conducted our experiments using the CIFAR-100 dataset under a non-IID ($\alpha = 0.5$) data distribution, following a setup similar to that in Table 1. For the hyperparameter of FedSpeed, we selected $\lambda = 0.1$ as the default value provided by the authors, and we selected $\alpha = 0.99$ for FedCross. In Table 25, we see that in the absence of any calibration, these advanced FL algorithms do improve accuracy compared to FedAvg (ACC: 61.34; ECE: 10.52). However, they also introduce higher calibration errors. By applying NUCFL to these advanced FL algorithms, we find that our approach effectively reduces calibration error and even outperforms the SOTA calibration method in terms of calibration results. These results also demonstrate the adaptability of our method, showing that it can be applied to any FL algorithm to improve calibration while maintaining accuracy.

| Calibration Method | FedAvg (McMahan et al., 2017) | | | FedProx (Sahu et al., 2018) | | | Scaffold (Karimireddy et al., 2019) | | |
|---|---|---|---|---|---|---|---|---|---|
| | Acc ↑ | ECE ↓ | SCE ↓ | Acc ↑ | ECE ↓ | SCE ↓ | Acc ↑ | ECE ↓ | SCE ↓ |
| Uncal. | 96.20 ± 0.41 | 0.88 ± 0.15 | 0.29 ± 0.08 | 96.17 ± 0.52 | 1.04 ± 0.14 | 0.51 ± 0.08 | 96.41 ± 1.02 | 1.89 ± 0.24 | 0.66 ± 0.10 |
| Focal (Lin et al., 2017) | 95.23 ± 0.52 | 2.71 ± 0.18 | 0.66 ± 0.10 | 97.10 ± 0.77 | 4.85 ± 1.10 | 1.11 ± 0.95 | 96.28 ± 1.22 | 2.85 ± 0.95 | 0.81 ± 0.44 |
| LS (Müller et al., 2019) | 96.30 ± 0.29 | 8.66 ± 1.88 | 3.31 ± 0.32 | 95.11 ± 0.95 | 7.49 ± 0.92 | 2.95 ± 0.24 | 97.00 ± 1.02 | 6.88 ± 1.61 | 2.67 ± 0.14 |
| BS (Brier, 1950) | 95.29 ± 0.98 | 2.74 ± 0.27 | 0.67 ± 0.10 | 95.14 ± 0.33 | 3.71 ± 0.10 | 0.76 ± 0.59 | 95.43 ± 0.61 | 2.18 ± 0.43 | 0.51 ± 0.12 |
| MMCE (Kumar et al., 2018) | 97.61 ± 0.62 | 4.88 ± 0.20 | 1.14 ± 0.11 | 97.23 ± 0.53 | 5.60 ± 0.33 | 1.18 ± 0.11 | 94.77 ± 1.22 | 12.19 ± 0.19 | 3.79 ± 0.10 |
| FLSD (Mukhoti et al., 2020) | 95.23 ± 0.33 | 8.64 ± 0.17 | 3.30 ± 0.09 | 95.44 ± 0.52 | 9.02 ± 0.23 | 3.42 ± 0.11 | 95.52 ± 0.24 | 10.60 ± 0.25 | 2.08 ± 0.14 |
| MbLS (Liu et al., 2021) | 95.50 ± 1.02 | 3.01 ± 1.30 | 1.10 ± 0.29 | 95.61 ± 0.66 | 4.11 ± 0.44 | 1.10 ± 0.25 | 96.31 ± 0.85 | 3.05 ± 0.47 | 0.77 ± 0.11 |
| DCA (Liang et al., 2020) | 96.43 ± 0.45 | 0.52 ± 0.08 | 0.20 ± 0.06 | 97.02 ± 0.50 | 0.68 ± 0.10 | 0.22 ± 0.07 | 97.71 ± 1.02 | 0.70 ± 0.11 | 0.23 ± 0.07 |
| NUCFL (DCA+COS) | 97.10 ± 0.52 | 0.49 ± 0.08 | 0.18 ± 0.05 | 96.94 ± 0.77 | 0.59 ± 0.09 | 0.19 ± 0.06 | 97.00 ± 1.22 | 0.69 ± 0.10 | 0.23 ± 0.07 |
| NUCFL (DCA+L-CKA) | 96.90 ± 0.41 | **0.44** ± 0.21 | **0.16** ± 0.10 | 97.07 ± 0.52 | 0.45 ± 0.07 | **0.15** ± 0.11 | 96.84 ± 1.02 | 0.56 ± 0.09 | **0.20** ± 0.06 |
| NUCFL (DCA+RBF-CKA) | 96.01 ± 0.52 | **0.44** ± 0.07 | 0.17 ± 0.05 | 96.91 ± 0.77 | **0.43** ± 0.20 | 0.16 ± 0.13 | 96.93 ± 1.22 | **0.53** ± 0.08 | 0.21 ± 0.06 |
| MDCA (Hebbalaguppe et al., 2022b) | 96.87 ± 0.41 | 0.61 ± 0.09 | 0.24 ± 0.07 | 97.04 ± 0.52 | 0.85 ± 0.13 | 0.30 ± 0.08 | 97.04 ± 1.02 | 0.92 ± 0.14 | 0.33 ± 0.09 |
| NUCFL (MDCA+COS) | 97.11 ± 0.64 | 0.54 ± 0.10 | **0.20** ± 0.09 | 96.95 ± 0.22 | 0.56 ± 0.10 | **0.19** ± 0.09 | 96.97 ± 1.00 | 0.61 ± 0.83 | 0.22 ± 0.20 |
| NUCFL (MDCA+L-CKA) | 96.89 ± 0.85 | 0.52 ± 0.27 | **0.20** ± 0.09 | 97.02 ± 0.25 | **0.55** ± 0.49 | 0.20 ± 0.10 | 97.11 ± 1.04 | **0.55** ± 0.08 | **0.20** ± 0.02 |
| NUCFL (MDCA+RBF-CKA) | 97.01 ± 0.95 | 0.55 ± 0.11 | 0.21 ± 0.06 | 96.97 ± 0.57 | 0.64 ± 0.14 | 0.23 ± 0.06 | 97.54 ± 0.83 | 0.70 ± 0.25 | 0.24 ± 0.07 |

| Calibration Method | FedDyn (Acar et al., 2021) | | | FedNova (Wang et al., 2020b) | | |
|---|---|---|---|---|---|---|
| | Acc ↑ | ECE ↓ | SCE ↓ | Acc ↑ | ECE ↓ | SCE ↓ |
| Uncal. | 97.23 ± 1.22 | 1.40 ± 0.69 | 0.55 ± 0.13 | 96.42 ± 1.04 | 0.89 ± 0.49 | 0.28 ± 0.11 |
| Focal (Lin et al., 2017) | 97.61 ± 0.54 | 4.37 ± 0.85 | 1.09 ± 0.10 | 95.23 ± 0.92 | 3.82 ± 0.94 | 0.80 ± 0.09 |
| LS (Müller et al., 2019) | 95.11 ± 0.59 | 5.02 ± 1.02 | 2.50 ± 0.20 | 96.61 ± 0.99 | 10.42 ± 1.31 | 3.62 ± 0.81 |
| BS (Brier, 1950) | 92.63 ± 1.04 | 8.49 ± 0.44 | 0.88 ± 0.09 | 96.10 ± 1.00 | 3.13 ± 0.17 | 0.70 ± 0.07 |
| MMCE (Kumar et al., 2018) | 97.10 ± 0.79 | 3.04 ± 0.26 | 0.79 ± 0.07 | 96.10 ± 0.82 | 5.44 ± 0.41 | 1.12 ± 0.15 |
| FLSD (Mukhoti et al., 2020) | 95.41 ± 0.39 | 3.29 ± 0.31 | 0.72 ± 0.11 | 95.44 ± 0.41 | 7.63 ± 0.52 | 2.81 ± 0.15 |
| MbLS (Liu et al., 2021) | 95.22 ± 0.92 | 3.09 ± 0.33 | 0.79 ± 0.10 | 97.07 ± 0.85 | 4.17 ± 0.42 | 1.06 ± 0.17 |
| DCA (Liang et al., 2020) | 97.39 ± 1.22 | 0.57 ± 0.09 | 0.30 ± 0.08 | 97.11 ± 1.04 | 0.64 ± 0.10 | 0.27 ± 0.07 |
| NUCFL (DCA+COS) | 97.61 ± 0.54 | 0.54 ± 0.08 | 0.25 ± 0.07 | 97.42 ± 0.92 | 0.57 ± 0.09 | 0.29 ± 0.08 |
| NUCFL (DCA+L-CKA) | 97.53 ± 1.22 | 0.46 ± 0.07 | **0.17** ± 0.09 | 97.18 ± 1.04 | **0.26** ± 0.13 | **0.16** ± 0.05 |
| NUCFL (DCA+RBF-CKA) | 96.75 ± 0.54 | **0.44** ± 0.07 | **0.17** ± 0.09 | 97.63 ± 0.92 | 0.29 ± 0.05 | 0.17 ± 0.05 |
| MDCA (Hebbalaguppe et al., 2022b) | 97.41 ± 1.22 | 0.65 ± 0.10 | 0.28 ± 0.08 | 97.04 ± 1.04 | 0.75 ± 0.11 | 0.28 ± 0.08 |
| NUCFL (MDCA+COS) | 96.93 ± 0.25 | 0.51 ± 1.00 | **0.25** ± 0.09 | 96.99 ± 1.11 | 0.62 ± 0.22 | 0.26 ± 0.10 |
| NUCFL (MDCA+L-CKA) | 97.09 ± 0.25 | **0.50** ± 0.33 | 0.26 ± 0.06 | 97.14 ± 1.00 | **0.54** ± 0.38 | **0.24** ± 0.09 |
| NUCFL (MDCA+RBF-CKA) | 97.86 ± 0.61 | 0.55 ± 0.09 | 0.26 ± 0.06 | 97.58 ± 0.52 | 0.60 ± 0.23 | 0.26 ± 0.10 |

Table 7: Accuracy (%), calibration measures ECE (%), and SCE (%) of various federated optimization methods with different calibration methods under IID scenario on the MNIST dataset.

| Calibration Method | FedAvg (McMahan et al., 2017) | | | FedProx (Sahu et al., 2018) | | | Scaffold (Karimireddy et al., 2019) | | |
|---|---|---|---|---|---|---|---|---|---|
| | Acc ↑ | ECE ↓ | SCE ↓ | Acc ↑ | ECE ↓ | SCE ↓ | Acc ↑ | ECE ↓ | SCE ↓ |
| Uncal. | 95.02 ± 0.41 | 1.89 ± 0.15 | 0.65 ± 0.08 | 94.55 ± 0.52 | 2.04 ± 0.14 | 0.65 ± 0.08 | 94.15 ± 1.02 | 4.54 ± 0.24 | 1.11 ± 0.10 |
| Focal (Lin et al., 2017) | 94.38 ± 0.52 | 4.46 ± 0.18 | 1.25 ± 0.10 | 93.44 ± 0.77 | 4.17 ± 1.10 | 1.20 ± 0.95 | 93.51 ± 1.22 | 7.65 ± 0.95 | 1.52 ± 0.44 |
| LS (Müller et al., 2019) | 94.53 ± 0.29 | 8.00 ± 1.88 | 3.30 ± 0.32 | 95.15 ± 0.95 | 3.91 ± 0.92 | 1.20 ± 0.24 | 95.25 ± 1.02 | 5.66 ± 1.31 | 4.50 ± 0.14 |
| BS (Brier, 1950) | 93.26 ± 0.98 | 4.02 ± 0.27 | 1.17 ± 0.10 | 92.58 ± 0.33 | 4.78 ± 0.10 | 1.26 ± 0.59 | 92.77 ± 0.61 | 4.44 ± 0.43 | 1.10 ± 0.12 |
| MMCE (Kumar et al., 2018) | 95.43 ± 0.62 | 7.82 ± 0.20 | 2.61 ± 0.11 | 93.74 ± 0.53 | 6.56 ± 0.33 | 1.95 ± 0.11 | 88.48 ± 1.22 | 7.88 ± 0.19 | 1.58 ± 0.10 |
| FLSD (Mukhoti et al., 2020) | 93.95 ± 0.33 | 8.46 ± 0.17 | 3.48 ± 0.09 | 92.29 ± 0.52 | 8.81 ± 0.23 | 3.88 ± 0.11 | 92.41 ± 0.24 | 7.38 ± 0.25 | 1.53 ± 0.14 |
| MbLS (Liu et al., 2021) | 94.22 ± 1.02 | 5.99 ± 1.30 | 1.83 ± 0.29 | 93.46 ± 0.66 | 4.84 ± 0.44 | 1.25 ± 0.25 | 95.55 ± 0.85 | 6.20 ± 0.47 | 2.41 ± 0.11 |
| DCA (Liang et al., 2020) | 95.94 ± 0.40 | 1.09 ± 0.10 | 0.49 ± 0.05 | 95.04 ± 0.50 | 1.87 ± 0.12 | 0.59 ± 0.06 | 95.42 ± 0.60 | 3.75 ± 0.08 | 1.08 ± 0.07 |
| NUCFL (DCA+COS) | 96.16 ± 0.35 | 0.81 ± 0.05 | 0.34 ± 0.03 | 95.16 ± 0.45 | **1.66** ± 0.04 | 0.56 ± 0.05 | 95.70 ± 0.55 | 2.91 ± 0.06 | 0.85 ± 0.08 |
| NUCFL (DCA+L-CKA) | 95.97 ± 0.35 | 0.78 ± 0.04 | 0.31 ± 0.02 | 95.58 ± 0.45 | 1.69 ± 0.05 | 0.56 ± 0.03 | 95.43 ± 0.55 | **2.78** ± 0.07 | **0.81** ± 0.08 |
| NUCFL (DCA+RBF-CKA) | 96.07 ± 0.37 | **0.74** ± 0.04 | **0.30** ± 0.02 | 95.67 ± 0.45 | **1.66** ± 0.04 | **0.55** ± 0.03 | 95.85 ± 0.55 | 3.05 ± 0.06 | 0.87 ± 0.08 |
| MDCA (Hebbalaguppe et al., 2022b) | 95.92 ± 0.41 | 1.38 ± 0.12 | 0.55 ± 0.06 | 94.69 ± 0.52 | 1.93 ± 0.14 | 0.62 ± 0.08 | 94.82 ± 1.02 | 3.26 ± 0.15 | 1.04 ± 0.09 |
| NUCFL (MDCA+COS) | 95.80 ± 0.40 | 0.88 ± 0.05 | **0.42** ± 0.03 | 95.18 ± 0.50 | 1.69 ± 0.04 | 0.57 ± 0.04 | 95.17 ± 0.60 | 3.01 ± 0.06 | 0.87 ± 0.07 |
| NUCFL (MDCA+L-CKA) | 95.95 ± 0.47 | **0.82** ± 0.09 | 0.43 ± 0.02 | 95.61 ± 0.50 | **1.64** ± 0.05 | **0.55** ± 0.03 | 94.99 ± 0.54 | **2.74** ± 0.06 | **0.80** ± 0.08 |
| NUCFL (MDCA+RBF-CKA) | 95.94 ± 0.41 | 0.85 ± 0.05 | **0.42** ± 0.03 | 94.58 ± 0.58 | 1.67 ± 0.10 | 0.58 ± 0.10 | 95.08 ± 0.75 | 2.83 ± 0.06 | 0.68 ± 0.08 |

| Calibration Method | FedDyn (Acar et al., 2021) | | | FedNova (Wang et al., 2020b) | | |
|---|---|---|---|---|---|---|
| | Acc ↑ | ECE ↓ | SCE ↓ | Acc ↑ | ECE ↓ | SCE ↓ |
| Uncal. | 96.87 ± 1.22 | 1.78 ± 0.69 | 0.61 ± 0.13 | 95.05 ± 1.04 | 1.54 ± 0.49 | 0.54 ± 0.11 |
| Focal (Lin et al., 2017) | 96.78 ± 0.54 | 2.21 ± 0.85 | 0.77 ± 0.10 | 94.56 ± 0.92 | 4.38 ± 0.94 | 1.10 ± 0.09 |
| LS (Müller et al., 2019) | 97.80 ± 0.59 | 5.65 ± 1.02 | 2.96 ± 0.20 | 96.48 ± 0.99 | 3.41 ± 1.31 | 1.94 ± 0.81 |
| BS (Brier, 1950) | 93.68 ± 1.04 | 5.81 ± 0.44 | 2.93 ± 0.09 | 92.96 ± 1.00 | 3.65 ± 0.17 | 1.98 ± 0.07 |
| MMCE (Kumar et al., 2018) | 96.48 ± 0.79 | 2.08 ± 0.26 | 0.74 ± 0.07 | 96.30 ± 0.82 | 6.75 ± 0.41 | 2.57 ± 0.15 |
| FLSD (Mukhoti et al., 2020) | 94.84 ± 0.39 | 2.49 ± 0.31 | 0.74 ± 0.11 | 93.93 ± 0.41 | 5.57 ± 0.52 | 2.45 ± 0.15 |
| MbLS (Liu et al., 2021) | 95.71 ± 0.92 | 3.95 ± 0.33 | 1.06 ± 0.10 | 94.24 ± 0.85 | 5.67 ± 0.42 | 2.82 ± 0.17 |
| DCA (Liang et al., 2020) | 96.47 ± 0.70 | 1.44 ± 0.09 | 0.47 ± 0.03 | 95.13 ± 0.50 | 0.94 ± 0.08 | 0.48 ± 0.05 |
| NUCFL (DCA+COS) | 96.93 ± 0.65 | 1.24 ± 0.07 | **0.43** ± 0.04 | 96.01 ± 0.55 | 0.78 ± 0.05 | 0.45 ± 0.03 |
| NUCFL (DCA+L-CKA) | 97.07 ± 0.65 | **1.23** ± 0.06 | **0.43** ± 0.02 | 96.05 ± 0.55 | **0.77** ± 0.05 | **0.44** ± 0.03 |
| NUCFL (DCA+RBF-CKA) | 97.03 ± 0.65 | 1.31 ± 0.05 | 0.47 ± 0.02 | 96.00 ± 0.55 | 0.85 ± 0.06 | 0.45 ± 0.03 |
| MDCA (Hebbalaguppe et al., 2022b) | 97.00 ± 1.22 | 1.40 ± 0.16 | 0.47 ± 0.04 | 95.10 ± 1.04 | 1.30 ± 0.18 | 0.50 ± 0.07 |
| NUCFL (MDCA+COS) | 96.64 ± 0.70 | 1.33 ± 0.08 | 0.45 ± 0.04 | 96.02 ± 0.60 | 0.82 ± 0.06 | 0.47 ± 0.03 |
| NUCFL (MDCA+L-CKA) | 97.10 ± 0.77 | **1.30** ± 0.22 | **0.44** ± 0.08 | 96.50 ± 0.65 | **0.80** ± 0.10 | **0.45** ± 0.09 |
| NUCFL (MDCA+RBF-CKA) | 96.66 ± 0.70 | 1.48 ± 0.05 | 0.49 ± 0.02 | 96.19 ± 0.56 | 0.85 ± 0.11 | 0.47 ± 0.10 |

Table 8: Accuracy (%), calibration measures ECE (%), and SCE (%) of various federated optimization methods with different calibration methods under non-IID ($\alpha = 0.5$) scenario on the MNIST dataset. Underlined values indicate the best calibration across all methods.

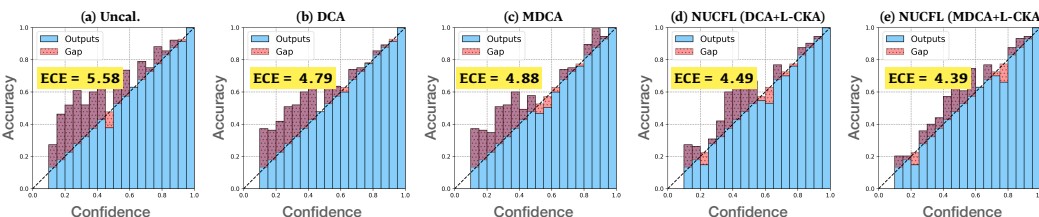

Figure 7: Reliability diagrams for non-IID FedAvg ($\alpha = 0.5$) using the FEMNIST dataset.

| Calibration Method | FedAvg (McMahan et al., 2017) | | | FedProx (Sahu et al., 2018) | | | Scaffold (Karimireddy et al., 2019) | | |
|---|---|---|---|---|---|---|---|---|---|
| | Acc ↑ | ECE ↓ | SCE ↓ | Acc ↑ | ECE ↓ | SCE ↓ | Acc ↑ | ECE ↓ | SCE ↓ |
| Uncal. | 92.68 ± 0.35 | 2.96 ± 0.22 | 1.04 ± 0.14 | 91.26 ± 0.63 | 4.21 ± 0.28 | 1.20 ± 0.11 | 92.00 ± 1.18 | 6.03 ± 0.37 | 1.92 ± 0.18 |
| Focal (Lin et al., 2017) | 90.25 ± 0.68 | 7.02 ± 0.27 | 2.58 ± 0.15 | 91.11 ± 0.95 | 6.93 ± 1.35 | 1.99 ± 0.55 | 88.24 ± 0.82 | 6.49 ± 1.10 | 1.92 ± 0.35 |
| LS (Müller et al., 2019) | 92.51 ± 0.28 | 4.28 ± 2.10 | 2.33 ± 0.48 | 90.26 ± 0.85 | 5.48 ± 1.45 | 1.62 ± 0.38 | 90.84 ± 1.25 | 9.26 ± 2.25 | 4.84 ± 0.50 |
| BS (Brier, 1950) | 90.09 ± 0.85 | 6.06 ± 0.32 | 2.49 ± 0.17 | 89.50 ± 0.48 | 5.58 ± 0.25 | 1.68 ± 0.75 | 90.06 ± 1.05 | 8.45 ± 0.55 | 3.03 ± 0.20 |
| MMCE (Kumar et al., 2018) | 90.22 ± 0.75 | 10.20 ± 0.35 | 5.16 ± 0.22 | 90.98 ± 0.70 | 8.28 ± 0.50 | 3.00 ± 0.17 | 89.87 ± 1.15 | 7.49 ± 0.28 | 2.61 ± 0.20 |
| FLSD (Mukhoti et al., 2020) | 90.57 ± 0.55 | 8.26 ± 0.24 | 3.01 ± 0.16 | 88.65 ± 0.75 | 6.00 ± 0.40 | 1.70 ± 0.20 | 90.37 ± 1.00 | 7.66 ± 0.45 | 2.65 ± 0.24 |
| MbLS (Liu et al., 2021) | 93.00 ± 1.20 | 5.22 ± 1.60 | 1.58 ± 0.38 | 91.47 ± 0.90 | 9.60 ± 0.70 | 4.86 ± 0.30 | 89.82 ± 0.75 | 9.29 ± 0.60 | 4.80 ± 0.22 |
| DCA (Liang et al., 2020) | 92.99 ± 0.35 | 2.06 ± 0.18 | 0.81 ± 0.22 | 91.95 ± 0.60 | 3.62 ± 0.45 | 1.04 ± 0.33 | 92.84 ± 0.80 | 4.49 ± 0.70 | 1.52 ± 0.40 |
| NUCFL (DCA+COS) | 93.00 ± 0.40 | 1.45 ± 0.20 | 0.70 ± 0.12 | 92.07 ± 0.48 | 2.81 ± 0.50 | 1.00 ± 0.18 | 92.88 ± 0.70 | 4.43 ± 0.65 | 1.52 ± 0.30 |
| NUCFL (DCA+L-CKA) | 92.41 ± 0.30 | 1.32 ± 0.15 | 0.66 ± 0.14 | 92.12 ± 0.55 | 2.83 ± 0.25 | 1.06 ± 0.20 | 92.44 ± 0.85 | 4.34 ± 0.55 | 1.48 ± 0.28 |
| NUCFL (DCA+RBF-CKA) | 93.11 ± 0.51 | 1.40 ± 0.22 | 0.71 ± 0.10 | 91.96 ± 0.65 | 2.83 ± 0.28 | 1.05 ± 0.15 | 92.18 ± 0.75 | 4.26 ± 0.70 | 1.40 ± 0.35 |
| MDCA (Hebbalaguppe et al., 2022b) | 92.94 ± 0.43 | 2.53 ± 0.35 | 0.89 ± 0.25 | 91.31 ± 0.70 | 3.78 ± 0.50 | 1.08 ± 0.28 | 91.88 ± 0.90 | 4.39 ± 0.65 | 1.53 ± 0.30 |
| NUCFL (MDCA+COS) | 92.89 ± 0.38 | 2.00 ± 0.18 | 0.82 ± 0.15 | 91.28 ± 0.60 | 3.05 ± 0.40 | 1.03 ± 0.22 | 92.04 ± 0.80 | 4.22 ± 0.60 | 1.50 ± 0.28 |
| NUCFL (MDCA+L-CKA) | 93.86 ± 0.33 | 1.95 ± 0.20 | 0.81 ± 0.12 | 92.09 ± 0.75 | 3.40 ± 0.30 | 1.05 ± 0.18 | 92.17 ± 0.85 | 4.13 ± 0.50 | 1.37 ± 0.25 |
| NUCFL (MDCA+RBF-CKA) | 93.35 ± 0.40 | 1.88 ± 0.25 | 0.79 ± 0.14 | 92.19 ± 0.55 | 3.57 ± 0.35 | 1.08 ± 0.17 | 91.68 ± 0.95 | 4.13 ± 0.60 | 1.40 ± 0.28 |

| Calibration Method | FedDyn (Acar et al., 2021) | | | FedNova (Wang et al., 2020b) | | |
|---|---|---|---|---|---|---|
| | Acc ↑ | ECE ↓ | SCE ↓ | Acc ↑ | ECE ↓ | SCE ↓ |
| Uncal. | 94.24 ± 0.90 | 4.28 ± 0.72 | 1.08 ± 0.09 | 94.30 ± 1.12 | 2.95 ± 0.35 | 0.89 ± 0.13 |
| Focal (Lin et al., 2017) | 95.03 ± 0.45 | 7.38 ± 0.65 | 2.61 ± 0.11 | 93.00 ± 1.15 | 9.32 ± 0.55 | 4.82 ± 0.13 |
| LS (Müller et al., 2019) | 95.16 ± 1.05 | 10.86 ± 1.65 | 5.20 ± 0.30 | 93.15 ± 1.18 | 9.00 ± 1.70 | 4.80 ± 0.95 |
| BS (Brier, 1950) | 95.26 ± 0.70 | 9.20 ± 0.47 | 4.83 ± 0.13 | 89.59 ± 0.95 | 4.23 ± 0.30 | 1.19 ± 0.16 |
| MMCE (Kumar et al., 2018) | 93.36 ± 0.60 | 5.95 ± 0.38 | 1.74 ± 0.14 | 93.88 ± 1.05 | 8.89 ± 0.55 | 4.77 ± 0.22 |
| FLSD (Mukhoti et al., 2020) | 93.37 ± 0.50 | 5.67 ± 0.43 | 1.70 ± 0.12 | 92.13 ± 0.85 | 10.97 ± 0.65 | 5.21 ± 0.27 |
| MbLS (Liu et al., 2021) | 91.52 ± 0.85 | 4.16 ± 0.50 | 1.06 ± 0.14 | 94.84 ± 1.30 | 4.21 ± 0.75 | 1.12 ± 0.25 |
| DCA (Liang et al., 2020) | 95.25 ± 0.50 | 3.18 ± 0.28 | 1.02 ± 0.15 | 93.84 ± 0.95 | 1.87 ± 0.38 | 0.79 ± 0.18 |
| NUCFL (DCA+COS) | 95.08 ± 0.65 | 2.57 ± 0.33 | 0.87 ± 0.22 | 94.85 ± 0.80 | 1.61 ± 0.28 | 0.75 ± 0.14 |
| NUCFL (DCA+L-CKA) | 94.30 ± 0.45 | 2.41 ± 0.25 | 0.86 ± 0.17 | 94.00 ± 0.90 | 1.40 ± 0.22 | 0.70 ± 0.12 |
| NUCFL (DCA+RBF-CKA) | 95.69 ± 0.55 | 2.50 ± 0.30 | 0.89 ± 0.14 | 93.37 ± 0.85 | 1.52 ± 0.25 | 0.73 ± 0.13 |
| MDCA (Hebbalaguppe et al., 2022b) | 95.08 ± 0.65 | 3.13 ± 0.38 | 1.02 ± 0.20 | 93.74 ± 1.00 | 2.77 ± 0.42 | 0.86 ± 0.24 |
| NUCFL (MDCA+COS) | 94.37 ± 0.55 | 2.85 ± 0.35 | 1.00 ± 0.18 | 95.23 ± 0.90 | 2.45 ± 0.30 | 0.82 ± 0.13 |
| NUCFL (MDCA+L-CKA) | 94.33 ± 0.45 | 2.84 ± 0.28 | 1.01 ± 0.15 | 94.36 ± 0.95 | 2.39 ± 0.25 | 0.80 ± 0.14 |
| NUCFL (MDCA+RBF-CKA) | 94.82 ± 0.60 | 2.51 ± 0.40 | 0.83 ± 0.22 | 93.68 ± 1.05 | 2.43 ± 0.28 | 0.80 ± 0.12 |

Table 9: Accuracy (%), calibration measures ECE (%), and SCE (%) of various federated optimization methods with different calibration methods under non-IID ($\alpha = 0.1$) scenario on the MNIST dataset. Underlined values indicate the best calibration across all methods.

| Calibration Method | FedAvg (McMahan et al., 2017) | | | FedProx (Sahu et al., 2018) | | | Scaffold (Karimireddy et al., 2019) | | |
|---|---|---|---|---|---|---|---|---|---|
| | Acc ↑ | ECE ↓ | SCE ↓ | Acc ↑ | ECE ↓ | SCE ↓ | Acc ↑ | ECE ↓ | SCE ↓ |
| Uncal. | 87.51 ± 0.45 | 6.22 ± 0.22 | 1.83 ± 0.15 | 87.02 ± 0.60 | 6.60 ± 0.30 | 2.22 ± 0.25 | 88.20 ± 0.80 | 10.96 ± 0.50 | 3.53 ± 0.35 |
| Focal (Lin et al., 2017) | 87.95 ± 1.55 | 10.43 ± 0.35 | 3.50 ± 0.20 | 84.22 ± 0.70 | 11.26 ± 0.95 | 3.73 ± 0.25 | 83.50 ± 0.85 | 8.37 ± 1.65 | 3.04 ± 0.88 |
| LS (Müller et al., 2019) | 86.48 ± 1.91 | 11.62 ± 1.15 | 3.91 ± 0.92 | 83.52 ± 0.69 | 7.49 ± 1.77 | 2.68 ± 0.88 | 84.89 ± 1.11 | 14.57 ± 1.85 | 4.64 ± 0.79 |
| BS (Brier, 1950) | 83.36 ± 0.85 | 5.65 ± 0.56 | 1.72 ± 0.35 | 87.96 ± 1.15 | 10.44 ± 1.61 | 3.64 ± 0.99 | 88.21 ± 1.55 | 19.55 ± 2.92 | 7.53 ± 1.05 |
| MMCE (Kumar et al., 2018) | 88.99 ± 1.65 | 12.18 ± 1.43 | 4.04 ± 0.77 | 86.32 ± 1.83 | 11.74 ± 0.96 | 3.75 ± 0.72 | 86.48 ± 1.29 | 14.33 ± 2.38 | 4.61 ± 0.67 |
| FLSD (Mukhoti et al., 2020) | 85.42 ± 0.77 | 9.28 ± 1.06 | 3.48 ± 0.45 | 85.52 ± 1.05 | 10.11 ± 1.85 | 3.60 ± 0.55 | 86.73 ± 1.35 | 11.03 ± 1.11 | 3.69 ± 0.78 |
| MbLS (Liu et al., 2021) | 68.96 ± 1.81 | 13.27 ± 1.55 | 4.51 ± 0.40 | 87.66 ± 1.19 | 11.59 ± 1.57 | 3.75 ± 0.89 | 89.73 ± 2.25 | 18.66 ± 2.79 | 7.05 ± 1.25 |
| DCA (Liang et al., 2020) | 88.48 ± 1.15 | 5.83 ± 0.80 | 1.75 ± 0.22 | 87.30 ± 0.99 | 5.68 ± 0.35 | 1.72 ± 0.28 | 88.69 ± 1.01 | 9.22 ± 0.72 | 3.44 ± 0.37 |
| NUCFL (DCA+COS) | 88.00 ± 0.71 | 5.38 ± 0.25 | 1.69 ± 0.28 | 88.00 ± 1.11 | 5.03 ± 0.22 | 1.41 ± 0.77 | 88.82 ± 1.33 | 8.69 ± 0.77 | 3.29 ± 0.37 |
| NUCFL (DCA+L-CKA) | 87.70 ± 1.14 | 5.20 ± 0.72 | 1.59 ± 0.62 | 87.33 ± 1.05 | 5.05 ± 0.17 | 1.41 ± 0.24 | 88.71 ± 1.13 | 8.55 ± 0.35 | 3.23 ± 0.20 |
| NUCFL (DCA+RBF-CKA) | 88.79 ± 1.03 | 5.20 ± 1.22 | 1.61 ± 0.87 | 88.14 ± 1.19 | 5.11 ± 0.20 | 1.47 ± 0.79 | 89.01 ± 1.03 | 8.52 ± 0.61 | 3.23 ± 0.61 |
| MDCA (Hebbalaguppe et al., 2022b) | 88.36 ± 1.60 | 5.41 ± 0.79 | 1.72 ± 0.25 | 88.13 ± 1.95 | 5.84 ± 0.81 | 1.77 ± 0.67 | 88.52 ± 1.15 | 9.51 ± 0.75 | 3.47 ± 0.30 |
| NUCFL (MDCA+COS) | 88.20 ± 1.21 | 5.30 ± 0.71 | 1.67 ± 0.29 | 88.29 ± 1.21 | 5.21 ± 0.68 | 1.60 ± 0.29 | 88.64 ± 1.09 | 8.67 ± 0.82 | 3.28 ± 0.29 |
| NUCFL (MDCA+L-CKA) | 87.33 ± 1.03 | 5.33 ± 0.64 | 1.68 ± 0.39 | 87.51 ± 1.40 | 5.11 ± 0.32 | 1.49 ± 0.09 | 88.60 ± 1.10 | 8.52 ± 0.35 | 3.22 ± 0.22 |
| NUCFL (MDCA+RBF-CKA) | 88.15 ± 0.96 | 5.27 ± 0.32 | 1.66 ± 0.15 | 88.00 ± 1.03 | 5.38 ± 0.20 | 1.70 ± 0.12 | 88.43 ± 1.00 | 8.69 ± 0.61 | 3.27 ± 0.18 |

| Calibration Method | FedDyn (Acar et al., 2021) | | | FedNova (Wang et al., 2020b) | | |
|---|---|---|---|---|---|---|
| | Acc ↑ | ECE ↓ | SCE ↓ | Acc ↑ | ECE ↓ | SCE ↓ |
| Uncal. | 88.51 ± 0.70 | 7.52 ± 0.40 | 2.67 ± 0.28 | 89.11 ± 0.90 | 9.68 ± 0.18 | 3.49 ± 0.22 |
| Focal (Lin et al., 2017) | 87.49 ± 1.65 | 9.24 ± 0.77 | 3.46 ± 0.93 | 87.23 ± 0.81 | 10.74 ± 0.50 | 3.50 ± 0.27 |
| LS (Müller et al., 2019) | 86.95 ± 1.85 | 7.20 ± 0.38 | 2.65 ± 0.32 | 85.84 ± 1.20 | 8.31 ± 0.77 | 3.27 ± 0.92 |
| BS (Brier, 1950) | 84.05 ± 0.95 | 6.85 ± 0.55 | 2.35 ± 0.33 | 87.29 ± 1.70 | 9.79 ± 1.28 | 3.49 ± 0.44 |
| MMCE (Kumar et al., 2018) | 87.49 ± 1.74 | 9.48 ± 1.22 | 3.50 ± 0.92 | 88.48 ± 1.10 | 11.90 ± 0.65 | 3.84 ± 0.69 |
| FLSD (Mukhoti et al., 2020) | 85.86 ± 1.10 | 6.97 ± 0.44 | 2.37 ± 0.31 | 86.00 ± 1.39 | 10.39 ± 2.80 | 3.46 ± 1.50 |
| MbLS (Liu et al., 2021) | 88.40 ± 0.95 | 9.05 ± 1.40 | 3.43 ± 0.92 | 87.47 ± 1.43 | 10.62 ± 1.55 | 3.51 ± 0.82 |
| DCA (Liang et al., 2020) | 88.52 ± 0.83 | 6.95 ± 0.39 | 2.37 ± 0.25 | 88.97 ± 1.00 | 8.09 ± 0.39 | 3.09 ± 0.13 |
| NUCFL (DCA+COS) | 89.00 ± 0.99 | 6.48 ± 0.28 | 2.30 ± 0.18 | 89.29 ± 1.11 | 7.66 ± 0.20 | 2.66 ± 0.73 |
| NUCFL (DCA+L-CKA) | 88.71 ± 1.23 | 6.39 ± 1.03 | 2.27 ± 0.82 | 89.50 ± 0.61 | 7.79 ± 1.02 | 2.72 ± 0.68 |
| NUCFL (DCA+RBF-CKA) | 88.90 ± 0.99 | 6.23 ± 0.73 | 2.25 ± 0.48 | 88.92 ± 0.61 | 6.97 ± 0.72 | 2.45 ± 0.17 |
| MDCA (Hebbalaguppe et al., 2022b) | 89.03 ± 1.25 | 6.26 ± 0.77 | 2.21 ± 0.49 | 88.60 ± 1.23 | 8.12 ± 1.05 | 3.09 ± 0.49 |
| NUCFL (MDCA+COS) | 88.90 ± 1.29 | 5.91 ± 0.68 | 1.79 ± 0.39 | 89.76 ± 1.29 | 7.61 ± 0.68 | 2.67 ± 0.23 |
| NUCFL (MDCA+L-CKA) | 89.09 ± 1.24 | 5.88 ± 0.49 | 1.79 ± 0.18 | 90.07 ± 1.15 | 7.55 ± 0.66 | 2.64 ± 0.37 |
| NUCFL (MDCA+RBF-CKA) | 88.76 ± 0.67 | 5.88 ± 1.28 | 1.80 ± 0.44 | 89.73 ± 1.33 | 7.41 ± 0.76 | 2.60 ± 0.09 |

Table 10: Accuracy (%), calibration measures ECE (%), and SCE (%) of various federated optimization methods with different calibration methods under non-IID ($\alpha = 0.05$) scenario on the MNIST dataset. Underlined values indicate the best calibration across all methods.

| Calibration Method | FedAvg (McMahan et al., 2017) | | | FedProx (Sahu et al., 2018) | | | Scaffold (Karimireddy et al., 2019) | | |
|---|---|---|---|---|---|---|---|---|---|
| | Acc ↑ | ECE ↓ | SCE ↓ | Acc ↑ | ECE ↓ | SCE ↓ | Acc ↑ | ECE ↓ | SCE ↓ |
| Uncal. | 92.48 ± 0.23 | 4.10 ± 0.15 | 1.66 ± 0.08 | 93.07 ± 0.27 | 4.49 ± 0.16 | 1.77 ± 0.07 | 91.77 ± 0.32 | 4.28 ± 0.19 | 1.66 ± 0.10 |
| Focal (Lin et al., 2017) | 91.06 ± 0.28 | 4.73 ± 0.17 | 1.98 ± 0.11 | 91.51 ± 0.44 | 5.18 ± 0.14 | 2.55 ± 0.07 | 90.85 ± 0.24 | 5.64 ± 0.04 | 2.73 ± 0.05 |
| LS (Müller et al., 2019) | 90.39 ± 0.32 | 3.98 ± 0.17 | 1.65 ± 0.19 | 92.96 ± 0.26 | 6.33 ± 0.22 | 3.16 ± 0.17 | 90.61 ± 0.66 | 4.22 ± 0.61 | 1.66 ± 0.29 |
| BS (Brier, 1950) | 92.11 ± 0.31 | 5.11 ± 0.30 | 2.55 ± 0.29 | 93.01 ± 0.32 | 6.28 ± 0.32 | 3.17 ± 0.19 | 90.00 ± 0.19 | 6.08 ± 0.44 | 3.00 ± 0.29 |
| MMCE (Kumar et al., 2018) | 92.15 ± 0.30 | 4.98 ± 0.29 | 2.52 ± 0.17 | 91.16 ± 0.38 | 3.89 ± 0.29 | 1.65 ± 0.11 | 92.75 ± 0.27 | 6.90 ± 0.19 | 3.52 ± 0.11 |
| FLSD (Mukhoti et al., 2020) | 91.47 ± 0.33 | 3.26 ± 0.29 | 1.44 ± 0.05 | 92.23 ± 0.28 | 5.93 ± 0.19 | 2.78 ± 0.10 | 90.41 ± 0.30 | 6.31 ± 0.18 | 3.17 ± 0.11 |
| MbLS (Liu et al., 2021) | 92.11 ± 0.25 | 4.61 ± 0.21 | 1.70 ± 0.19 | 92.00 ± 0.29 | 4.98 ± 0.32 | 2.52 ± 0.19 | 90.55 ± 0.44 | 4.30 ± 0.32 | 1.68 ± 0.08 |
| DCA (Liang et al., 2020) | 92.54 ± 0.23 | 4.00 ± 0.14 | 1.63 ± 0.09 | 93.40 ± 0.25 | 4.28 ± 0.15 | 1.65 ± 0.11 | 91.45 ± 0.24 | 4.22 ± 0.23 | 1.64 ± 0.10 |
| NUCFL (DCA+COS) | 92.67 ± 0.21 | 3.55 ± 0.11 | 1.51 ± 0.06 | 93.16 ± 0.22 | 3.81 ± 0.13 | 1.60 ± 0.09 | 92.10 ± 0.24 | **4.11** ± 0.22 | **1.60** ± 0.14 |
| NUCFL (DCA+L-CKA) | 92.39 ± 0.22 | **3.49** ± 0.20 | **1.46** ± 0.11 | 93.06 ± 0.23 | **3.70** ± 0.14 | **1.55** ± 0.11 | 92.73 ± 0.24 | 4.16 ± 0.15 | 1.61 ± 0.13 |
| NUCFL (DCA+RBF-CKA) | 92.58 ± 0.11 | 3.50 ± 0.17 | 1.49 ± 0.11 | 93.11 ± 0.36 | **3.70** ± 0.25 | 1.56 ± 0.18 | 92.81 ± 0.49 | 4.12 ± 0.25 | 1.61 ± 0.14 |
| MDCA (Hebbalaguppe et al., 2022b) | 93.05 ± 0.17 | 3.96 ± 0.12 | 1.63 ± 0.10 | 93.36 ± 0.24 | 4.30 ± 0.13 | 1.66 ± 0.10 | 92.03 ± 0.23 | 4.20 ± 0.16 | 1.63 ± 0.11 |
| NUCFL (MDCA+COS) | 92.59 ± 0.28 | 3.78 ± 0.30 | 1.60 ± 0.15 | 93.14 ± 0.24 | 4.04 ± 0.12 | 1.63 ± 0.06 | 92.02 ± 0.19 | 4.12 ± 0.22 | **1.60** ± 0.14 |
| NUCFL (MDCA+L-CKA) | 92.57 ± 0.21 | 3.70 ± 0.25 | 1.58 ± 0.13 | 93.12 ± 0.55 | **3.87** ± 0.41 | 1.60 ± 0.21 | 92.13 ± 0.24 | **4.10** ± 0.20 | 1.61 ± 0.13 |
| NUCFL (MDCA+RBF-CKA) | 92.65 ± 0.11 | **3.61** ± 0.12 | **1.53** ± 0.09 | 93.21 ± 0.17 | 4.12 ± 0.25 | 1.61 ± 0.11 | 92.14 ± 0.31 | **4.10** ± 0.14 | **1.60** ± 0.08 |

| Calibration Method | FedDyn (Acar et al., 2021) | | | FedNova (Wang et al., 2020b) | | |
|---|---|---|---|---|---|---|
| | Acc ↑ | ECE ↓ | SCE ↓ | Acc ↑ | ECE ↓ | SCE ↓ |
| Uncal. | 92.53 ± 0.25 | 5.13 ± 0.18 | 2.55 ± 0.09 | 92.91 ± 0.31 | 4.75 ± 0.17 | 2.38 ± 0.08 |
| Focal (Lin et al., 2017) | 91.63 ± 0.27 | 6.08 ± 0.24 | 3.03 ± 0.11 | 91.33 ± 0.42 | 4.75 ± 0.27 | 2.39 ± 0.19 |
| LS (Müller et al., 2019) | 91.84 ± 0.28 | 6.88 ± 0.17 | 3.48 ± 0.19 | 91.13 ± 0.37 | 5.40 ± 0.16 | 2.60 ± 0.11 |
| BS (Brier, 1950) | 91.40 ± 0.61 | 5.23 ± 0.21 | 2.57 ± 0.17 | 91.44 ± 0.71 | 4.81 ± 0.32 | 2.40 ± 0.19 |
| MMCE (Kumar et al., 2018) | 93.30 ± 0.49 | 7.73 ± 0.42 | 3.85 ± 0.14 | 92.44 ± 0.61 | 5.96 ± 0.68 | 3.03 ± 0.37 |
| FLSD (Mukhoti et al., 2020) | 92.40 ± 0.27 | 5.03 ± 0.16 | 2.32 ± 0.08 | 91.95 ± 0.40 | 4.94 ± 0.33 | 2.44 ± 0.19 |
| MbLS (Liu et al., 2021) | 91.60 ± 0.30 | 5.18 ± 0.32 | 2.56 ± 0.17 | 92.00 ± 0.55 | 4.88 ± 0.42 | 2.41 ± 0.17 |
| DCA (Liang et al., 2020) | 93.00 ± 0.27 | 4.60 ± 0.16 | 2.30 ± 0.13 | 93.06 ± 0.26 | 4.25 ± 0.17 | 1.68 ± 0.13 |
| NUCFL (DCA+COS) | 93.08 ± 0.35 | 4.38 ± 0.29 | 1.67 ± 0.17 | 93.26 ± 0.23 | 4.14 ± 0.30 | 1.59 ± 0.17 |
| NUCFL (DCA+L-CKA) | 92.87 ± 0.26 | **4.22** ± 0.27 | **1.64** ± 0.15 | 93.06 ± 0.24 | **4.03** ± 0.17 | **1.54** ± 0.11 |
| NUCFL (DCA+RBF-CKA) | 93.09 ± 0.27 | 4.25 ± 0.19 | 1.65 ± 0.11 | 92.99 ± 0.25 | 4.15 ± 0.17 | 1.58 ± 0.10 |
| MDCA (Hebbalaguppe et al., 2022b) | 92.70 ± 0.26 | 4.77 ± 0.14 | 2.38 ± 0.08 | 92.89 ± 0.25 | 4.37 ± 0.13 | 1.66 ± 0.09 |
| NUCFL (MDCA+COS) | 93.25 ± 0.19 | 4.41 ± 0.14 | 1.64 ± 0.09 | 93.06 ± 0.22 | 4.21 ± 0.14 | 1.64 ± 0.10 |
| NUCFL (MDCA+L-CKA) | 92.87 ± 0.26 | **4.29** ± 0.24 | **1.65** ± 0.11 | 92.85 ± 0.07 | **4.17** ± 0.09 | **1.62** ± 0.09 |
| NUCFL (MDCA+RBF-CKA) | 93.00 ± 0.15 | 4.36 ± 0.22 | 1.66 ± 0.10 | 92.92 ± 0.19 | 4.22 ± 0.13 | 1.64 ± 0.07 |

Table 11: Accuracy (%), calibration measures ECE (%), and SCE (%) of various FL algorithms with different calibration methods under IID scenario on the FEMNIST dataset. Underlined values indicate the best calibration across all methods.

| Calibration Method | FedAvg (McMahan et al., 2017) | | | FedProx (Sahu et al., 2018) | | | Scaffold (Karimireddy et al., 2019) | | |
|---|---|---|---|---|---|---|---|---|---|
| | Acc ↑ | ECE ↓ | SCE ↓ | Acc ↑ | ECE ↓ | SCE ↓ | Acc ↑ | ECE ↓ | SCE ↓ |
| Uncal. | 90.95 ± 0.64 | 4.17 ± 1.04 | 1.66 ± 0.72 | 91.45 ± 1.17 | 4.61 ± 1.03 | 1.95 ± 0.99 | 91.28 ± 1.29 | 5.02 ± 1.19 | 2.51 ± 0.88 |
| Focal (Lin et al., 2017) | 90.63 ± 0.98 | 4.77 ± 1.30 | 1.96 ± 0.75 | 90.00 ± 1.29 | 5.39 ± 1.43 | 2.77 ± 1.05 | 91.13 ± 1.66 | 5.38 ± 1.27 | 2.72 ± 1.04 |
| LS (Müller et al., 2019) | 91.07 ± 1.23 | 3.75 ± 1.72 | 1.62 ± 1.02 | 90.37 ± 1.38 | 4.52 ± 1.14 | 1.90 ± 1.04 | 91.33 ± 1.75 | 5.18 ± 1.08 | 2.64 ± 0.99 |
| BS (Brier, 1950) | 91.48 ± 0.88 | 5.19 ± 1.24 | 2.65 ± 0.83 | 90.98 ± 1.00 | 5.42 ± 0.95 | 2.80 ± 0.61 | 88.72 ± 1.11 | 4.19 ± 1.43 | 1.81 ± 1.06 |
| MMCE (Kumar et al., 2018) | 90.22 ± 1.11 | 4.85 ± 1.23 | 2.30 ± 1.05 | 90.12 ± 1.32 | 4.01 ± 2.04 | 1.79 ± 1.38 | 91.44 ± 1.33 | 5.07 ± 1.68 | 2.55 ± 1.21 |
| FLSD (Mukhoti et al., 2020) | 90.02 ± 1.37 | 4.93 ± 1.28 | 2.95 ± 0.74 | 90.39 ± 1.25 | 4.99 ± 1.38 | 2.04 ± 1.06 | 92.88 ± 0.84 | 5.19 ± 1.14 | 2.58 ± 0.92 |
| MbLS (Liu et al., 2021) | 90.62 ± 1.59 | 4.27 ± 1.03 | 1.99 ± 0.80 | 91.49 ± 1.56 | 4.79 ± 1.50 | 1.94 ± 0.84 | 92.87 ± 1.64 | 5.39 ± 1.35 | 2.60 ± 0.91 |
| DCA (Liang et al., 2020) | 91.84 ± 1.06 | 3.61 ± 0.92 | 1.52 ± 0.53 | 92.03 ± 1.04 | 4.25 ± 0.58 | 1.61 ± 0.69 | 92.04 ± 1.30 | 4.44 ± 1.08 | 1.82 ± 0.88 |
| NUCFL (DCA+COS) | 91.77 ± 1.02 | 3.52 ± 0.79 | 1.49 ± 0.45 | 91.95 ± 1.28 | 3.61 ± 1.04 | 1.35 ± 0.96 | 92.42 ± 1.02 | 4.39 ± 1.37 | 1.77 ± 1.08 |
| NUCFL (DCA+L-CKA) | 91.63 ± 0.95 | 3.52 ± 0.88 | 1.47 ± 0.61 | 92.10 ± 0.96 | **3.60** ± 0.88 | **1.30** ± 0.85 | 92.19 ± 0.95 | **4.09** ± 0.83 | **1.60** ± 0.65 |
| NUCFL (DCA+RBF-CKA) | 91.74 ± 1.15 | **3.49** ± 1.00 | **1.40** ± 0.78 | 91.99 ± 1.24 | 3.77 ± 1.04 | 1.35 ± 1.03 | 91.74 ± 1.33 | 4.15 ± 1.29 | 1.62 ± 0.94 |
| MDCA (Hebbalaguppe et al., 2022b) | 91.64 ± 0.85 | 3.75 ± 1.29 | 1.53 ± 0.95 | 92.17 ± 1.33 | 4.42 ± 1.39 | 2.05 ± 1.06 | 92.95 ± 1.38 | 4.61 ± 1.29 | 1.90 ± 1.07 |
| NUCFL (MDCA+COS) | 91.29 ± 0.95 | 3.61 ± 1.20 | 1.44 ± 1.04 | 91.95 ± 1.00 | 3.95 ± 1.29 | 1.77 ± 0.93 | 93.07 ± 1.19 | **4.14** ± 1.22 | **1.62** ± 0.57 |
| NUCFL (MDCA+L-CKA) | 91.97 ± 1.06 | **3.28** ± 1.11 | **1.28** ± 0.93 | 92.20 ± 1.20 | **3.88** ± 1.04 | **1.51** ± 1.22 | 92.77 ± 1.33 | 4.20 ± 1.29 | 1.80 ± 1.08 |
| NUCFL (MDCA+RBF-CKA) | 91.35 ± 1.13 | 3.56 ± 0.99 | 1.40 ± 0.61 | 92.21 ± 0.95 | 4.00 ± 1.33 | 1.77 ± 1.16 | 93.04 ± 1.52 | 4.19 ± 1.63 | 1.79 ± 1.05 |

| Calibration Method | FedDyn (Acar et al., 2021) | | | FedNova (Wang et al., 2020b) | | |
|---|---|---|---|---|---|---|
| | Acc ↑ | ECE ↓ | SCE ↓ | Acc ↑ | ECE ↓ | SCE ↓ |
| Uncal. | 92.75 ± 1.56 | 5.24 ± 1.22 | 2.60 ± 1.03 | 92.22 ± 1.00 | 4.92 ± 0.95 | 2.44 ± 0.61 |
| Focal (Lin et al., 2017) | 91.15 ± 1.07 | 5.84 ± 1.05 | 2.91 ± 0.84 | 91.13 ± 1.20 | 5.39 ± 1.49 | 2.63 ± 0.92 |
| LS (Müller et al., 2019) | 93.08 ± 1.90 | 5.19 ± 1.65 | 2.63 ± 1.01 | 92.04 ± 1.11 | 4.95 ± 1.32 | 2.46 ± 0.85 |
| BS (Brier, 1950) | 91.39 ± 1.24 | 5.37 ± 0.98 | 2.62 ± 0.86 | 90.38 ± 1.33 | 4.65 ± 1.52 | 1.92 ± 1.04 |
| MMCE (Kumar et al., 2018) | 92.11 ± 1.43 | 4.93 ± 1.29 | 2.49 ± 1.00 | 91.35 ± 1.59 | 5.08 ± 1.29 | 2.49 ± 0.96 |
| FLSD (Mukhoti et al., 2020) | 92.06 ± 1.69 | 5.44 ± 1.77 | 2.75 ± 1.24 | 90.39 ± 1.71 | 4.61 ± 1.33 | 1.92 ± 1.06 |
| MbLS (Liu et al., 2021) | 92.06 ± 1.69 | 5.44 ± 1.77 | 2.75 ± 1.24 | 90.39 ± 1.71 | 4.61 ± 1.33 | 1.92 ± 1.06 |
| DCA (Liang et al., 2020) | 93.08 ± 1.44 | 4.61 ± 1.58 | 1.92 ± 1.06 | 92.37 ± 1.35 | 4.31 ± 1.24 | 1.71 ± 0.91 |
| NUCFL (DCA+COS) | 93.22 ± 1.11 | **4.20** ± 1.25 | **1.65** ± 0.93 | 92.25 ± 1.04 | 4.13 ± 0.95 | 1.68 ± 0.74 |
| NUCFL (DCA+L-CKA) | 93.45 ± 0.96 | 4.22 ± 1.14 | **1.54** ± 0.69 | 92.33 ± 1.22 | **4.02** ± 1.09 | **1.54** ± 0.83 |
| NUCFL (DCA+RBF-CKA) | 92.95 ± 1.61 | 4.34 ± 1.33 | 1.75 ± 1.10 | 92.41 ± 1.15 | 4.11 ± 1.03 | 1.68 ± 0.65 |
| MDCA (Hebbalaguppe et al., 2022b) | 93.19 ± 1.07 | 4.71 ± 0.94 | 1.93 ± 0.75 | 92.05 ± 1.28 | 4.62 ± 1.14 | 1.82 ± 0.85 |
| NUCFL (MDCA+COS) | 93.11 ± 1.64 | **4.31** ± 1.22 | **1.75** ± 0.99 | 91.99 ± 1.22 | 4.28 ± 0.98 | 1.67 ± 0.59 |
| NUCFL (MDCA+L-CKA) | 93.11 ± 1.64 | **4.31** ± 1.22 | **1.75** ± 0.99 | 91.99 ± 1.22 | 4.28 ± 0.98 | 1.67 ± 0.59 |
| NUCFL (MDCA+RBF-CKA) | 93.04 ± 1.23 | 4.40 ± 1.35 | 1.80 ± 1.05 | 92.39 ± 0.95 | **4.17** ± 0.93 | **1.67** ± 0.85 |

Table 12: Accuracy (%), calibration measures ECE (%), and SCE (%) of various FL algorithms with different calibration methods under non-IID ($\alpha = 0.5$) scenario on the FEMNIST dataset. Values in boldface represent the best calibration provided by our method for the auxiliary calibration method, and underlined values indicate the best calibration across all methods.

| Calibration Method | FedAvg (McMahan et al., 2017) | | | FedProx (Sahu et al., 2018) | | | Scaffold (Karimireddy et al., 2019) | | |
|---|---|---|---|---|---|---|---|---|---|
| | Acc ↑ | ECE ↓ | SCE ↓ | Acc ↑ | ECE ↓ | SCE ↓ | Acc ↑ | ECE ↓ | SCE ↓ |
| Uncal. | 88.91 ± 0.53 | 5.58 ± 0.22 | 2.83 ± 0.35 | 88.67 ± 0.47 | 5.73 ± 0.16 | 2.79 ± 0.10 | 90.07 ± 0.62 | 6.67 ± 0.24 | 3.40 ± 0.17 |
| Focal (Lin et al., 2017) | 89.71 ± 0.48 | 6.89 ± 0.33 | 3.51 ± 0.22 | 89.06 ± 0.52 | 6.13 ± 0.17 | 3.06 ± 0.09 | 89.25 ± 0.57 | 7.71 ± 0.30 | 3.84 ± 0.17 |
| LS (Müller et al., 2019) | 88.89 ± 0.45 | 6.14 ± 0.17 | 3.06 ± 0.15 | 87.51 ± 0.49 | 6.49 ± 0.19 | 3.33 ± 0.26 | 90.11 ± 0.55 | 8.37 ± 0.66 | 3.41 ± 0.30 |
| BS (Brier, 1950) | 89.99 ± 0.52 | 8.26 ± 0.23 | 3.39 ± 0.17 | 90.51 ± 0.56 | 5.73 ± 0.58 | 2.77 ± 0.24 | 87.53 ± 0.47 | 6.23 ± 0.34 | 3.10 ± 0.22 |
| MMCE (Kumar et al., 2018) | 89.65 ± 0.92 | 8.13 ± 0.78 | 3.35 ± 0.44 | 88.66 ± 0.92 | 8.06 ± 0.40 | 3.30 ± 0.21 | 90.25 ± 1.08 | 7.15 ± 0.82 | 3.05 ± 0.39 |
| FLSD (Mukhoti et al., 2020) | 86.97 ± 0.67 | 5.41 ± 0.33 | 2.61 ± 0.23 | 89.73 ± 0.71 | 9.09 ± 0.32 | 3.49 ± 0.19 | 90.91 ± 0.71 | 8.46 ± 0.30 | 3.36 ± 0.22 |
| MbLS (Liu et al., 2021) | 86.93 ± 1.09 | 7.39 ± 0.77 | 3.59 ± 0.39 | 89.43 ± 0.98 | 7.62 ± 0.41 | 3.82 ± 0.32 | 89.05 ± 0.77 | 8.24 ± 0.21 | 3.39 ± 0.20 |
| DCA (Liang et al., 2020) | 89.04 ± 0.57 | 4.79 ± 0.22 | 1.98 ± 0.14 | 89.90 ± 0.61 | 4.91 ± 0.23 | 2.51 ± 0.15 | 90.16 ± 0.64 | 6.08 ± 0.24 | 3.00 ± 0.17 |
| NUCFL (DCA+COS) | 90.01 ± 0.72 | 4.50 ± 0.34 | 1.77 ± 0.19 | 88.86 ± 0.88 | 4.62 ± 0.39 | 1.72 ± 0.25 | 90.80 ± 0.71 | 5.86 ± 0.41 | 2.69 ± 0.29 |
| NUCFL (DCA+L-CKA) | 89.89 ± 0.53 | 4.49 ± 0.19 | 1.77 ± 0.12 | 88.99 ± 0.57 | **4.56** ± 0.20 | **1.69** ± 0.11 | 90.23 ± 0.60 | **5.61** ± 0.39 | **2.66** ± 0.20 |
| NUCFL (DCA+RBF-CKA) | 89.07 ± 0.61 | **4.38** ± 0.25 | **1.71** ± 0.14 | 89.61 ± 0.58 | 4.77 ± 0.22 | 1.88 ± 0.12 | 90.31 ± 0.61 | 5.70 ± 0.23 | 2.68 ± 0.14 |
| MDCA (Hebbalaguppe et al., 2022b) | 89.18 ± 0.37 | 4.88 ± 0.22 | 2.34 ± 0.13 | 88.88 ± 0.58 | 4.87 ± 0.24 | 2.50 ± 0.15 | 91.03 ± 0.60 | 5.95 ± 0.22 | 2.98 ± 0.16 |
| NUCFL (MDCA+COS) | 89.03 ± 1.00 | 4.44 ± 0.39 | 1.78 ± 0.28 | 88.74 ± 0.91 | 4.77 ± 0.67 | 1.90 ± 0.32 | 90.52 ± 0.55 | 5.79 ± 0.33 | 2.67 ± 0.18 |
| NUCFL (MDCA+L-CKA) | 89.11 ± 0.78 | **4.39** ± 0.61 | **1.73** ± 0.30 | 88.96 ± 0.51 | 4.61 ± 0.22 | 1.72 ± 0.23 | 90.47 ± 0.54 | **5.67** ± 0.23 | 2.66 ± 0.14 |
| NUCFL (MDCA+RBF-CKA) | 88.99 ± 0.48 | 4.87 ± 0.21 | 2.35 ± 0.15 | 89.91 ± 0.52 | 4.58 ± 0.22 | **1.69** ± 0.13 | 90.84 ± 0.53 | **5.67** ± 0.24 | **2.65** ± 0.14 |

| Calibration Method | FedDyn (Acar et al., 2021) | | | FedNova (Wang et al., 2020b) | | |
|---|---|---|---|---|---|---|
| | Acc ↑ | ECE ↓ | SCE ↓ | Acc ↑ | ECE ↓ | SCE ↓ |
| Uncal. | 89.03 ± 0.58 | 6.02 ± 0.23 | 3.00 ± 0.17 | 89.56 ± 0.51 | 5.88 ± 0.22 | 2.76 ± 0.19 |
| Focal (Lin et al., 2017) | 89.13 ± 0.30 | 8.48 ± 0.22 | 3.45 ± 0.20 | 89.42 ± 0.49 | 6.12 ± 0.30 | 3.07 ± 0.21 |
| LS (Müller et al., 2019) | 88.34 ± 0.53 | 9.03 ± 0.44 | 3.50 ± 0.23 | 88.63 ± 0.50 | 7.57 ± 0.25 | 3.20 ± 0.10 |
| BS (Brier, 1950) | 88.90 ± 0.71 | 7.38 ± 0.33 | 3.59 ± 0.14 | 88.94 ± 0.48 | 7.43 ± 0.25 | 3.61 ± 0.14 |
| MMCE (Kumar et al., 2018) | 88.80 ± 1.57 | 7.90 ± 1.01 | 3.33 ± 0.49 | 88.94 ± 1.05 | 7.11 ± 0.77 | 3.04 ± 0.39 |
| FLSD (Mukhoti et al., 2020) | 88.90 ± 0.57 | 7.18 ± 0.24 | 3.05 ± 0.13 | 87.45 ± 1.04 | 6.09 ± 0.88 | 2.99 ± 0.39 |
| MbLS (Liu et al., 2021) | 88.10 ± 0.80 | 7.33 ± 0.37 | 3.59 ± 0.11 | 86.74 ± 0.91 | 5.14 ± 0.41 | 2.60 ± 0.29 |
| DCA (Liang et al., 2020) | 90.20 ± 0.66 | 5.72 ± 0.25 | 2.77 ± 0.16 | 90.11 ± 0.59 | 5.00 ± 0.22 | 2.45 ± 0.14 |
| NUCFL (DCA+COS) | 90.10 ± 0.56 | 5.33 ± 0.22 | 2.58 ± 0.28 | 90.96 ± 0.54 | 4.79 ± 0.40 | 2.00 ± 0.39 |
| NUCFL (DCA+L-CKA) | 89.37 ± 0.77 | **5.17** ± 0.23 | **2.55** ± 0.14 | 90.57 ± 1.39 | **4.62** ± 0.89 | **1.97** ± 0.30 |
| NUCFL (DCA+RBF-CKA) | 89.59 ± 0.63 | 5.40 ± 0.39 | 2.61 ± 0.28 | 90.19 ± 1.01 | 4.80 ± 0.72 | 1.99 ± 0.31 |
| MDCA (Hebbalaguppe et al., 2022b) | 90.10 ± 0.62 | 5.81 ± 0.25 | 2.79 ± 0.14 | 90.03 ± 0.57 | 5.27 ± 0.23 | 2.50 ± 0.12 |
| NUCFL (MDCA+COS) | 90.03 ± 0.58 | **5.45** ± 0.49 | **2.63** ± 0.20 | 90.36 ± 0.71 | 5.15 ± 0.69 | 2.50 ± 0.30 |
| NUCFL (MDCA+L-CKA) | 90.57 ± 1.56 | 5.66 ± 1.24 | 2.69 ± 0.70 | 89.62 ± 0.33 | 4.93 ± 0.22 | 2.05 ± 0.19 |
| NUCFL (MDCA+RBF-CKA) | 89.30 ± 0.71 | 5.73 ± 0.38 | 2.70 ± 0.29 | 89.66 ± 0.53 | **4.69** ± 0.31 | **1.99** ± 0.09 |

Table 13: Accuracy (%), calibration measures ECE (%), and SCE (%) of various FL algorithms with different calibration methods under non-IID ($\alpha = 0.1$) scenario on the FEMNIST dataset. Underlined values indicate the best calibration across all methods.

| Calibration Method | FedAvg (McMahan et al., 2017) | | | FedProx (Sahu et al., 2018) | | | Scaffold (Karimireddy et al., 2019) | | |
|---|---|---|---|---|---|---|---|---|---|
| | Acc ↑ | ECE ↓ | SCE ↓ | Acc ↑ | ECE ↓ | SCE ↓ | Acc ↑ | ECE ↓ | SCE ↓ |
| Uncal. | 86.98 ± 0.92 | 8.78 ± 0.28 | 3.49 ± 0.18 | 87.57 ± 1.01 | 7.03 ± 0.32 | 3.68 ± 0.17 | 87.17 ± 0.95 | 9.73 ± 0.35 | 3.70 ± 0.19 |
| Focal (Lin et al., 2017) | 84.71 ± 1.10 | 10.09 ± 0.77 | 3.55 ± 0.28 | 86.06 ± 1.26 | 9.33 ± 0.77 | 3.42 ± 0.69 | 87.25 ± 1.00 | 11.99 ± 0.36 | 4.35 ± 0.25 |
| LS (Müller et al., 2019) | 84.89 ± 0.91 | 8.34 ± 0.39 | 3.35 ± 0.22 | 87.01 ± 0.97 | 8.69 ± 0.33 | 3.49 ± 0.29 | 86.91 ± 0.92 | 9.67 ± 0.33 | 3.71 ± 0.11 |
| BS (Brier, 1950) | 85.99 ± 0.95 | 9.46 ± 0.34 | 3.44 ± 0.20 | 86.51 ± 1.03 | 8.93 ± 0.88 | 3.44 ± 0.37 | 86.53 ± 0.89 | 9.43 ± 0.27 | 3.69 ± 0.16 |
| MMCE (Kumar et al., 2018) | 86.65 ± 1.01 | 9.33 ± 0.47 | 3.43 ± 0.39 | 85.66 ± 0.88 | 6.26 ± 0.57 | 3.08 ± 0.19 | 87.25 ± 0.94 | 13.35 ± 0.32 | 4.52 ± 0.19 |
| FLSD (Mukhoti et al., 2020) | 85.97 ± 0.88 | 9.61 ± 0.35 | 3.47 ± 0.23 | 86.73 ± 0.91 | 9.29 ± 0.29 | 3.42 ± 0.17 | 87.91 ± 1.79 | 14.66 ± 1.46 | 5.07 ± 0.72 |
| MbLS (Liu et al., 2021) | 85.93 ± 0.77 | 9.59 ± 0.66 | 3.51 ± 0.28 | 86.43 ± 1.22 | 8.82 ± 0.61 | 3.47 ± 0.32 | 86.05 ± 1.33 | 10.44 ± 0.72 | 3.81 ± 0.38 |
| DCA (Liang et al., 2020) | 87.14 ± 1.04 | 8.39 ± 0.24 | 3.41 ± 0.11 | 87.40 ± 1.22 | 6.13 ± 0.49 | 3.06 ± 0.31 | 88.05 ± 1.12 | 9.08 ± 0.44 | 3.52 ± 0.31 |
| NUCFL (DCA+COS) | 87.07 ± 1.02 | 8.02 ± 0.26 | 3.30 ± 0.17 | 88.16 ± 1.25 | 5.96 ± 0.33 | 3.01 ± 0.21 | 87.80 ± 1.07 | 8.79 ± 0.61 | 3.49 ± 0.44 |
| NUCFL (DCA+L-CKA) | 86.89 ± 1.24 | 7.84 ± 0.71 | 3.27 ± 0.33 | 87.56 ± 1.04 | **5.06** ± 0.29 | 2.54 ± 0.17 | 87.23 ± 1.34 | 8.51 ± 0.72 | **3.45** ± 0.31 |
| NUCFL (DCA+RBF-CKA) | 86.93 ± 1.21 | **7.80** ± 0.49 | **3.25** ± 0.22 | 88.11 ± 1.21 | 5.09 ± 0.46 | **2.53** ± 0.25 | 88.31 ± 1.03 | **8.47** ± 0.66 | 3.46 ± 0.29 |
| MDCA (Hebbalaguppe et al., 2022b) | 87.07 ± 1.12 | 8.22 ± 0.58 | 3.34 ± 0.35 | 89.06 ± 1.15 | 6.28 ± 0.38 | 3.08 ± 0.11 | 88.23 ± 1.18 | 8.99 ± 0.43 | 3.51 ± 0.27 |
| NUCFL (MDCA+COS) | 87.03 ± 1.10 | 8.20 ± 0.41 | 3.36 ± 0.27 | 88.74 ± 1.14 | 5.71 ± 0.38 | 2.79 ± 0.26 | 87.52 ± 1.33 | **8.59** ± 0.87 | **3.45** ± 0.41 |
| NUCFL (MDCA+L-CKA) | 87.11 ± 1.33 | 7.99 ± 0.92 | 3.30 ± 0.61 | 87.96 ± 1.42 | 5.34 ± 0.91 | 2.70 ± 0.33 | 87.47 ± 1.00 | 8.61 ± 0.76 | 3.46 ± 0.41 |
| NUCFL (MDCA+RBF-CKA) | 87.39 ± 1.31 | **7.87** ± 0.61 | **3.25** ± 0.44 | 88.01 ± 1.14 | 5.29 ± 0.41 | 2.67 ± 0.29 | 87.84 ± 1.31 | 8.65 ± 0.41 | 3.47 ± 0.28 |

| Calibration Method | FedDyn (Acar et al., 2021) | | | FedNova (Wang et al., 2020b) | | |
|---|---|---|---|---|---|---|
| | Acc ↑ | ECE ↓ | SCE ↓ | Acc ↑ | ECE ↓ | SCE ↓ |
| Uncal. | 88.03 ± 1.08 | 10.51 ± 0.33 | 3.66 ± 0.21 | 87.56 ± 1.02 | 9.08 ± 0.30 | 3.50 ± 0.18 |
| Focal (Lin et al., 2017) | 87.13 ± 1.03 | 11.28 ± 0.77 | 3.69 ± 0.39 | 88.00 ± 0.94 | 12.92 ± 0.49 | 4.47 ± 0.35 |
| LS (Müller et al., 2019) | 88.34 ± 1.44 | 15.23 ± 1.31 | 5.16 ± 0.72 | 88.63 ± 0.98 | 10.77 ± 0.33 | 3.69 ± 0.25 |
| BS (Brier, 1950) | 87.90 ± 0.97 | 12.58 ± 0.54 | 4.45 ± 0.21 | 86.94 ± 0.94 | 10.13 ± 0.28 | 3.64 ± 0.15 |
| MMCE (Kumar et al., 2018) | 87.80 ± 1.19 | 15.10 ± 0.48 | 5.13 ± 0.22 | 87.11 ± 1.03 | 13.31 ± 0.73 | 4.52 ± 0.49 |
| FLSD (Mukhoti et al., 2020) | 86.90 ± 0.94 | 9.07 ± 0.53 | 3.52 ± 0.34 | 86.45 ± 1.12 | 10.29 ± 0.49 | 3.65 ± 0.30 |
| MbLS (Liu et al., 2021) | 87.16 ± 1.07 | 15.53 ± 0.61 | 5.17 ± 0.32 | 85.74 ± 0.82 | 14.84 ± 0.30 | 5.06 ± 0.29 |
| DCA (Liang et al., 2020) | 88.70 ± 1.11 | 8.92 ± 0.41 | 3.45 ± 0.19 | 89.06 ± 1.08 | 8.60 ± 0.42 | 3.47 ± 0.29 |
| NUCFL (DCA+COS) | 88.10 ± 1.09 | 8.53 ± 0.41 | 3.41 ± 0.30 | 87.96 ± 1.06 | 8.49 ± 0.30 | 3.41 ± 0.18 |
| NUCFL (DCA+L-CKA) | 88.37 ± 1.09 | **7.87** ± 0.24 | **3.26** ± 0.11 | 87.56 ± 1.03 | 8.38 ± 0.47 | 3.39 ± 0.22 |
| NUCFL (DCA+RBF-CKA) | 88.59 ± 1.10 | 8.60 ± 0.57 | 3.41 ± 0.22 | 87.34 ± 1.22 | **8.30** ± 0.17 | **3.37** ± 0.09 |
| MDCA (Hebbalaguppe et al., 2022b) | 88.10 ± 1.16 | 9.18 ± 0.44 | 3.54 ± 0.31 | 87.73 ± 1.33 | 8.67 ± 0.69 | 3.47 ± 0.42 |
| NUCFL (MDCA+COS) | 88.11 ± 1.00 | 8.75 ± 0.49 | 3.49 ± 0.32 | 88.00 ± 1.24 | 7.55 ± 0.61 | **3.27** ± 0.35 |
| NUCFL (MDCA+L-CKA) | 88.07 ± 1.13 | **8.66** ± 0.52 | **3.42** ± 0.29 | 87.59 ± 1.23 | **7.51** ± 0.61 | 3.28 ± 0.39 |
| NUCFL (MDCA+RBF-CKA) | 88.30 ± 1.29 | 8.83 ± 0.44 | 3.50 ± 0.31 | 87.96 ± 1.24 | 7.80 ± 0.39 | 3.33 ± 0.22 |

Table 14: Accuracy (%), calibration measures ECE (%), and SCE (%) of various FL algorithms with different calibration methods under non-IID ($\alpha = 0.05$) scenario on the FEMNIST dataset. Underlined values indicate the best calibration across all methods.

| Calibration | FedAvg (McMahan et al., 2017) | | | FedProx (Sahu et al., 2018) | | | Scaffold (Karimireddy et al., 2019) | | |
|---|---|---|---|---|---|---|---|---|---|
| Method | Acc ↑ | ECE ↓ | SCE ↓ | Acc ↑ | ECE ↓ | SCE ↓ | Acc ↑ | ECE ↓ | SCE ↓ |
| Uncal. | 80.23 ± 1.23 | 7.60 ± 0.53 | 2.70 ± 0.31 | 81.52 ± 0.87 | 9.41 ± 0.72 | 3.52 ± 0.26 | 81.27 ± 1.09 | 6.91 ± 0.82 | 2.35 ± 0.41 |
| Focal (Lin et al., 2017) | 79.46 ± 1.14 | 10.79 ± 0.42 | 3.51 ± 0.28 | 77.37 ± 1.09 | 12.74 ± 0.99 | 4.76 ± 0.23 | 80.78 ± 1.26 | 8.36 ± 0.72 | 3.05 ± 0.49 |
| LS (Müller et al., 2019) | 79.78 ± 1.20 | 8.30 ± 0.51 | 3.03 ± 0.41 | 79.08 ± 1.05 | 9.38 ± 0.93 | 3.53 ± 0.32 | 79.25 ± 1.42 | 9.49 ± 0.85 | 3.55 ± 0.50 |
| BS (Brier, 1950) | 76.90 ± 1.32 | 5.27 ± 0.49 | 1.64 ± 0.35 | 77.97 ± 1.07 | 8.82 ± 0.64 | 3.13 ± 0.34 | 78.54 ± 1.28 | 6.32 ± 0.75 | 2.33 ± 0.45 |
| MMCE (Kumar et al., 2018) | 79.49 ± 1.07 | 9.82 ± 0.41 | 3.50 ± 0.33 | 78.54 ± 0.91 | 8.69 ± 0.74 | 3.10 ± 0.22 | 79.26 ± 1.16 | 10.61 ± 0.73 | 3.52 ± 0.42 |
| FLSD (Mukhoti et al., 2020) | 79.76 ± 1.25 | 9.52 ± 0.61 | 3.44 ± 0.43 | 80.01 ± 1.41 | 11.46 ± 0.82 | 3.79 ± 0.54 | 80.11 ± 1.00 | 9.43 ± 0.69 | 3.55 ± 0.45 |
| MbLS (Liu et al., 2021) | 81.73 ± 2.12 | 11.17 ± 0.65 | 3.83 ± 0.40 | 79.30 ± 1.03 | 10.01 ± 0.61 | 3.49 ± 0.29 | 79.57 ± 1.24 | 9.11 ± 0.82 | 3.46 ± 0.52 |
| DCA (Liang et al., 2020) | 80.33 ± 1.09 | 6.37 ± 0.47 | 1.91 ± 0.34 | 82.51 ± 1.12 | 7.35 ± 0.64 | 2.61 ± 0.25 | 80.23 ± 1.20 | 5.02 ± 0.83 | 1.49 ± 0.44 |
| NUCFL (DCA+COS) | 80.35 ± 1.19 | 5.75 ± 0.48 | 1.75 ± 0.28 | 81.66 ± 0.95 | 6.71 ± 0.50 | 2.30 ± 0.39 | 81.32 ± 1.14 | 4.92 ± 0.77 | 1.49 ± 0.31 |
| NUCFL (DCA+L-CKA) | 81.05 ± 1.01 | 5.77 ± 0.50 | 1.75 ± 0.34 | 79.66 ± 1.13 | 6.29 ± 0.51 | 1.88 ± 0.28 | 80.92 ± 1.00 | 4.66 ± 0.81 | 1.34 ± 0.48 |
| NUCFL (DCA+RBF-CKA) | 80.48 ± 1.11 | 5.64 ± 0.68 | 1.71 ± 0.29 | 81.55 ± 1.12 | 6.64 ± 1.22 | 2.28 ± 0.52 | 81.63 ± 1.26 | 4.65 ± 0.92 | 1.35 ± 0.66 |
| MDCA (Hebbalaguppe et al., 2022b) | 81.40 ± 1.14 | 6.94 ± 0.59 | 2.37 ± 0.33 | 82.02 ± 1.24 | 7.84 ± 0.54 | 2.97 ± 0.41 | 81.41 ± 1.32 | 6.33 ± 0.81 | 2.29 ± 0.48 |
| NUCFL (MDCA+COS) | 80.80 ± 1.12 | 6.19 ± 0.64 | 1.85 ± 0.28 | 81.41 ± 1.03 | 7.07 ± 0.54 | 2.61 ± 0.30 | 81.92 ± 1.18 | 5.77 ± 0.76 | 1.72 ± 0.51 |
| NUCFL (MDCA+L-CKA) | 80.37 ± 1.04 | 5.38 ± 0.52 | 1.68 ± 0.31 | 82.00 ± 1.08 | 6.79 ± 0.56 | 2.29 ± 0.29 | 81.22 ± 1.21 | 5.35 ± 0.82 | 1.68 ± 0.50 |
| NUCFL (MDCA+RBF-CKA) | 81.22 ± 1.61 | 5.58 ± 0.51 | 1.70 ± 0.30 | 81.70 ± 1.33 | 6.67 ± 0.92 | 2.26 ± 0.71 | 81.69 ± 0.91 | 5.26 ± 0.78 | 1.65 ± 0.49 |

| Calibration | FedDyn (Acar et al., 2021) | | | FedNova (Wang et al., 2020b) | | |
|---|---|---|---|---|---|---|
| Method | Acc ↑ | ECE ↓ | SCE ↓ | Acc ↑ | ECE ↓ | SCE ↓ |
| Uncal. | 81.05 ± 1.39 | 9.11 ± 0.64 | 3.47 ± 0.53 | 81.69 ± 1.11 | 10.88 ± 0.72 | 3.52 ± 0.21 |
| Focal (Lin et al., 2017) | 79.91 ± 1.31 | 10.12 ± 0.52 | 3.49 ± 0.32 | 80.37 ± 1.13 | 13.61 ± 0.93 | 4.54 ± 0.63 |
| LS (Müller et al., 2019) | 80.49 ± 1.15 | 11.20 ± 0.62 | 3.77 ± 0.65 | 79.63 ± 1.08 | 13.37 ± 0.84 | 4.50 ± 0.43 |
| BS (Brier, 1950) | 79.41 ± 1.19 | 11.87 ± 0.91 | 3.85 ± 0.64 | 77.06 ± 1.05 | 9.04 ± 0.71 | 3.48 ± 0.23 |
| MMCE (Kumar et al., 2018) | 79.03 ± 1.02 | 12.60 ± 0.54 | 4.77 ± 0.51 | 80.22 ± 1.13 | 11.02 ± 0.81 | 3.88 ± 0.24 |
| FLSD (Mukhoti et al., 2020) | 80.09 ± 1.28 | 12.88 ± 0.65 | 4.80 ± 0.53 | 78.50 ± 1.31 | 13.54 ± 1.21 | 4.51 ± 0.71 |
| MbLS (Liu et al., 2021) | 79.65 ± 1.14 | 11.09 ± 0.54 | 3.80 ± 0.45 | 81.47 ± 1.07 | 14.63 ± 0.75 | 4.74 ± 0.25 |
| DCA (Liang et al., 2020) | 81.40 ± 1.11 | 7.83 ± 0.59 | 2.66 ± 0.41 | 81.39 ± 1.04 | 9.11 ± 0.71 | 3.49 ± 0.29 |
| NUCFL (DCA+COS) | 81.64 ± 0.92 | 6.62 ± 0.55 | 2.27 ± 0.26 | 81.74 ± 1.00 | 8.66 ± 0.91 | 3.10 ± 0.19 |
| NUCFL (DCA+L-CKA) | 80.89 ± 0.90 | 6.08 ± 0.81 | 1.82 ± 0.39 | 81.47 ± 1.08 | 7.79 ± 0.62 | 2.67 ± 0.28 |
| NUCFL (DCA+RBF-CKA) | 80.76 ± 1.07 | 6.16 ± 0.54 | 1.84 ± 0.36 | 82.04 ± 0.97 | 7.29 ± 0.61 | 2.65 ± 0.30 |
| MDCA (Hebbalaguppe et al., 2022b) | 81.12 ± 1.15 | 8.16 ± 0.68 | 3.00 ± 1.03 | 81.88 ± 1.29 | 9.07 ± 0.77 | 3.48 ± 0.33 |
| NUCFL (MDCA+COS) | 81.59 ± 1.15 | 7.20 ± 0.82 | 2.54 ± 0.40 | 82.05 ± 1.11 | 8.81 ± 0.53 | 3.29 ± 0.27 |
| NUCFL (MDCA+L-CKA) | 80.91 ± 0.84 | 6.96 ± 0.47 | 2.39 ± 0.38 | 81.36 ± 1.12 | 7.13 ± 0.93 | 2.63 ± 0.41 |
| NUCFL (MDCA+RBF-CKA) | 80.98 ± 1.11 | 7.14 ± 1.22 | 2.53 ± 0.61 | 81.56 ± 0.88 | 7.72 ± 0.57 | 2.94 ± 0.25 |

Table 15: Accuracy (%), calibration measures ECE (%), and SCE (%) of various federated optimization methods with different calibration methods under IID scenario on the CIFAR-10 dataset. Underlined values indicate the best calibration across all methods.

| Calibration | FedAvg (McMahan et al., 2017) | | | FedProx (Sahu et al., 2018) | | | Scaffold (Karimireddy et al., 2019) | | |
|---|---|---|---|---|---|---|---|---|---|
| Method | Acc ↑ | ECE ↓ | SCE ↓ | Acc ↑ | ECE ↓ | SCE ↓ | Acc ↑ | ECE ↓ | SCE ↓ |
| Uncal. | 78.83 ± 1.05 | 9.69 ± 0.65 | 3.50 ± 0.40 | 79.27 ± 1.12 | 9.36 ± 0.54 | 3.49 ± 0.34 | 79.19 ± 1.09 | 10.99 ± 0.59 | 3.55 ± 0.37 |
| Focal (Lin et al., 2017) | 74.24 ± 1.08 | 13.81 ± 0.61 | 4.71 ± 0.38 | 78.02 ± 1.11 | 10.24 ± 0.71 | 3.54 ± 0.44 | 77.00 ± 1.07 | 11.37 ± 0.54 | 3.56 ± 0.35 |
| LS (Müller et al., 2019) | 77.10 ± 1.10 | 10.04 ± 0.57 | 3.55 ± 0.35 | 77.42 ± 1.09 | 14.71 ± 0.65 | 4.72 ± 0.38 | 77.54 ± 1.12 | 14.44 ± 0.51 | 4.71 ± 0.22 |
| BS (Brier, 1950) | 78.50 ± 1.09 | 10.87 ± 0.55 | 3.58 ± 0.34 | 78.34 ± 1.13 | 10.26 ± 1.17 | 3.56 ± 0.88 | 76.40 ± 1.22 | 8.38 ± 0.58 | 3.06 ± 0.35 |
| MMCE (Kumar et al., 2018) | 74.72 ± 1.33 | 7.40 ± 1.22 | 2.71 ± 0.36 | 80.39 ± 1.07 | 14.27 ± 1.03 | 4.73 ± 0.51 | 76.53 ± 1.09 | 9.03 ± 1.00 | 3.42 ± 0.61 |
| FLSD (Mukhoti et al., 2020) | 78.15 ± 1.10 | 14.08 ± 1.11 | 4.79 ± 0.88 | 78.57 ± 1.80 | 14.29 ± 0.60 | 4.73 ± 0.36 | 75.66 ± 0.77 | 7.37 ± 0.81 | 2.65 ± 0.33 |
| MbLS (Liu et al., 2021) | 79.07 ± 0.81 | 13.70 ± 0.89 | 4.80 ± 0.61 | 78.46 ± 1.13 | 11.31 ± 0.66 | 3.56 ± 0.41 | 79.18 ± 1.10 | 13.67 ± 0.93 | 4.87 ± 0.61 |
| DCA (Liang et al., 2020) | 78.84 ± 1.18 | 8.45 ± 0.58 | 3.11 ± 0.46 | 80.11 ± 1.24 | 9.10 ± 0.54 | 3.45 ± 0.34 | 79.12 ± 1.29 | 9.23 ± 0.53 | 3.50 ± 0.34 |
| NUCFL (DCA+COS) | 79.09 ± 1.00 | 7.40 ± 0.46 | 2.72 ± 0.32 | 79.13 ± 0.91 | 8.71 ± 0.66 | 3.43 ± 0.23 | 79.33 ± 1.14 | 8.74 ± 0.49 | 3.36 ± 0.25 |
| NUCFL (DCA+L-CKA) | 78.66 ± 1.13 | 7.06 ± 0.66 | 2.60 ± 0.41 | 79.96 ± 1.12 | 8.66 ± 0.61 | 3.39 ± 0.39 | 80.54 ± 9.15 | 8.43 ± 1.00 | 3.11 ± 0.35 |
| NUCFL (DCA+RBF-CKA) | 79.94 ± 1.29 | 8.03 ± 0.45 | 3.03 ± 0.30 | 80.90 ± 1.00 | 8.90 ± 0.43 | 3.45 ± 0.29 | 79.31 ± 1.14 | 8.11 ± 0.99 | 3.06 ± 0.54 |
| MDCA (Hebbalaguppe et al., 2022b) | 78.92 ± 1.07 | 8.44 ± 1.05 | 3.11 ± 0.82 | 79.42 ± 1.12 | 8.86 ± 0.60 | 3.40 ± 0.35 | 79.66 ± 1.10 | 10.20 ± 0.67 | 3.55 ± 0.40 |
| NUCFL (MDCA+COS) | 78.67 ± 1.19 | 7.63 ± 0.89 | 2.73 ± 0.42 | 80.17 ± 0.91 | 8.67 ± 0.47 | 3.38 ± 0.29 | 80.56 ± 0.88 | 9.80 ± 0.61 | 3.48 ± 0.22 |
| NUCFL (MDCA+L-CKA) | 79.14 ± 1.29 | 6.96 ± 0.51 | 2.52 ± 0.31 | 79.94 ± 1.11 | 8.42 ± 0.71 | 3.33 ± 0.22 | 79.77 ± 1.02 | 9.37 ± 0.49 | 3.44 ± 0.35 |
| NUCFL (MDCA+RBF-CKA) | 79.08 ± 1.09 | 6.95 ± 0.54 | 2.51 ± 0.22 | 79.93 ± 1.02 | 7.99 ± 0.44 | 3.08 ± 0.29 | 80.02 ± 1.11 | 9.61 ± 0.52 | 3.47 ± 0.44 |

| Calibration | FedDyn (Acar et al., 2021) | | | FedNova (Wang et al., 2020b) | | |
|---|---|---|---|---|---|---|
| Method | Acc ↑ | ECE ↓ | SCE ↓ | Acc ↑ | ECE ↓ | SCE ↓ |
| Uncal. | 79.50 ± 1.14 | 12.84 ± 0.85 | 4.80 ± 0.48 | 78.99 ± 1.08 | 15.67 ± 0.69 | 4.94 ± 0.36 |
| Focal (Lin et al., 2017) | 76.89 ± 1.15 | 13.96 ± 0.86 | 4.89 ± 0.53 | 74.25 ± 1.09 | 10.21 ± 0.61 | 3.67 ± 0.37 |
| LS (Müller et al., 2019) | 74.76 ± 2.36 | 17.35 ± 0.79 | 7.67 ± 1.22 | 77.57 ± 1.10 | 14.40 ± 0.56 | 4.70 ± 0.61 |
| BS (Brier, 1950) | 78.13 ± 1.14 | 14.35 ± 1.74 | 4.70 ± 0.66 | 79.00 ± 1.12 | 15.80 ± 0.58 | 4.98 ± 0.39 |
| MMCE (Kumar et al., 2018) | 76.72 ± 1.26 | 10.23 ± 1.08 | 3.53 ± 0.61 | 76.20 ± 1.12 | 14.34 ± 0.82 | 4.71 ± 0.55 |
| FLSD (Mukhoti et al., 2020) | 74.84 ± 1.13 | 9.53 ± 0.73 | 3.51 ± 0.49 | 71.66 ± 1.09 | 9.82 ± 0.64 | 3.65 ± 0.41 |
| MbLS (Liu et al., 2021) | 77.66 ± 1.12 | 10.62 ± 1.79 | 3.50 ± 0.61 | 79.09 ± 1.10 | 17.03 ± 0.65 | 7.63 ± 0.36 |
| DCA (Liang et al., 2020) | 80.11 ± 1.22 | 10.00 ± 0.65 | 3.48 ± 0.56 | 78.93 ± 1.19 | 13.72 ± 0.59 | 4.87 ± 0.42 |
| NUCFL (DCA+COS) | 79.80 ± 1.13 | 9.27 ± 0.61 | 3.49 ± 0.39 | 79.11 ± 0.91 | 11.94 ± 1.11 | 3.69 ± 0.66 |
| NUCFL (DCA+L-CKA) | 80.88 ± 1.20 | 8.62 ± 0.71 | 3.34 ± 0.29 | 78.49 ± 1.12 | 10.19 ± 0.68 | 3.59 ± 0.44 |
| NUCFL (DCA+RBF-CKA) | 79.95 ± 1.10 | 8.58 ± 1.50 | 3.16 ± 0.62 | 79.04 ± 0.82 | 10.70 ± 0.87 | 3.62 ± 0.44 |
| MDCA (Hebbalaguppe et al., 2022b) | 79.20 ± 1.13 | 10.27 ± 0.46 | 3.47 ± 0.21 | 79.00 ± 1.22 | 13.27 ± 0.61 | 4.87 ± 0.39 |
| NUCFL (MDCA+COS) | 79.30 ± 0.91 | 8.23 ± 0.59 | 3.10 ± 0.44 | 79.53 ± 1.41 | 11.74 ± 1.23 | 3.67 ± 0.66 |
| NUCFL (MDCA+L-CKA) | 80.17 ± 1.22 | 8.47 ± 0.66 | 3.13 ± 0.42 | 79.05 ± 1.12 | 10.21 ± 0.55 | 3.60 ± 0.34 |
| NUCFL (MDCA+RBF-CKA) | 79.61 ± 1.08 | 9.08 ± 0.51 | 3.34 ± 0.60 | 80.07 ± 1.10 | 10.49 ± 1.06 | 3.62 ± 0.48 |

Table 16: Accuracy (%), calibration measures ECE (%), and SCE (%) of various federated optimization methods with different calibration methods under non-IID ($\alpha = 0.5$) scenario on the CIFAR-10 dataset. Underlined values indicate the best calibration across all methods.

| Calibration Method | FedAvg (McMahan et al., 2017) | | | FedProx (Sahu et al., 2018) | | | Scaffold (Karimireddy et al., 2019) | | |
|---|---|---|---|---|---|---|---|---|---|
| | Acc ↑ | ECE ↓ | SCE ↓ | Acc ↑ | ECE ↓ | SCE ↓ | Acc ↑ | ECE ↓ | SCE ↓ |
| Uncal. | 64.86 ± 1.21 | 12.62 ± 1.13 | 4.42 ± 0.92 | 66.77 ± 1.61 | 13.19 ± 1.53 | 4.48 ± 0.85 | 66.34 ± 1.18 | 14.95 ± 1.10 | 4.69 ± 0.90 |
| Focal (Lin et al., 2017) | 60.18 ± 1.00 | 15.21 ± 2.08 | 4.81 ± 0.87 | 65.25 ± 1.27 | 16.20 ± 1.32 | 4.99 ± 0.82 | 63.48 ± 1.33 | 11.02 ± 1.07 | 3.61 ± 1.09 |
| LS (Müller et al., 2019) | 62.13 ± 1.20 | 12.92 ± 1.07 | 4.45 ± 0.86 | 57.29 ± 1.64 | 10.99 ± 1.24 | 3.60 ± 0.88 | 61.94 ± 1.71 | 10.27 ± 1.12 | 3.55 ± 0.91 |
| BS (Brier, 1950) | 59.18 ± 2.16 | 9.21 ± 1.10 | 3.50 ± 0.89 | 62.16 ± 2.23 | 10.61 ± 1.16 | 3.56 ± 0.82 | 68.09 ± 2.20 | 19.42 ± 2.12 | 9.59 ± 1.91 |
| MMCE (Kumar et al., 2018) | 65.62 ± 1.39 | 18.25 ± 2.18 | 9.05 ± 1.37 | 68.76 ± 2.07 | 16.91 ± 1.80 | 4.93 ± 1.05 | 64.90 ± 1.22 | 9.02 ± 1.16 | 3.42 ± 0.89 |
| FLSD (Mukhoti et al., 2020) | 68.31 ± 2.17 | 17.21 ± 1.92 | 8.65 ± 1.31 | 63.21 ± 1.24 | 10.25 ± 1.66 | 3.54 ± 0.85 | 69.03 ± 3.20 | 18.53 ± 2.18 | 9.13 ± 1.17 |
| MbLS (Liu et al., 2021) | 63.06 ± 1.22 | 12.77 ± 1.26 | 4.45 ± 0.98 | 60.92 ± 1.26 | 10.03 ± 1.14 | 3.50 ± 0.85 | 64.95 ± 1.21 | 14.91 ± 1.20 | 4.71 ± 0.96 |
| DCA (Liang et al., 2020) | 64.80 ± 1.20 | 10.32 ± 1.10 | 3.51 ± 0.82 | 67.23 ± 1.25 | 11.47 ± 1.17 | 3.64 ± 0.86 | 66.99 ± 1.19 | 13.90 ± 1.14 | 4.50 ± 0.89 |
| NUCFL (DCA+COS) | 65.60 ± 1.42 | 9.94 ± 1.32 | 3.47 ± 1.05 | 67.24 ± 1.11 | 10.74 ± 1.22 | 3.58 ± 1.13 | 67.47 ± 0.97 | 12.75 ± 1.62 | 4.40 ± 0.89 |
| NUCFL (DCA+L-CKA) | 64.77 ± 1.03 | 8.43 ± 1.00 | 3.12 ± 0.89 | 67.03 ± 1.22 | 10.42 ± 1.41 | 3.56 ± 1.06 | 66.99 ± 1.17 | 12.27 ± 1.45 | 4.38 ± 1.05 |
| NUCFL (DCA+RBF-CKA) | 65.00 ± 1.29 | 8.39 ± 1.38 | 3.12 ± 0.72 | 67.08 ± 1.23 | 9.99 ± 2.12 | 3.50 ± 1.08 | 66.17 ± 1.19 | 12.14 ± 1.10 | 4.36 ± 0.95 |
| MDCA (Hebbalaguppe et al., 2022b) | 65.02 ± 1.77 | 10.93 ± 1.08 | 3.54 ± 1.24 | 67.48 ± 1.22 | 10.20 ± 1.39 | 3.54 ± 0.86 | 67.55 ± 1.02 | 13.77 ± 2.15 | 4.49 ± 1.93 |
| NUCFL (MDCA+COS) | 65.25 ± 1.88 | 9.74 ± 1.05 | 3.44 ± 1.01 | 66.54 ± 1.39 | 9.37 ± 1.37 | 3.47 ± 0.85 | 67.62 ± 1.45 | 12.80 ± 1.11 | 4.40 ± 1.37 |
| NUCFL (MDCA+L-CKA) | 64.97 ± 1.03 | 9.27 ± 1.06 | 3.41 ± 1.03 | 66.91 ± 1.06 | 9.01 ± 1.77 | 3.32 ± 1.24 | 66.94 ± 1.03 | 12.63 ± 3.13 | 4.40 ± 2.39 |
| NUCFL (MDCA+RBF-CKA) | 65.07 ± 1.69 | 10.16 ± 1.07 | 3.52 ± 1.33 | 66.92 ± 2.09 | 9.84 ± 1.35 | 3.50 ± 1.03 | 67.04 ± 1.19 | 11.06 ± 2.10 | 3.96 ± 1.88 |

| Calibration Method | FedDyn (Acar et al., 2021) | | | FedNova (Wang et al., 2020b) | | |
|---|---|---|---|---|---|---|
| | Acc ↑ | ECE ↓ | SCE ↓ | Acc ↑ | ECE ↓ | SCE ↓ |
| Uncal. | 62.26 ± 1.58 | 13.03 ± 1.09 | 4.47 ± 0.99 | 66.71 ± 1.91 | 19.29 ± 2.42 | 9.48 ± 2.15 |
| LS (Müller et al., 2019) | 64.12 ± 1.25 | 15.09 ± 1.08 | 4.68 ± 0.80 | 63.71 ± 1.30 | 13.66 ± 1.35 | 4.57 ± 0.93 |
| BS (Brier, 1950) | 63.62 ± 1.26 | 19.23 ± 1.19 | 9.33 ± 0.95 | 60.25 ± 1.24 | 9.54 ± 1.41 | 3.58 ± 0.97 |
| MMCE (Kumar et al., 2018) | 63.99 ± 1.25 | 15.53 ± 1.21 | 4.71 ± 0.93 | 64.50 ± 1.41 | 14.95 ± 2.35 | 4.69 ± 2.00 |
| FLSD (Mukhoti et al., 2020) | 64.69 ± 1.27 | 20.53 ± 2.72 | 9.70 ± 1.95 | 63.58 ± 1.25 | 14.67 ± 1.33 | 4.69 ± 1.02 |
| MbLS (Liu et al., 2021) | 64.16 ± 1.24 | 17.37 ± 1.17 | 7.90 ± 0.93 | 62.64 ± 1.23 | 10.69 ± 1.30 | 3.58 ± 0.92 |
| DCA (Liang et al., 2020) | 65.88 ± 1.23 | 13.00 ± 1.16 | 4.44 ± 0.87 | 67.09 ± 1.21 | 11.11 ± 1.28 | 3.64 ± 0.91 |
| NUCFL (DCA+COS) | 65.98 ± 0.95 | 12.19 ± 2.06 | 4.34 ± 1.39 | 67.54 ± 1.23 | 9.93 ± 1.77 | 3.70 ± 1.25 |
| NUCFL (DCA+L-CKA) | 64.88 ± 1.52 | 11.39 ± 1.28 | 4.07 ± 1.55 | 67.41 ± 1.36 | 9.15 ± 1.18 | 3.67 ± 0.77 |
| NUCFL (DCA+RBF-CKA) | 64.77 ± 2.32 | 12.27 ± 1.48 | 4.36 ± 1.00 | 67.20 ± 1.18 | 10.19 ± 1.16 | 3.71 ± 0.92 |
| MDCA (Hebbalaguppe et al., 2022b) | 65.70 ± 1.24 | 13.56 ± 1.33 | 4.47 ± 1.08 | 67.88 ± 2.01 | 12.56 ± 1.19 | 3.97 ± 1.68 |
| NUCFL (MDCA+COS) | 65.33 ± 1.42 | 12.03 ± 1.32 | 4.32 ± 0.97 | 66.68 ± 1.99 | 10.52 ± 1.04 | 3.61 ± 1.00 |
| NUCFL (MDCA+L-CKA) | 65.75 ± 1.23 | 12.38 ± 1.44 | 4.36 ± 1.07 | 66.50 ± 1.40 | 10.12 ± 2.46 | 3.60 ± 1.65 |
| NUCFL (MDCA+RBF-CKA) | 65.88 ± 2.01 | 12.44 ± 2.30 | 4.36 ± 2.15 | 66.45 ± 1.28 | 11.36 ± 1.00 | 3.66 ± 1.09 |

Table 17: Accuracy (%), calibration measures ECE (%), and SCE (%) of various federated optimization methods with different calibration methods under non-IID ($\alpha = 0.1$) scenario on the CIFAR-10 dataset. Underlined values indicate the best calibration across all methods.

| Calibration Method | FedAvg (McMahan et al., 2017) | | | FedProx (Sahu et al., 2018) | | | Scaffold (Karimireddy et al., 2019) | | |
|---|---|---|---|---|---|---|---|---|---|
| | Acc ↑ | ECE ↓ | SCE ↓ | Acc ↑ | ECE ↓ | SCE ↓ | Acc ↑ | ECE ↓ | SCE ↓ |
| Uncal. | 59.84 ± 1.20 | 14.30 ± 1.03 | 4.60 ± 1.00 | 61.50 ± 1.25 | 16.03 ± 1.18 | 4.99 ± 0.95 | 61.99 ± 1.18 | 13.88 ± 1.10 | 4.61 ± 1.14 |
| Focal (Lin et al., 2017) | 53.13 ± 1.20 | 10.08 ± 1.08 | 3.52 ± 0.98 | 55.91 ± 1.22 | 11.72 ± 1.10 | 4.18 ± 0.96 | 59.30 ± 1.17 | 12.03 ± 1.11 | 4.52 ± 1.10 |
| LS (Müller et al., 2019) | 59.86 ± 1.18 | 18.10 ± 1.07 | 7.63 ± 0.95 | 60.91 ± 1.71 | 22.68 ± 1.14 | 8.54 ± 0.92 | 60.23 ± 1.55 | 17.79 ± 1.29 | 5.34 ± 1.09 |
| BS (Brier, 1950) | 61.80 ± 1.12 | 22.48 ± 1.13 | 8.57 ± 1.01 | 60.75 ± 1.21 | 17.98 ± 1.88 | 6.36 ± 1.94 | 59.20 ± 1.15 | 10.42 ± 1.08 | 4.08 ± 1.05 |
| MMCE (Kumar et al., 2018) | 60.72 ± 1.19 | 20.59 ± 1.07 | 8.11 ± 0.96 | 58.96 ± 1.37 | 13.52 ± 1.10 | 4.53 ± 0.66 | 59.50 ± 1.17 | 13.69 ± 1.09 | 4.64 ± 1.07 |
| FLSD (Mukhoti et al., 2020) | 58.83 ± 1.62 | 18.22 ± 1.06 | 7.72 ± 0.97 | 59.22 ± 1.21 | 10.01 ± 1.19 | 3.49 ± 0.93 | 60.65 ± 1.40 | 15.15 ± 1.10 | 4.79 ± 1.50 |
| MbLS (Liu et al., 2021) | 60.73 ± 1.20 | 18.81 ± 2.15 | 7.76 ± 1.61 | 62.09 ± 1.52 | 20.28 ± 2.11 | 8.13 ± 2.31 | 60.27 ± 2.18 | 14.51 ± 1.13 | 4.78 ± 1.13 |
| DCA (Liang et al., 2020) | 59.94 ± 1.17 | 13.07 ± 1.06 | 4.31 ± 0.93 | 62.28 ± 1.22 | 14.57 ± 1.66 | 4.64 ± 0.89 | 60.78 ± 1.15 | 11.07 ± 1.07 | 4.10 ± 1.04 |
| NUCFL (DCA+COS) | 60.46 ± 1.33 | 11.28 ± 1.72 | 4.19 ± 0.94 | 62.00 ± 1.03 | 13.84 ± 1.13 | 4.59 ± 0.92 | 62.06 ± 1.16 | 10.53 ± 1.29 | 4.03 ± 1.02 |
| NUCFL (DCA+L-CKA) | 60.30 ± 1.20 | 10.62 ± 1.02 | 3.92 ± 0.92 | 62.11 ± 1.23 | 13.03 ± 1.00 | 4.57 ± 0.42 | 62.09 ± 1.18 | 9.06 ± 1.33 | 3.84 ± 1.20 |
| NUCFL (DCA+RBF-CKA) | 60.52 ± 1.21 | 10.76 ± 1.24 | 3.94 ± 0.91 | 62.90 ± 1.22 | 13.90 ± 1.11 | 4.60 ± 0.88 | 61.99 ± 1.00 | 10.14 ± 1.52 | 4.00 ± 1.11 |
| MDCA (Hebbalaguppe et al., 2022b) | 60.15 ± 1.71 | 12.99 ± 1.05 | 4.30 ± 1.16 | 62.33 ± 1.21 | 15.18 ± 1.11 | 4.70 ± 0.91 | 61.22 ± 1.22 | 12.26 ± 1.16 | 4.50 ± 1.52 |
| NUCFL (MDCA+COS) | 60.00 ± 1.92 | 10.53 ± 1.03 | 3.90 ± 1.15 | 62.20 ± 1.73 | 14.01 ± 1.09 | 4.61 ± 0.90 | 61.37 ± 1.33 | 12.00 ± 1.37 | 4.50 ± 1.00 |
| NUCFL (MDCA+L-CKA) | 60.49 ± 1.72 | 10.05 ± 1.01 | 3.88 ± 1.14 | 62.01 ± 1.22 | 13.92 ± 1.08 | 4.61 ± 0.89 | 62.38 ± 1.32 | 11.35 ± 1.13 | 4.17 ± 1.71 |
| NUCFL (MDCA+RBF-CKA) | 59.92 ± 1.32 | 11.03 ± 1.02 | 4.00 ± 1.13 | 61.62 ± 1.33 | 13.44 ± 1.32 | 4.53 ± 0.78 | 61.69 ± 1.00 | 10.39 ± 1.12 | 4.00 ± 1.00 |

| Calibration Method | FedDyn (Acar et al., 2021) | | | FedNova (Wang et al., 2020b) | | |
|---|---|---|---|---|---|---|
| | Acc ↑ | ECE ↓ | SCE ↓ | Acc ↑ | ECE ↓ | SCE ↓ |
| Uncal. | 60.00 ± 1.22 | 15.35 ± 1.05 | 4.68 ± 0.87 | 62.07 ± 1.23 | 16.45 ± 1.19 | 5.05 ± 0.98 |
| Focal (Lin et al., 2017) | 55.12 ± 1.21 | 10.35 ± 1.07 | 3.57 ± 0.89 | 60.22 ± 1.22 | 15.68 ± 1.18 | 4.92 ± 0.97 |
| LS (Müller et al., 2019) | 61.26 ± 2.22 | 19.35 ± 1.22 | 6.96 ± 1.12 | 59.71 ± 1.24 | 17.29 ± 1.25 | 5.18 ± 1.03 |
| BS (Brier, 1950) | 59.62 ± 1.18 | 14.23 ± 1.13 | 4.33 ± 0.93 | 60.25 ± 1.20 | 19.54 ± 1.20 | 8.40 ± 1.01 |
| MMCE (Kumar et al., 2018) | 59.39 ± 1.20 | 19.78 ± 1.67 | 7.00 ± 1.32 | 61.11 ± 1.21 | 17.95 ± 1.17 | 5.42 ± 1.08 |
| FLSD (Mukhoti et al., 2020) | 58.63 ± 1.22 | 12.60 ± 1.12 | 4.33 ± 0.95 | 60.58 ± 1.23 | 17.67 ± 1.19 | 5.22 ± 1.09 |
| MbLS (Liu et al., 2021) | 60.13 ± 1.21 | 17.37 ± 1.42 | 5.34 ± 1.02 | 62.64 ± 1.22 | 20.69 ± 1.16 | 7.88 ± 0.95 |
| DCA (Liang et al., 2020) | 60.95 ± 1.32 | 13.60 ± 1.09 | 4.60 ± 0.71 | 62.11 ± 1.20 | 15.39 ± 1.16 | 4.90 ± 0.95 |
| NUCFL (DCA+COS) | 60.26 ± 1.62 | 12.17 ± 1.38 | 4.49 ± 0.88 | 62.54 ± 1.22 | 14.13 ± 1.37 | 4.70 ± 1.00 |
| NUCFL (DCA+L-CKA) | 60.68 ± 1.24 | 11.55 ± 1.21 | 4.20 ± 0.32 | 62.41 ± 1.42 | 13.00 ± 1.16 | 4.58 ± 0.99 |
| NUCFL (DCA+RBF-CKA) | 60.67 ± 1.42 | 11.27 ± 1.09 | 4.15 ± 0.87 | 61.99 ± 1.42 | 13.19 ± 1.88 | 4.58 ± 1.38 |
| MDCA (Hebbalaguppe et al., 2022b) | 61.09 ± 1.18 | 14.32 ± 1.10 | 4.64 ± 0.94 | 62.80 ± 1.61 | 15.00 ± 1.32 | 4.87 ± 1.04 |
| NUCFL (MDCA+COS) | 60.71 ± 1.19 | 14.56 ± 1.11 | 4.86 ± 0.93 | 62.68 ± 1.28 | 14.17 ± 1.42 | 4.71 ± 1.21 |
| NUCFL (MDCA+L-CKA) | 61.75 ± 1.18 | 12.81 ± 1.92 | 4.52 ± 1.32 | 62.50 ± 1.58 | 14.12 ± 1.29 | 4.71 ± 1.33 |
| NUCFL (MDCA+RBF-CKA) | 60.89 ± 1.32 | 13.18 ± 1.09 | 4.61 ± 0.48 | 62.45 ± 1.57 | 14.36 ± 1.00 | 4.73 ± 0.43 |

Table 18: Accuracy (%), calibration measures ECE (%), and SCE (%) of various federated optimization methods with different calibration methods under non-IID ($\alpha = 0.05$) scenario on the CIFAR-10 dataset. Underlined values indicate the best calibration across all methods.

| Calibration Method | FedAvg McMahan et al. (2017) | | | FedProx Sahu et al. (2018) | | | Scaffold Karimireddy et al. (2019) | | |
|---|---|---|---|---|---|---|---|---|---|
| | Acc ↑ | ECE ↓ | SCE ↓ | Acc ↑ | ECE ↓ | SCE ↓ | Acc ↑ | ECE ↓ | SCE ↓ |
| Uncal. | 65.19 ± 1.25 | 7.61 ± 1.04 | 3.25 ± 0.89 | 66.13 ± 1.33 | 6.52 ± 1.04 | 3.19 ± 0.95 | 66.29 ± 1.30 | 8.24 ± 1.47 | 3.37 ± 1.08 |
| Focal Lin et al. (2017) | 63.88 ± 1.61 | 11.21 ± 1.32 | 3.78 ± 1.09 | 65.92 ± 1.55 | 9.25 ± 1.24 | 3.55 ± 1.07 | 65.03 ± 1.04 | 10.00 ± 0.92 | 3.59 ± 0.58 |
| LS Müller et al. (2019) | 63.17 ± 1.67 | 12.19 ± 1.08 | 3.88 ± 0.94 | 66.80 ± 1.30 | 10.21 ± 1.49 | 6.74 ± 1.10 | 65.92 ± 1.27 | 7.01 ± 1.35 | 3.67 ± 1.11 |
| BS Brier (1950) | 65.00 ± 1.71 | 8.51 ± 1.24 | 3.38 ± 1.00 | 65.19 ± 1.38 | 9.15 ± 1.20 | 3.51 ± 0.92 | 63.82 ± 1.75 | 7.11 ± 1.06 | 3.04 ± 0.81 |
| MMCE Kumar et al. (2018) | 65.03 ± 1.61 | 10.01 ± 1.20 | 3.73 ± 1.01 | 64.82 ± 1.79 | 9.20 ± 1.22 | 3.54 ± 1.41 | 65.22 ± 1.91 | 8.92 ± 1.85 | 3.50 ± 1.38 |
| FLSD Mukhoti et al. (2020) | 63.11 ± 1.37 | 8.15 ± 1.04 | 3.36 ± 0.50 | 61.85 ± 1.21 | 8.18 ± 1.04 | 3.36 ± 0.92 | 64.37 ± 1.69 | 9.27 ± 1.47 | 3.55 ± 1.24 |
| MbLS Liu et al. (2021) | 64.17 ± 1.95 | 8.60 ± 1.64 | 3.47 ± 1.41 | 64.92 ± 1.63 | 7.63 ± 1.25 | 3.23 ± 1.00 | 65.99 ± 1.24 | 9.20 ± 1.06 | 3.52 ± 1.04 |
| DCA Liang et al. (2020) | 65.63 ± 1.14 | 6.11 ± 1.05 | 3.06 ± 0.92 | 66.71 ± 1.23 | 5.95 ± 1.07 | 2.97 ± 0.85 | 66.58 ± 1.30 | 7.11 ± 1.08 | 3.03 ± 0.92 |
| NUCFL (DCA+COS) | 65.66 ± 1.07 | 6.04 ± 1.10 | 3.01 ± 0.85 | 66.73 ± 1.02 | 5.61 ± 0.96 | 2.92 ± 0.59 | 66.92 ± 0.95 | 3.00 ± 0.90 |
| NUCFL (DCA+L-CKA) | 65.43 ± 1.12 | 5.98 ± 1.03 | 2.99 ± 0.78 | 66.66 ± 1.25 | 5.54 ± 1.20 | 2.68 ± 0.92 | 66.28 ± 0.98 | 6.77 ± 0.75 | 2.96 ± 0.61 |
| NUCFL (DCA+RBF-CKA) | 65.31 ± 0.95 | 5.94 ± 1.13 | 2.96 ± 0.92 | 66.17 ± 1.06 | 5.58 ± 1.03 | 2.70 ± 0.78 | 66.37 ± 0.98 | 6.79 ± 1.03 | 2.96 ± 0.85 |
| MDCA Hebbalaguppe et al. (2022b) | 65.32 ± 1.25 | 6.25 ± 1.06 | 3.12 ± 0.69 | 65.83 ± 1.28 | 6.04 ± 1.04 | 3.02 ± 0.56 | 66.71 ± 1.35 | 7.09 ± 3.00 | 3.51 ± 1.05 |
| NUCFL (MDCA+COS) | 65.41 ± 1.04 | 6.17 ± 1.11 | 3.10 ± 0.82 | 65.92 ± 1.44 | 5.92 ± 1.24 | 2.95 ± 0.93 | 66.65 ± 1.04 | 6.83 ± 1.22 | 2.99 ± 0.91 |
| NUCFL (MDCA+L-CKA) | 65.50 ± 1.27 | 6.04 ± 1.09 | 3.02 ± 0.66 | 65.85 ± 1.03 | 5.71 ± 1.00 | 2.93 ± 0.85 | 66.70 ± 1.19 | 6.77 ± 1.06 | 2.97 ± 0.83 |
| NUCFL (MDCA+RBF-CKA) | 65.44 ± 1.06 | 6.08 ± 1.23 | 3.05 ± 0.92 | 65.88 ± 1.11 | 5.82 ± 1.35 | 2.93 ± 1.06 | 66.73 ± 1.04 | 6.85 ± 1.20 | 2.99 ± 0.97 |

| Calibration Method | FedDyn Acar et al. (2021) | | | FedNova Wang et al. (2020b) | | |
|---|---|---|---|---|---|---|
| | Acc ↑ | ECE ↓ | SCE ↓ | Acc ↑ | ECE ↓ | SCE ↓ |
| Uncal. | 65.88 ± 1.52 | 9.27 ± 1.47 | 3.55 ± 1.06 | 67.42 ± 1.10 | 9.39 ± 0.99 | 3.58 ± 0.63 |
| Focal Lin et al. (2017) | 63.77 ± 1.31 | 9.04 ± 1.29 | 3.50 ± 1.11 | 64.95 ± 1.69 | 10.00 ± 1.52 | 3.70 ± 1.14 |
| LS Müller et al. (2019) | 64.49 ± 1.59 | 11.52 ± 1.37 | 3.83 ± 1.21 | 66.71 ± 1.71 | 12.61 ± 1.04 | 3.72 ± 0.92 |
| BS Brier (1950) | 65.92 ± 1.06 | 10.62 ± 0.94 | 3.66 ± 0.70 | 65.17 ± 1.49 | 7.91 ± 1.24 | 3.30 ± 1.00 |
| MMCE Kumar et al. (2018) | 65.61 ± 1.45 | 9.11 ± 1.27 | 3.56 ± 1.04 | 65.92 ± 1.52 | 10.55 ± 1.22 | 3.63 ± 0.89 |
| FLSD Mukhoti et al. (2020) | 63.33 ± 1.93 | 8.21 ± 1.40 | 3.37 ± 1.31 | 64.89 ± 1.91 | 9.99 ± 1.74 | 3.70 ± 1.56 |
| MbLS Liu et al. (2021) | 63.79 ± 0.92 | 10.91 ± 0.49 | 3.80 ± 0.37 | 63.21 ± 1.04 | 10.25 ± 1.28 | 3.77 ± 0.91 |
| DCA Liang et al. (2020) | 66.27 ± 1.34 | 8.20 ± 0.92 | 3.37 ± 0.83 | 67.39 ± 1.49 | 8.08 ± 1.93 | 3.31 ± 0.91 |
| NUCFL (DCA+COS) | 66.15 ± 1.23 | 8.04 ± 1.05 | 3.29 ± 0.71 | 67.44 ± 1.02 | 7.81 ± 1.00 | 3.26 ± 0.87 |
| NUCFL (DCA+L-CKA) | 66.02 ± 1.04 | 8.04 ± 1.23 | 3.28 ± 1.02 | 67.52 ± 1.02 | 7.77 ± 1.23 | 3.24 ± 0.95 |
| NUCFL (DCA+RBF-CKA) | 65.92 ± 1.31 | 8.01 ± 1.06 | 3.26 ± 0.91 | 67.61 ± 1.23 | 7.79 ± 1.04 | 3.26 ± 0.85 |
| MDCA Hebbalaguppe et al. (2022b) | 66.35 ± 1.27 | 8.24 ± 1.23 | 3.33 ± 1.07 | 67.90 ± 1.39 | 8.14 ± 1.08 | 3.35 ± 0.91 |
| NUCFL (MDCA+COS) | 66.27 ± 1.04 | 7.96 ± 1.00 | 3.27 ± 0.74 | 67.73 ± 1.09 | 8.02 ± 1.13 | 3.27 ± 0.82 |
| NUCFL (MDCA+L-CKA) | 66.03 ± 1.33 | 7.90 ± 1.25 | 3.24 ± 0.99 | 67.92 ± 1.31 | 7.85 ± 1.42 | 3.27 ± 1.11 |
| NUCFL (MDCA+RBF-CKA) | 66.19 ± 1.06 | 8.00 ± 0.96 | 3.26 ± 0.58 | 67.85 ± 1.06 | 7.89 ± 1.04 | 3.28 ± 0.81 |

Table 19: Accuracy (%), calibration measures ECE (%), and SCE (%) of various federated optimization methods with different calibration methods under IID scenario on the CIFAR-100 dataset. Underlined values indicate the best calibration across all methods.

| Calibration Method | FedAvg (McMahan et al., 2017) | | | FedProx (Sahu et al., 2018) | | | Scaffold (Karimireddy et al., 2019) | | |
|---|---|---|---|---|---|---|---|---|---|
| | Acc ↑ | ECE ↓ | SCE ↓ | Acc ↑ | ECE ↓ | SCE ↓ | Acc ↑ | ECE ↓ | SCE ↓ |
| Uncal. | 61.34 ± 1.31 | 10.52 ± 1.17 | 3.61 ± 0.93 | 61.88 ± 1.55 | 9.37 ± 1.21 | 3.58 ± 1.05 | 62.08 ± 0.97 | 11.42 ± 2.05 | 3.82 ± 1.55 |
| Focal (Lin et al., 2017) | 60.59 ± 1.42 | 12.88 ± 0.95 | 3.74 ± 1.35 | 62.07 ± 1.39 | 11.45 ± 1.04 | 3.79 ± 0.91 | 61.39 ± 1.15 | 13.28 ± 1.71 | 4.88 ± 1.30 |
| LS (Müller et al., 2019) | 59.35 ± 2.15 | 15.61 ± 1.27 | 6.28 ± 1.74 | 62.23 ± 1.48 | 14.37 ± 1.35 | 6.14 ± 1.55 | 62.15 ± 1.53 | 11.69 ± 1.84 | 4.05 ± 1.07 |
| BS (Brier, 1950) | 61.20 ± 1.95 | 11.32 ± 1.44 | 3.80 ± 1.10 | 60.43 ± 1.04 | 10.15 ± 1.38 | 3.61 ± 1.17 | 60.39 ± 1.30 | 10.92 ± 1.33 | 3.80 ± 1.10 |
| MMCE (Kumar et al., 2018) | 60.00 ± 1.07 | 13.41 ± 2.05 | 4.93 ± 1.72 | 60.11 ± 1.25 | 11.32 ± 1.22 | 3.77 ± 1.39 | 61.03 ± 1.22 | 11.37 ± 1.52 | 3.83 ± 1.08 |
| FLSD (Mukhoti et al., 2020) | 58.71 ± 2.59 | 11.51 ± 1.33 | 3.92 ± 1.05 | 59.28 ± 1.49 | 10.66 ± 1.39 | 3.81 ± 1.04 | 60.49 ± 1.35 | 13.61 ± 1.51 | 5.15 ± 0.99 |
| MbLS (Liu et al., 2021) | 59.62 ± 2.31 | 9.42 ± 1.58 | 3.55 ± 0.96 | 61.93 ± 1.11 | 11.37 ± 1.90 | 3.84 ± 1.25 | 61.38 ± 1.67 | 11.99 ± 1.93 | 4.07 ± 1.25 |
| DCA (Liang et al., 2020) | 61.24 ± 1.20 | 7.69 ± 1.11 | 3.24 ± 1.03 | 61.93 ± 1.17 | 8.74 ± 1.22 | 3.40 ± 0.93 | 62.11 ± 1.20 | 9.25 ± 1.32 | 3.55 ± 0.88 |
| NUCFL (DCA+COS) | 61.88 ± 0.95 | 6.21 ± 0.95 | 3.11 ± 0.84 | 62.38 ± 1.04 | 8.15 ± 1.01 | 3.35 ± 0.77 | 62.17 ± 1.06 | 8.88 ± 1.04 | 3.50 ± 0.79 |
| NUCFL (DCA+L-CKA) | 62.05 ± 1.17 | 6.14 ± 1.03 | 3.07 ± 0.95 | 62.31 ± 1.21 | 8.04 ± 1.19 | 3.30 ± 0.95 | 62.25 ± 1.14 | 8.41 ± 1.15 | 3.45 ± 1.00 |
| NUCFL (DCA+RBF-CKA) | 61.59 ± 0.95 | 6.19 ± 1.22 | 3.11 ± 1.13 | 61.89 ± 1.10 | 8.17 ± 1.04 | 3.35 ± 0.73 | 61.94 ± 0.92 | 8.52 ± 2.00 | 3.45 ± 1.33 |
| MDCA (Hebbalaguppe et al., 2022b) | 61.03 ± 1.31 | 7.71 ± 1.50 | 3.29 ± 1.19 | 62.00 ± 1.73 | 8.21 ± 1.45 | 3.37 ± 1.10 | 62.23 ± 1.10 | 9.04 ± 1.29 | 3.51 ± 1.00 |
| NUCFL (MDCA+COS) | 62.00 ± 1.33 | 6.38 ± 1.07 | 3.14 ± 0.98 | 61.93 ± 1.34 | 7.94 ± 1.29 | 3.29 ± 1.33 | 62.17 ± 0.87 | 8.31 ± 1.17 | 3.40 ± 0.75 |
| NUCFL (MDCA+L-CKA) | 62.17 ± 1.40 | 6.25 ± 1.43 | 3.11 ± 1.24 | 62.03 ± 1.25 | 7.88 ± 1.04 | 3.25 ± 0.95 | 62.22 ± 1.04 | 8.30 ± 1.25 | 3.40 ± 0.95 |
| NUCFL (MDCA+RBF-CKA) | 61.54 ± 1.08 | 6.20 ± 1.66 | 3.09 ± 1.39 | 61.79 ± 1.04 | 8.02 ± 1.38 | 3.29 ± 1.07 | 62.15 ± 1.22 | 8.42 ± 1.03 | 3.45 ± 1.11 |

| Calibration Method | FedDyn (Acar et al., 2021) | | | FedNova (Wang et al., 2020b) | | |
|---|---|---|---|---|---|---|
| | Acc ↑ | ECE ↓ | SCE ↓ | Acc ↑ | ECE ↓ | SCE ↓ |
| Uncal. | 62.39 ± 1.25 | 12.53 ± 1.25 | 3.84 ± 1.10 | 63.01 ± 1.62 | 11.45 ± 1.05 | 3.82 ± 0.79 |
| Focal (Lin et al., 2017) | 60.21 ± 1.04 | 11.22 ± 1.00 | 3.81 ± 0.95 | 61.15 ± 1.82 | 13.21 ± 1.34 | 4.71 ± 1.06 |
| LS (Müller et al., 2019) | 61.34 ± 1.45 | 15.19 ± 2.08 | 6.58 ± 1.54 | 63.05 ± 2.25 | 16.03 ± 2.00 | 6.89 ± 1.85 |
| BS (Brier, 1950) | 62.17 ± 1.42 | 13.81 ± 1.27 | 4.91 ± 0.91 | 60.44 ± 1.04 | 11.39 ± 1.61 | 4.06 ± 1.24 |
| MMCE (Kumar et al., 2018) | 61.35 ± 1.07 | 12.93 ± 1.35 | 3.95 ± 1.11 | 61.28 ± 1.33 | 12.55 ± 1.40 | 3.90 ± 1.09 |
| FLSD (Mukhoti et al., 2020) | 60.94 ± 2.11 | 11.83 ± 1.04 | 4.04 ± 0.91 | 60.49 ± 1.52 | 13.81 ± 1.23 | 4.88 ± 0.99 |
| MbLS (Liu et al., 2021) | 62.30 ± 1.51 | 14.95 ± 1.44 | 6.50 ± 1.22 | 62.93 ± 1.95 | 13.66 ± 1.44 | 4.69 ± 1.04 |
| DCA (Liang et al., 2020) | 62.93 ± 1.04 | 10.17 ± 1.04 | 3.75 ± 0.92 | 63.15 ± 1.38 | 9.27 ± 0.94 | 3.55 ± 0.61 |
| NUCFL (DCA+COS) | 62.81 ± 0.91 | 9.29 ± 0.95 | 3.54 ± 0.67 | 63.24 ± 1.29 | 8.85 ± 1.11 | 3.51 ± 0.82 |
| NUCFL (DCA+L-CKA) | 62.94 ± 1.22 | 9.14 ± 1.22 | 3.52 ± 0.84 | 63.27 ± 1.30 | 8.52 ± 0.94 | 3.43 ± 0.64 |
| NUCFL (DCA+RBF-CKA) | 62.84 ± 1.15 | 9.21 ± 1.04 | 3.55 ± 0.92 | 63.17 ± 1.17 | 8.01 ± 1.08 | 3.30 ± 0.85 |
| MDCA (Hebbalaguppe et al., 2022b) | 62.84 ± 0.82 | 10.24 ± 0.85 | 3.72 ± 0.69 | 63.29 ± 1.06 | 10.00 ± 1.15 | 3.71 ± 0.74 |
| NUCFL (MDCA+COS) | 62.91 ± 1.04 | 9.33 ± 1.02 | 3.56 ± 0.91 | 63.14 ± 1.11 | 9.16 ± 1.24 | 3.58 ± 0.99 |
| NUCFL (MDCA+L-CKA) | 62.88 ± 0.95 | 9.19 ± 0.84 | 3.53 ± 0.62 | 63.14 ± 1.51 | 9.03 ± 1.48 | 3.51 ± 1.17 |
| NUCFL (MDCA+RBF-CKA) | 62.65 ± 1.12 | 9.24 ± 1.20 | 3.55 ± 0.94 | 63.22 ± 1.35 | 8.59 ± 1.27 | 3.47 ± 1.06 |

Table 20: Accuracy (%), calibration measures ECE (%), and SCE (%) of various federated optimization methods with different calibration methods under non-IID ($\alpha = 0.5$) scenario on the CIFAR-100 dataset. Values in boldface represent the best calibration provided by our method for the auxiliary calibration method, and underlined values indicate the best calibration across all methods.

| Calibration Method | FedAvg (McMahan et al., 2017) | | | FedProx (Sahu et al., 2018) | | | Scaffold (Karimireddy et al., 2019) | | |
|---|---|---|---|---|---|---|---|---|---|
| | Acc ↑ | ECE ↓ | SCE ↓ | Acc ↑ | ECE ↓ | SCE ↓ | Acc ↑ | ECE ↓ | SCE ↓ |
| Uncal. | 58.34 ± 1.66 | 12.02 ± 1.39 | 3.80 ± 0.97 | 58.88 ± 1.29 | 10.61 ± 1.44 | 3.63 ± 1.20 | 59.08 ± 1.02 | 13.92 ± 2.00 | 4.75 ± 1.33 |
| Focal (Lin et al., 2017) | 57.59 ± 1.13 | 14.28 ± 1.20 | 4.60 ± 1.22 | 58.07 ± 1.66 | 11.85 ± 1.28 | 4.19 ± 1.11 | 58.39 ± 1.36 | 14.68 ± 1.92 | 4.78 ± 1.49 |
| LS (Müller et al., 2019) | 56.35 ± 2.42 | 12.01 ± 1.54 | 3.81 ± 1.95 | 59.23 ± 1.72 | 14.77 ± 1.58 | 4.80 ± 1.75 | 59.15 ± 1.80 | 15.09 ± 2.13 | 5.09 ± 1.21 |
| BS (Brier, 1950) | 56.20 ± 2.19 | 11.72 ± 1.66 | 4.20 ± 1.24 | 56.43 ± 1.21 | 10.55 ± 1.57 | 3.60 ± 1.33 | 57.39 ± 1.51 | 12.32 ± 1.52 | 3.82 ± 1.24 |
| MMCE (Kumar et al., 2018) | 57.00 ± 1.34 | 11.81 ± 2.29 | 3.83 ± 1.00 | 57.11 ± 1.47 | 11.72 ± 1.44 | 4.17 ± 1.03 | 58.03 ± 1.44 | 11.77 ± 1.72 | 4.03 ± 1.21 |
| FLSD (Mukhoti et al., 2020) | 58.71 ± 1.85 | 11.91 ± 1.52 | 3.77 ± 1.14 | 56.28 ± 1.70 | 10.06 ± 1.57 | 3.89 ± 1.15 | 57.49 ± 1.08 | 14.01 ± 1.00 | 4.65 ± 0.92 |
| MbLS (Liu et al., 2021) | 58.62 ± 1.49 | 13.82 ± 1.82 | 4.75 ± 0.95 | 57.39 ± 0.82 | 11.77 ± 1.12 | 4.14 ± 1.03 | 59.38 ± 1.05 | 14.39 ± 1.15 | 4.72 ± 1.01 |
| DCA (Liang et al., 2020) | 59.19 ± 1.12 | 10.45 ± 0.87 | 3.61 ± 0.54 | 58.95 ± 1.15 | 8.57 ± 0.74 | 3.45 ± 0.45 | 59.10 ± 1.28 | 11.09 ± 0.83 | 3.83 ± 0.51 |
| NUCFL (DCA+COS) | 58.97 ± 1.25 | 9.35 ± 0.90 | 3.55 ± 0.65 | 59.11 ± 1.12 | 8.02 ± 0.88 | __3.41__ ± 0.60 | 59.13 ± 1.35 | 10.87 ± 0.79 | 3.68 ± 0.52 |
| NUCFL (DCA+L-CKA) | 60.73 ± 1.20 | __9.14__ ± 0.85 | __3.50__ ± 0.45 | 59.16 ± 1.32 | __7.99__ ± 0.71 | 3.42 ± 0.31 | 60.09 ± 1.29 | 10.95 ± 0.73 | 3.70 ± 0.64 |
| NUCFL (DCA+RBF-CKA) | 59.20 ± 1.30 | 10.16 ± 0.94 | 3.60 ± 0.61 | 59.95 ± 1.24 | 8.09 ± 0.42 | __3.41__ ± 0.10 | 60.05 ± 1.33 | __10.33__ ± 0.69 | __3.61__ ± 0.57 |
| MDCA (Hebbalaguppe et al., 2022b) | 59.47 ± 1.22 | 10.19 ± 1.14 | 3.60 ± 0.66 | 59.17 ± 1.15 | 8.61 ± 0.92 | 3.47 ± 0.59 | 60.85 ± 1.11 | 12.19 ± 1.03 | 4.15 ± 0.74 |
| NUCFL (MDCA+COS) | 59.24 ± 1.15 | 9.74 ± 0.95 | __3.59__ ± 0.69 | 58.78 ± 1.20 | __8.12__ ± 0.86 | 3.44 ± 0.57 | 60.07 ± 1.12 | 11.46 ± 1.01 | 4.06 ± 0.63 |
| NUCFL (MDCA+L-CKA) | 59.64 ± 1.44 | __9.70__ ± 1.31 | __3.59__ ± 1.07 | 59.69 ± 1.18 | 8.29 ± 0.92 | 3.45 ± 0.59 | 59.64 ± 1.15 | 10.51 ± 0.89 | __3.62__ ± 0.65 |
| NUCFL (MDCA+RBF-CKA) | 58.71 ± 1.14 | 10.09 ± 0.89 | 3.61 ± 0.60 | 58.81 ± 1.13 | 8.31 ± 0.91 | 3.45 ± 0.53 | 59.10 ± 1.00 | 10.64 ± 0.87 | 3.64 ± 0.68 |

| Calibration Method | FedDyn (Acar et al., 2021) | | | FedNova (Wang et al., 2020b) | | |
|---|---|---|---|---|---|---|
| | Acc ↑ | ECE ↓ | SCE ↓ | Acc ↑ | ECE ↓ | SCE ↓ |
| Uncal. | 59.39 ± 1.30 | 13.03 ± 1.30 | 5.04 ± 1.23 | 60.01 ± 1.33 | 13.95 ± 1.14 | 4.97 ± 0.92 |
| Focal (Lin et al., 2017) | 57.21 ± 1.27 | 13.62 ± 1.22 | 5.05 ± 1.07 | 58.15 ± 2.05 | 14.61 ± 1.53 | 5.19 ± 1.25 |
| LS (Müller et al., 2019) | 58.34 ± 1.67 | 15.59 ± 2.34 | 5.38 ± 1.72 | 60.05 ± 2.56 | 16.43 ± 1.31 | 5.69 ± 2.05 |
| BS (Brier, 1950) | 59.17 ± 1.63 | 14.21 ± 1.51 | 4.81 ± 1.02 | 57.44 ± 1.24 | 11.79 ± 1.86 | 4.19 ± 1.35 |
| MMCE (Kumar et al., 2018) | 58.35 ± 1.27 | 15.33 ± 1.58 | 5.35 ± 1.24 | 58.28 ± 1.55 | 12.95 ± 1.62 | 4.98 ± 1.06 |
| FLSD (Mukhoti et al., 2020) | 57.94 ± 2.31 | 12.23 ± 1.22 | 4.85 ± 1.01 | 60.49 ± 1.73 | 16.21 ± 1.41 | 5.64 ± 1.09 |
| MbLS (Liu et al., 2021) | 59.30 ± 1.71 | 15.35 ± 1.62 | 5.33 ± 1.30 | 59.93 ± 1.17 | 14.06 ± 1.84 | 4.75 ± 1.00 |
| DCA (Liang et al., 2020) | 60.77 ± 1.39 | 12.13 ± 0.99 | 4.81 ± 0.68 | 59.98 ± 1.41 | 11.23 ± 0.82 | 3.85 ± 0.53 |
| NUCFL (DCA+COS) | 59.48 ± 1.40 | 11.53 ± 0.94 | 3.85 ± 0.75 | 60.14 ± 1.35 | 10.62 ± 0.72 | 3.66 ± 0.59 |
| NUCFL (DCA+L-CKA) | 60.49 ± 1.33 | 11.29 ± 0.89 | 3.83 ± 0.58 | 60.33 ± 1.01 | __10.61__ ± 0.70 | __3.65__ ± 0.67 |
| NUCFL (DCA+RBF-CKA) | 59.23 ± 1.29 | __11.04__ ± 0.95 | __3.79__ ± 0.69 | 60.69 ± 1.39 | 10.65 ± 0.79 | __3.65__ ± 0.54 |
| MDCA (Hebbalaguppe et al., 2022b) | 59.61 ± 1.24 | 12.01 ± 1.12 | 4.39 ± 0.95 | 60.95 ± 1.17 | 10.58 ± 0.93 | 3.65 ± 0.66 |
| NUCFL (MDCA+COS) | 59.76 ± 1.01 | 10.93 ± 1.09 | 3.69 ± 0.81 | 60.93 ± 1.11 | 9.78 ± 0.88 | 3.63 ± 0.55 |
| NUCFL (MDCA+L-CKA) | 60.28 ± 1.29 | 10.60 ± 1.12 | 3.67 ± 0.92 | 59.92 ± 1.00 | __9.65__ ± 0.94 | __3.60__ ± 0.63 |
| NUCFL (MDCA+RBF-CKA) | 60.27 ± 1.17 | __9.95__ ± 1.10 | __3.63__ ± 0.84 | 59.98 ± 1.00 | 10.03 ± 0.87 | 3.63 ± 0.62 |

Table 21: Accuracy (%), calibration measures ECE (%), and SCE (%) of various federated optimization methods with different calibration methods under non-IID ($\alpha = 0.1$) scenario on the CIFAR-100 dataset. Underlined values indicate the best calibration across all methods.

| Calibration Method | FedAvg (McMahan et al., 2017) | | | FedProx (Sahu et al., 2018) | | | Scaffold (Karimireddy et al., 2019) | | |
|---|---|---|---|---|---|---|---|---|---|
| | Acc ↑ | ECE ↓ | SCE ↓ | Acc ↑ | ECE ↓ | SCE ↓ | Acc ↑ | ECE ↓ | SCE ↓ |
| Uncal. | 56.77 ± 1.42 | 14.80 ± 0.91 | 4.77 ± 0.51 | 57.95 ± 1.33 | 15.29 ± 0.74 | 5.11 ± 0.39 | 57.00 ± 1.37 | 13.88 ± 0.93 | 4.70 ± 0.52 |
| Focal (Lin et al., 2017) | 55.61 ± 1.18 | 15.96 ± 0.84 | 5.20 ± 0.56 | 58.00 ± 1.47 | 16.40 ± 0.78 | 5.58 ± 0.44 | 57.63 ± 1.35 | 16.92 ± 0.87 | 4.77 ± 5.61 |
| LS (Müller et al., 2019) | 56.26 ± 1.45 | 14.59 ± 0.98 | 4.71 ± 0.60 | 58.09 ± 1.52 | 17.08 ± 0.83 | 5.81 ± 0.52 | 57.90 ± 1.48 | 15.13 ± 0.74 | 5.11 ± 0.43 |
| BS (Brier, 1950) | 55.26 ± 1.56 | 13.98 ± 0.92 | 4.72 ± 0.49 | 57.72 ± 1.39 | 16.81 ± 0.85 | 5.75 ± 0.42 | 56.01 ± 1.11 | 15.59 ± 0.71 | 5.18 ± 0.51 |
| MMCE (Kumar et al., 2018) | 57.64 ± 1.29 | 16.66 ± 0.95 | 3.87 ± 0.57 | 57.64 ± 1.43 | 16.26 ± 0.77 | 5.53 ± 0.45 | 57.46 ± 1.34 | 17.77 ± 0.91 | 5.85 ± 0.61 |
| FLSD (Mukhoti et al., 2020) | 54.10 ± 1.35 | 11.29 ± 0.89 | 3.56 ± 0.51 | 59.23 ± 1.52 | 19.45 ± 0.99 | 6.50 ± 0.71 | 55.39 ± 1.42 | 10.84 ± 0.76 | 3.61 ± 0.54 |
| MbLS (Liu et al., 2021) | 55.16 ± 1.55 | 14.99 ± 1.08 | 4.80 ± 0.61 | 56.42 ± 1.48 | 15.94 ± 0.79 | 5.18 ± 0.51 | 55.11 ± 1.44 | 12.47 ± 0.82 | 4.21 ± 0.58 |
| DCA (Liang et al., 2020) | 56.19 ± 0.98 | 13.05 ± 0.42 | 3.99 ± 0.35 | 56.99 ± 1.10 | 13.17 ± 0.31 | 4.05 ± 0.29 | 58.10 ± 1.05 | 11.69 ± 0.34 | 3.85 ± 0.37 |
| NUCFL (DCA+COS) | 56.97 ± 1.05 | 11.95 ± 0.40 | 3.64 ± 0.33 | 57.64 ± 0.97 | 11.12 ± 0.28 | 3.53 ± 0.22 | 57.33 ± 1.02 | 10.97 ± 0.31 | 3.80 ± 0.29 |
| NUCFL (DCA+L-CKA) | 57.03 ± 1.21 | __10.24__ ± 0.42 | __3.52__ ± 0.11 | 57.96 ± 1.01 | __10.09__ ± 0.32 | __3.50__ ± 0.19 | 57.75 ± 1.08 | __9.05__ ± 0.29 | __3.28__ ± 0.28 |
| NUCFL (DCA+RBF-CKA) | 56.90 ± 1.11 | 10.76 ± 0.39 | 3.54 ± 0.34 | 58.05 ± 1.03 | 10.19 ± 0.92 | 3.52 ± 0.61 | 57.85 ± 1.00 | 10.94 ± 0.30 | 3.81 ± 0.26 |
| MDCA (Hebbalaguppe et al., 2022b) | 57.32 ± 1.01 | 12.79 ± 0.48 | 3.90 ± 0.39 | 57.77 ± 1.12 | 14.21 ± 0.31 | 4.38 ± 0.26 | 57.95 ± 1.07 | 10.99 ± 0.37 | 3.54 ± 0.40 |
| NUCFL (MDCA+COS) | 56.94 ± 1.23 | 11.34 ± 0.42 | 3.76 ± 0.32 | 57.58 ± 1.02 | 13.92 ± 0.30 | 4.11 ± 0.31 | 57.87 ± 1.09 | __9.06__ ± 0.32 | __3.30__ ± 0.27 |
| NUCFL (MDCA+L-CKA) | 57.44 ± 1.33 | __11.30__ ± 0.41 | 3.76 ± 0.29 | 57.49 ± 1.03 | __11.89__ ± 0.29 | __3.83__ ± 0.23 | 56.94 ± 1.11 | 10.11 ± 0.35 | 3.38 ± 0.31 |
| NUCFL (MDCA+RBF-CKA) | 56.51 ± 1.04 | 11.69 ± 0.46 | 3.82 ± 0.34 | 58.11 ± 1.06 | 14.07 ± 0.31 | 4.35 ± 0.24 | 57.90 ± 1.09 | 10.24 ± 0.33 | 3.39 ± 0.30 |

| Calibration Method | FedDyn (Acar et al., 2021) | | | FedNova (Wang et al., 2020b) | | |
|---|---|---|---|---|---|---|
| | Acc ↑ | ECE ↓ | SCE ↓ | Acc ↑ | ECE ↓ | SCE ↓ |
| Uncal. | 58.13 ± 1.40 | 14.26 ± 1.07 | 4.59 ± 0.76 | 58.43 ± 1.29 | 17.25 ± 0.82 | 6.14 ± 0.48 |
| Focal (Lin et al., 2017) | 57.64 ± 1.50 | 17.00 ± 1.05 | 4.80 ± 0.68 | 56.48 ± 1.42 | 14.81 ± 1.12 | 4.61 ± 0.74 |
| LS (Müller et al., 2019) | 57.65 ± 1.53 | 18.57 ± 1.09 | 6.44 ± 0.71 | 54.26 ± 1.49 | 16.83 ± 0.88 | 4.77 ± 0.59 |
| BS (Brier, 1950) | 57.83 ± 1.57 | 16.06 ± 1.18 | 5.23 ± 0.81 | 56.26 ± 1.20 | 16.77 ± 0.87 | 4.73 ± 0.32 |
| MMCE (Kumar et al., 2018) | 56.15 ± 1.46 | 10.27 ± 1.08 | 3.54 ± 0.40 | 56.99 ± 1.55 | 17.32 ± 0.80 | 6.18 ± 0.68 |
| FLSD (Mukhoti et al., 2020) | 56.95 ± 1.60 | 11.19 ± 0.96 | 3.55 ± 0.63 | 57.35 ± 1.51 | 17.83 ± 0.84 | 6.17 ± 0.99 |
| MbLS (Liu et al., 2021) | 56.58 ± 1.41 | 10.69 ± 1.06 | 3.59 ± 0.79 | 57.38 ± 1.48 | 16.66 ± 0.77 | 4.74 ± 0.59 |
| DCA (Liang et al., 2020) | 58.57 ± 1.02 | 12.53 ± 0.48 | 3.91 ± 0.45 | 58.68 ± 0.97 | 15.83 ± 0.39 | 5.21 ± 0.30 |
| NUCFL (DCA+COS) | 58.28 ± 1.09 | 11.13 ± 0.42 | 3.85 ± 0.36 | 58.64 ± 0.77 | 14.72 ± 0.37 | 4.59 ± 0.31 |
| NUCFL (DCA+L-CKA) | 58.19 ± 1.11 | 11.39 ± 0.45 | 3.87 ± 0.41 | 59.17 ± 1.07 | 14.21 ± 0.34 | 4.56 ± 0.29 |
| NUCFL (DCA+RBF-CKA) | 58.11 ± 1.09 | __10.90__ ± 0.44 | 3.81 ± 0.38 | 58.49 ± 1.06 | 14.25 ± 0.62 | 4.56 ± 0.44 |
| MDCA (Hebbalaguppe et al., 2022b) | 58.41 ± 1.09 | 11.61 ± 0.45 | 3.90 ± 0.49 | 58.75 ± 1.03 | 15.18 ± 0.33 | 5.09 ± 0.32 |
| NUCFL (MDCA+COS) | 59.00 ± 1.14 | 10.09 ± 0.48 | 3.48 ± 0.41 | 58.73 ± 1.08 | 14.38 ± 0.36 | 4.76 ± 0.30 |
| NUCFL (MDCA+L-CKA) | 58.78 ± 1.13 | 10.00 ± 0.46 | __3.47__ ± 0.45 | 58.69 ± 1.13 | 14.25 ± 0.38 | 4.57 ± 0.36 |
| NUCFL (MDCA+RBF-CKA) | 58.67 ± 1.10 | __9.95__ ± 0.43 | __3.47__ ± 0.37 | 58.84 ± 1.07 | __14.13__ ± 0.35 | __4.55__ ± 0.28 |

Table 22: Accuracy (%), calibration measures ECE (%), and SCE (%) of various federated optimization methods with different calibration methods under non-IID ($\alpha = 0.05$) scenario on the CIFAR-100 dataset. Underlined values indicate the best calibration across all methods.

| Dataset | Calibration Method | IID | | | Non-IID ($\alpha = 0.5$) | | | Non-IID ($\alpha = 0.1$) | | | Non-IID ($\alpha = 0.05$) | | |
|---|---|---|---|---|---|---|---|---|---|---|---|---|---|
| | | Acc ↑ | ECE ↓ | SCE ↓ | Acc ↑ | ECE ↓ | SCE ↓ | Acc ↑ | ECE ↓ | SCE ↓ | Acc ↑ | ECE ↓ | SCE ↓ |
| MNIST | UnCal. | 96.20 | 0.88 | 0.29 | 95.02 | 1.89 | 0.65 | 92.68 | 2.96 | 1.04 | 87.51 | 6.22 | 1.83 |
| | FedCal | 96.20 | 0.69 | 0.24 | 95.02 | 0.94 | 0.41 | 92.68 | 2.01 | 0.91 | 87.51 | 6.00 | 1.77 |
| | NUCFL(DCA+L-CKA) | **96.90** | **0.44** | **0.16** | **95.97** | **0.78** | **0.31** | 92.41 | **1.32** | **0.66** | **87.70** | **5.20** | **1.59** |
| FEMNIST | UnCal. | 92.48 | 4.10 | 1.66 | 90.95 | 4.17 | 1.66 | 88.91 | 5.58 | 2.83 | 86.98 | 8.78 | 3.49 |
| | FedCal | 92.48 | 3.92 | 1.66 | 90.95 | 4.02 | 1.62 | 88.91 | 5.01 | 2.62 | 86.98 | 8.00 | 3.42 |
| | NUCFL(DCA+L-CKA) | 92.39 | **3.49** | **1.46** | **91.63** | **3.52** | **1.47** | 89.89 | **4.49** | **1.77** | 86.89 | **7.84** | **3.27** |
| CIFAR-10 | UnCal. | 80.23 | 7.60 | 2.70 | 78.83 | 9.69 | 3.50 | 64.86 | 12.62 | 4.42 | 59.84 | 14.30 | 4.60 |
| | FedCal | 80.23 | 7.55 | 2.70 | 78.83 | 8.27 | 3.19 | 64.86 | 11.39 | 4.40 | 59.84 | 14.07 | 4.58 |
| | NUCFL(DCA+L-CKA) | **81.05** | **5.77** | **1.75** | 78.66 | **7.06** | **2.60** | 64.77 | **8.43** | **3.12** | **60.30** | **10.62** | **3.92** |
| CIFAR-100 | UnCal. | 65.19 | 7.61 | 3.25 | 61.34 | 10.52 | 3.61 | 58.34 | 12.02 | 3.80 | 56.77 | 14.80 | 4.77 |
| | FedCal | 65.19 | 7.01 | 3.20 | 61.34 | 8.80 | 3.49 | 58.34 | 12.00 | 3.80 | 56.77 | 13.39 | 4.56 |
| | NUCFL(DCA+L-CKA) | **65.43** | **5.98** | **2.99** | **62.05** | **6.14** | **3.07** | **60.73** | **9.14** | **3.50** | **57.03** | **10.24** | **3.52** |

Table 23: Comparison with SOTA baseline (FedCal) on different datasets using FedAvg under different data distribution. The average ACC and calibration error are reported. Our method outperforms the baseline in terms of calibration error and sometimes in average accuracy as well.

## B.3 MITIGATING OVER/UNDER-CONFIDENCE

In addition to the over-confidence examples discussed in Section 5.2, we also observe cases of under-confidence in FL algorithms. Figure 6 and 7 presents the reliability histograms for non-IID ($\alpha = 0.5$) FedAvg using the MNIST and FEMNIST datasets, where the results show a gap that exceeds the diagonal line, indicating under-confidence issues. While over-confidence is often viewed as more problematic in deep neural networks for real-world applications, under-confidence can also pose challenges in specific scenarios. For example, in a medical diagnosis scenario, a threshold of 0.95 for negative predictions means that any instance classified as negative with a probability $\leq$ 0.95 requires manual review. An under-confident model may classify an instance as negative with a probability of 0.51, suggesting a true probability of correctness that is at least 0.51, yet this does not clarify whether it surpasses the threshold. This leads to compulsory manual inspection regardless of the actual correctness, thus increasing the workload. Among all the figures, we see that our proposed method effectively calibrates FL and exhibits a smaller gap (red region), demonstrating improved performance over other approaches.

## B.4 COMPARISON WITH FEDCAL

This section presents additional experimental results and comparisons with FedCal (Peng et al., 2024) from Section 5.2. Table 23 provides complete comparisons across different datasets. The results in Table 23 indicate that our proposed NUCFL outperforms FedCal. FedCal improves local client calibration using a local scaler design and aggregates these scalers to create a global scaler for the global model. However, relying solely on an aggregated scaler from local clients can lower global ECE but may neglect crucial interactions between global and local calibration needs. Our method accounts for these interactions, mitigating biases toward local heterogeneity and more effectively meeting global calibration requirements.

## B.5 MORE DIFFICULT SCENARIO

In addition to evaluating our method in a cross-silo scenario where all clients participate simultaneously, we also assess its performance in a cross-device scenario with partial participation. To accomplish this, we divided the CIFAR-100 training set among $M = 100$ clients, each with a distinct data distribution. We then implemented FedAvg using ResNet-34, setting the client participation ratio to 10% for each round. This experiment aims to provide deeper insights into how our method performs under more practical conditions, where participation rates are typically low. Table 24 shows that in this practical scenario, our NUCFL continues to demonstrate its benefits. While the global model may not fully represent the global distribution due to the bias toward clients participating in the previous round, our approach can effectively minimize calibration bias towards current participants.

## B.6 TRADE-OFF BETWEEN ACCURACY AND RELIABILITY

During this study, we observed two types of trade-offs between accuracy and reliability: the Calibration Method Trade-Off and the FL Algorithm Trade-Off.

**Calibration Method Trade-Off.** In a centralized setup, there are two types of calibration methods, train-time calibration and post-hoc calibration. Post-hoc calibration, which is often used for

| Calibration Method | IID | | | Non-IID ($\alpha = 0.5$) | | | Non-IID ($\alpha = 0.1$) | | | Non-IID ($\alpha = 0.05$) | | |
|---|---|---|---|---|---|---|---|---|---|---|---|---|
| | Acc ↑ | ECE ↓ | SCE ↓ | Acc ↑ | ECE ↓ | SCE ↓ | Acc ↑ | ECE ↓ | SCE ↓ | Acc ↑ | ECE ↓ | SCE ↓ |
| UnCal. | 57.33 | 10.21 | 4.39 | 54.99 | 11.49 | 4.46 | 51.34 | 14.02 | 4.70 | 49.33 | 18.80 | 5.97 |
| DCA | 57.61 | 8.21 | 3.00 | 55.10 | 9.28 | 3.41 | 51.24 | 11.02 | 3.61 | 49.29 | 14.13 | 4.76 |
| MDCA | 57.49 | 9.40 | 3.52 | 55.23 | 10.33 | 3.64 | 51.63 | 12.39 | 3.82 | 49.61 | 14.66 | 4.81 |
| FedCal | 57.33 | 9.33 | 3.49 | 54.99 | 9.49 | 3.46 | 51.34 | 10.93 | 3.60 | 49.33 | 13.91 | 4.74 |
| NUCFL(DCA+L-CKA) | **57.55** | **8.00** | **2.96** | **55.23** | **9.08** | **3.37** | **51.93** | 10.67 | 3.57 | 49.29 | **12.30** | **3.86** |
| NUCFL(MDCA+L-CKA) | 57.21 | 8.47 | 3.17 | **55.39** | 9.44 | 3.46 | 51.49 | 10.02 | 3.51 | **49.69** | 12.79 | 4.06 |

Table 24: Comparison under a partial participants scenario (10% clients in each round) using FedAvg on the CIFAR-100 dataset, where representative calibration methods are selected for comparison. Improved calibration error shows that our method significantly enhances model calibration.

| Calibration Method | FedSpeed (Sun et al., 2023) | | | FedSAM (Qu et al., 2022) | | | FedMR (Hu et al., 2023) | | | FedCross (Hu et al., 2022) | | |
|---|---|---|---|---|---|---|---|---|---|---|---|---|
| | Acc ↑ | ECE ↓ | SCE ↓ | Acc ↑ | ECE ↓ | SCE ↓ | Acc ↑ | ECE ↓ | SCE ↓ | Acc ↑ | ECE ↓ | SCE ↓ |
| UnCal. | 65.10 | 15.33 | 7.11 | 64.23 | 13.95 | 6.92 | 64.95 | 13.18 | 6.70 | 65.30 | 15.81 | 7.19 |
| FedCal | 65.10 | 12.95 | 6.53 | 64.23 | 12.08 | 6.48 | 64.95 | 12.85 | 6.49 | 65.30 | 14.39 | 6.98 |
| NUCFL(DCA+L-CKA) | 65.17 | **11.15** | **5.92** | 64.52 | **10.22** | **5.55** | 64.95 | **12.11** | **6.47** | 65.33 | **13.01** | **6.53** |

Table 25: Comparison using other FL optimization methods shows that our method can adapt to any FL algorithm and improve calibration error.

centralized models, adjusts the calibration error without impacting accuracy. However, as shown in Table 6, when we examine train-time calibration methods in a centralized context, we observe that in over 25% of cases, these methods improve calibration at the cost of reduced accuracy, unlike the stable post-hoc methods. But given that train-time methods are more practical for FL due to the lack of post-hoc dataset, we take this concern into consideration. We found that existing train-time calibration methods tend to degrade accuracy in FL settings, although they vary in their effectiveness at reducing calibration error. This observation further guided our goal to design a calibration method that could improve reliability without compromising accuracy.

**FL Algorithm Trade-Off.** Another trade-off we identified is between accuracy and reliability across different FL algorithms. While advanced FL algorithms, such as FedDyn and FedNova, can achieve higher accuracy than naive FedAvg, they also tend to introduce greater calibration error. For example, in Table 1, the accuracy (ACC) for FedAvg is 61.34, while FedDyn and FedNova reach 62.39 and 63.01, respectively. However, their calibration error also increases from 10.52 in FedAvg to 12.53 in FedDyn and 11.45 in FedNova. With the help of our method applied to FedDyn and FedNova, we improve their accuracies to 62.84 and 63.17, respectively, while significantly reducing their calibration error to 10.24 and 8.01—both lower than the original FedAvg calibration error of 10.52. This trend is also found in other datasets and data distributions, as shown in our result tables in Appendix B.2. This shows our method's effectiveness in improving reliability without sacrificing accuracy across various FL algorithms.

## B.7 COMPARISON WITH FL OPTIMIZATION METHODS USING "CALIBRATION"

While searching for related work, we found that some FL studies Zhang et al. (2022); Luo et al. (2021) use the term "calibration." Although these works do not focus on "model calibration" as discussed in this paper, we consider them advanced FL optimization methods and evaluate their performance in terms of model calibration. Specifically, we evaluate both the uncalibrated versions of FedLC Zhang et al. (2022) (with $\tau = 1$) and CCVR Luo et al. (2021) (on FedAvg) as well as their calibrated counterparts by integrating our proposed NUCFL. We follow the same experimental setup as described in Table 1, utilizing the CIFAR-100 dataset under a non-IID ($\alpha = 0.5$) data distribution scenario. The results are provided in Table 27.

Comparing uncalibrated CCVR and FedLC with FedAvg, we observe an increase in accuracy alongside a rise in calibration error. This indicates that these algorithms are advanced FL optimization methods designed primarily to improve accuracy rather than address model calibration. Since we view CCVR and FedLC as advanced FL optimization methods, we can apply NUCFL to them.

| Calibration Method | local epoch $E = 1$ | | | local epoch $E = 5$ (default) | | | local epoch $E = 10$ | | |
|---|---|---|---|---|---|---|---|---|---|
| | Acc ↑ | ECE ↓ | SCE ↓ | Acc ↑ | ECE ↓ | SCE ↓ | Acc ↑ | ECE ↓ | SCE ↓ |
| UnCal. | 59.23 | 8.92 | 3.47 | 61.34 | 10.52 | 3.61 | 62.00 | 11.39 | 3.98 |
| NUCFL(DCA+L-CKA) | **59.47** | **6.40** | **3.29** | **62.05** | **6.14** | **3.07** | **62.05** | **6.08** | **3.00** |

Table 26: Calibration performance with different local epochs using FedAvg under non-IID conditions ($\alpha = 0.5$ on CIFAR-100 dataset).

| Calibration Method | FedAvg | | | CCVR | | | FedLC | | |
|---|---|---|---|---|---|---|---|---|---|
| | Acc ↑ | ECE ↓ | SCE ↓ | Acc ↑ | ECE ↓ | SCE ↓ | Acc ↑ | ECE ↓ | SCE ↓ |
| UnCal. | 61.34 | 10.52 | 3.61 | 63.01 | 13.69 | 4.99 | 62.73 | 14.58 | 6.11 |
| NUCFL(DCA+L-CKA) | **62.05** | **6.14** | **3.07** | **63.05** | **9.28** | **3.53** | **62.77** | **10.98** | **3.79** |

Table 27: Calibration performance of FL algorithms incorporating "calibration."

Comparing the original versions with their NUCFL-calibrated counterparts, we observe improvements in both accuracy and calibration error. This demonstrates the adaptability of our method and its compatibility with any FL algorithm.

---

**Algorithm 2** Applying NUCFL to FL

---

1: **Input:** global model $w$, local model $w_m$ for client $m$, local epochs $E$, and rounds $T$.
2: **for** each round $t = 1, 2, ..., T$ **do**
3:     Server sends $w^{(t-1)}$ to all clients.
4:     **for** each client $m \in M$ **do**
5:       Initialize local model $w_m^{(t,0)} \leftarrow w^{(t-1)}$
6:       **for** each epoch $e = 1, 2, ..., E$ **do**
7:         Get the gradient for each client: $\delta_m^{(t,e)} = w^{(t-1)} - w_m^{(t,e)}$
8:         Calculate the similarity with the previous accumulated gradient: $\beta_m = \text{sim}(\delta^{(t-1)}, \delta_m^{(t,e)})$,
9:         Decide calibration loss function $\ell_{cal}$ based on DCA (6) or MDCA (7).
10:        Each client obtains calibrated local loss $\mathcal{L}_m^{cal}(w_m^{(t,e)})$ based on (5).
11:        $w_m^{(t,e)} \leftarrow \text{ClientOPT}(w_m^{(t,e-1)}, \mathcal{L}_m^{cal}(w_m^{(t,e)}))$
12:       **end for**
13:       $w_m^{(t,E)}$ denotes the result after performing $E$ epochs of local updates.
14:       Client sends $\delta_m^{(t)} = w^{(t-1)} - w_m^{(t,E)}$ to the server after local training.
15:     **end for**
16:     Server computes aggregate update $\delta^{(t)} = \sum_{m \in M} \frac{|\mathcal{D}_m|}{|\mathcal{D}|} \delta_m^{(t)}$
17:     Server updates global model $w^{(t)} \leftarrow \text{ServerOPT}(w^{(t-1)}, \delta^{(t)})$
18: **end for**

---

