# OpenReview forum: "Unlocking the Potential of Model Calibration in Federated Learning"
_ICLR.cc/2025/Conference — ICLR 2025 Poster_

### Official Review · Reviewer_zKQ7 · 2024-11-01

**Soundness:** 3
**Presentation:** 3
**Contribution:** 2
**Rating:** 6
**Confidence:** 4

**Summary:**

This paper contributes to a well-established line of research that adapts standard machine learning (ML) techniques for federated settings by leveraging the similarity between global and local models to weight client contributions. Here, the ML technique under adaptation is model calibration, achieved through an auxiliary loss function. The paper’s primary contribution is a proposal to weight this auxiliary calibration loss based on a customizable similarity metric between global and local models, such as cosine similarity or other advanced measures. In this context, the “local” model refers to the locally trained replica of the global model. The authors provide an extensive empirical evaluation on standard federated learning datasets for computer vision, using LDA to generate non-IID partitions for IID datasets and some naturally heterogeneous datasets as well (FEMNIST). Their results indicate improved performance over baseline methods across all datasets, models, and levels of data heterogeneity.

**Strengths:**

- Comprehensive Evaluation: Experiments are extensive and demonstrate consistent improvements across datasets, models, and non-IID settings.
- Writing and Presentation: The paper is well-written, with clear diagrams and formatting that aid readability.
- Practical Relevance: The method holds strong practical implications, bringing effective calibration methods to FL.

**Weaknesses:**

I must mention that I am much more familiar with FL than model calibration and thus focus on this side of the work.

- Limited Novelty in Core Insight: Although the similarity-based weighting approach is practical, it is not particularly novel, e.g.,[1,2,3]. The authors do acknowledge the influence of prior FL works that leverage model similarity metrics.
- Potential Sensitivity to Hyperparameters: The method might be sensitive to local training hyperparameters like learning rate and epoch count, which could impact the similarity measure’s effectiveness. In cases closer to FedSGD with only one or a few steps performed, data heterogeneity may minimally impact the model replica, potentially diminishing the relevance of the similarity function. Similarly, scenarios involving extensive local training or very high learning rates may affect the similarity measure’s utility as all models may diverge very far from the global model regardless of the degree of local data heterogeneity.
- No Exploration of Alternative Similarity References: Using only the client’s local model replica as the similarity reference could limit the applicability; other approaches (e.g., auxiliary persistent local models or specific personalized layers) might better capture client-specific characteristics.




[1] Tian Li, Shengyuan Hu, Ahmad Beirami, Virginia Smith:
Ditto: Fair and Robust Federated Learning Through Personalization. ICML 2021: 6357-6368

[2] Tao Yu, Eugene Bagdasaryan, Vitaly Shmatikov:
Salvaging Federated Learning by Local Adaptation. CoRR abs/2002.04758 (2020)

[3] Yutao Huang, Lingyang Chu, Zirui Zhou, Lanjun Wang, Jiangchuan Liu, Jian Pei, Yong Zhang:
Personalized Cross-Silo Federated Learning on Non-IID Data. AAAI 2021: 7865-7873

**Questions:**

1. Can the authors provide any insight on how would the proposed approach perform under different local training conditions? For example with minimal local training (≤ 1 epoch or with very low learning rates) or extensive local training (e.g., training the local model to convergence or near convergence every round).
2. Have the authors considered the use of auxiliary local models or personalized layers, to compute similarity?
3. Why did the authors decide to use raw similarity/relevance values rather than a softmaxed version with a normalization factor constructed based on the entire distribution of client similarities?

---

> ### Author Response · Authors · 2024-11-18
> **Response to Reviewer zKQ7 (1/3)**
>
> ## 1. Response to Weakness 1: Novelty in core insight
>
> Thank you for your feedback. We would like to emphasize the novelty of our work lies in three key contributions:
> - Addressing an overlooked but critical aspect—model calibration in FL—which has not been sufficiently explored in prior research.
> - Considering the global and local relationships using similarity measurement effectively captures data heterogeneity and adjusts calibration needs.
> - Our proposed framework is adaptable to various similarity measurements on any FL algorithm, promoting model calibration across different scenarios.
> While our approach leverages model similarity, similar to some previous FL studies, this choice is both logical and effective, as model similarity has consistently demonstrated soundness and stability in FL settings. The similarity-based approach is a practical strategy for capturing the relationship between the global and local models, which is crucial in FL.
>
> We would like to clearly differentiate our contributions from those of existing works [1, 2, 3]. Unlike these studies, which do not consider model calibration, we leverage similarity scores to adjust the model calibration penalty, such as “Difference between Confidence and Accuracy (DCA)” or “Multi-Class DCA (MDCA).” Clients with high similarity to the global model receive a larger calibration penalty, as their local models, closely resembling the global model, are likely to capture global characteristics well. This allows room for improving calibration error with minimal accuracy reduction. Conversely, clients with low similarity are assigned a smaller calibration penalty, as their dissimilar or heterogeneous local models may benefit more from prioritizing the original local objective (e.g., improving accuracy) rather than enforcing strict global alignment. These new insights are new to our work and have not been considered in [1,2,3] that use similarity for different purposes.
>
> We thank the reviewer for this thoughtful comments and have added this discussion and the references in to **Section 4.2** of our updated manuscript.
>
> > [1] Ditto: Fair and Robust Federated Learning Through Personalization, ICML ‘21
> >
> > [2] Salvaging Federated Learning by Local Adaptation, 2020
> >
> > [3] Personalized Cross-Silo Federated Learning on Non-IID Data. AAAI ‘21

---

> ### Author Response · Authors · 2024-11-18
> **Response to Reviewer zKQ7 (2/3)**
>
> ## 2. Response to Weakness 2 and Question 1:  local epoch
>
> Thank you for this comment. In our experiments, we set each client to perform 5 local epochs per round, which is a commonly used configuration and aligns with the setup reported in the baselines for the considered FL algorithms. In response to your suggestion, we conducted an additional ablation study to examine the effect of varying local epochs on performance. Using the same configurations as in Table 1, we ran FedAvg with both reduced and increased local epochs, with results shown in the table below.
>
> ||FedAvg: 1 epoch | 5 epochs (default) | 10 epochs
> |---|---|---|---|
> || ACC, ECE | ACC, ECE| ACC, ECE
> Uncal.|  59.23, 8.92  | 61.34, 10.52 | 62.00  11.39
> NUCFL(DCA + L-CKA) | 59.47,  6.40 | 62.05, 6.14 | 62.14  6.08
>
> With fewer local training epochs, we observed an accuracy drop for uncalibrated FedAvg, but NUCFL still reduced calibration error. We believe that in cases with very few local training steps, the local model has limited ability to capture the client’s unique data distribution, remaining similar to the received global model even after local training. This may limit the effectiveness of similarity measurement in NUCFL, as the local and global models become too similar to meaningfully capture their difference. Consequently, [1-epoch FedAvg + NUCFL] does introduce slightly higher error than [5-epoch FedAvg + NUCFL], although it still demonstrates improvement over [uncalibrated 1-epoch FedAvg]. While reducing the number of local epochs diminishes the model's ability to capture client-specific characteristics, a uniform local training setup across clients can still reflect local distribution, albeit less strongly. Though our method is slightly less effective with lower local epoch numbers, we would like to emphasize that in practical FL scenarios, the number of local SGD steps is often large. This is because fewer SGD steps before global aggregation result in more frequent communications of model parameters to the server to achieve favorable performance, which is resource-intensive and generally avoided.
>
> Conversely, when local training is more extensive, the similarity measurement in NUCFL may more accurately capture the relationship between client and server, as each local model better reflects the client’s unique data distribution. This enhanced similarity measurement can offer advantages for non-uniform calibration. For example, we observed slightly better ECE for [10-epoch FedAvg + NUCFL] compared to [5-epoch FedAvg + NUCFL]. Although the calibration error may vary slightly across FL algorithms with different local epoch settings, our method consistently shows improvement in calibration compared to uncalibrated FL with the same epoch configurations.
>
> We thank the reviewer for this comment and have included this comparison in **Appendix B.2 and Table 26**.

---

> ### Author Response · Authors · 2024-11-18
> **Response to Reviewer zKQ7 (3/3)**
>
> ## 3. Response to Weakness 3 and Question 2: Alternative Similarity References
>
> Thank you for this thoughtful comment. Our goal is to capture the similarity between clients and the server to determine appropriate calibration adjustments. We believe that using the full view of each client’s local model replica for similarity computation with the global model aligns with most FL work, as global servers and local clients generally share the same model structure.
>
> In this work, we focus on model calibration for general FL, which is optimized to produce a better global model rather than personalized models. While personalized layers can indeed enhance FL’s personalization capabilities, they may introduce challenges when determining similarity. For example, as seen in Fig. 1 of [1], each client maintains a base model structure shared with the server along with a personalized layer. However, because this personalized layer has no equivalent counterpart on the server, calculating similarity between the server and client models becomes problematic.
>
> Given the commonly aligned model structure between servers and clients in most FL literature, our approach focuses on the client’s full model replica to compute the similarity needed for calibration adjustments. This ensures that our method is generalizable and practical within standard FL frameworks. Future work could explore model calibration for personalized FL settings, where personalized layers might be incorporated.
>
> We thank the reviewer for this insightful comment and have added this as the future work in the **Section 6** of our updated manuscript.
>
> > [1] Federated Learning with Personalization Layers, Arivazhagan et al.
>
>
>
> ## 4. Response to Question 3: raw similarity/relevance values
>
> We thank the reviewer for this insightful question. Using raw similarity/relevance values was a design choice aimed at directly leveraging the unnormalized relationship between the local and global models. It can also maintain the inherent characteristics of different similarity measurements. This approach ensures that the similarity values reflect the actual relationships between local and global models without being influenced by distributional normalization. However, we agree that normalization using a softmax function, which considers the entire distribution of client similarities, is also a valid approach. This is an interesting direction that we will explore further in future work.
>
>
>
> Again, we appreciate the reviewer for the helpful comments. We are happy to answer any additional questions you may have.

---

> ### Comment · Reviewer_zKQ7 · 2024-11-20
>
> I thank the authors for their detailed response, hard work, and the new results that address my first question. I also appreciate the thoughtful reasoning you gave me about my other questions.
>
> My primary concern with this work is that the proposed method—weighting the auxiliary local loss by the similarity between the local replica and the server model—could, in principle, be applied to many reasonable auxiliary loss terms from the centralized ML literature, with a similar justification as provided here for the model calibration loss. If considered novel, this approach could potentially result in as many publications as there are auxiliary loss terms unexplored in the FL literature.
>
> Can the authors justify why their similarity-weighting method is **particularly** well-suited for the model calibration loss rather than only a generic approach that would generalize to many more loss functions?

---

> > ### Author Response · Authors · 2024-11-24
> > **Response to Reviewer zKQ7**
> >
> > We thank the reviewer for this insightful question. We have given this a lot of thought over the past few days. We will first provide more detailed insights into our approach, and then discuss other loss functions.
> >
> > 1. To the best of our knowledge, our work is the first to apply the concept of similarity to model calibration in FL. We show that incorporating our similarity-weighting mechanism into the local calibration loss can significantly enhance calibration performance while maintaining high accuracy. Choosing too large of a weight for the auxiliary loss may lead to compromised accuracy, while choosing too small of a weight for the auxiliary loss may lead to large expected calibration error (ECE), making the choice of an appropriate weight a non-trivial problem. One of our key contributions is to solve this problem by imposing a different weight to individual client's auxiliary loss dynamically according to their data heterogeneity, which dictates whether they should focus on optimizing accuracy in the current round or if they can prioritize calibration. Specifically, during local training, when a client observes a large similarity between its local gradient and the previous global gradient, the auxiliary loss is promoted, while a large dissimilarity (i.e., high diversity) will retain focus on accuracy. As demonstrated in our ablation study (Table 3), without this dynamic similarity-based adjustment, performance degradation occurs, corroborating the beneficial role this approach plays in achieving joint accuracy and calibration outcomes in FL.
> >
> > 2. More generally, to the best of our knowledge, our work is the first to dynamically adapt weights of auxiliary loss functions based on gradient similarity metrics in FL. This potential can provide positive impact to the community where other metrics are becoming increasingly important. Below, we discuss other auxiliary losses that have been investigated in FL, and explain potential benefits of integrating our approach into these works. **Overall, we agree with the reviewer that if others can directly apply our method to enhance other metrics, those works would not be a novel publication as they become incremental over our approach.** However, considering that (i) our work is the first to take this approach, and (ii) has the potential for improving other metrics, we hope the reviewer can view this potential generalizability (albeit requiring further study) as a positive aspect of our approach for advancing the field of FL.
> >
> > 3. In the first paper [1] that the reviewer suggested, the authors incorporate a regularization term that encourages the personalized models to get close to the optimal global model, aiming to enhance accuracy via personalization while offering fairness and robustness benefits. The weight $\lambda$ for the regularization term is a hyperparameter, and each client $\lambda$ is determined locally based on its local validation data. Specifically, if a client has more than 5 validation data points, $\lambda$ is chosen from {0.05, 0.1, 0.2} for strong attacks, while $\lambda$ selected from {0.1, 1, 2} for all other attacks. This necessitates selecting appropriate hyperparameter candidates and entails a significant communication burden for hyperparameter tuning in a federated learning scenario. Our intuition is that by using our approach to dynamically design the weight based on the similarity between the local and global gradients, a smaller weight $\lambda$ can be assigned in cases of high similarity to focus more on personalization. This is because a high similarity indicates that the local model already shares similar characteristics with the global model, making strong regularization unnecessary. Although a comprehensive study is needed, this approach has the potential to achieve a better balance between personalization accuracy, fairness, and robustness, all without requiring additional hyperparameter tuning. This is a significant advantage in federated learning, where hyperparameter tuning is particularly challenging due to the high communication costs associated with re-running the federated learning process. In [2] and [3] (papers suggested by the reviewer), the authors also introduce the weight parameter for the auxiliary term (targeting local adaptation via multi-task learning or personalization), introducing an additional hyperparameter that requires tuning without clear insights. Once again, our approach has the potential to address these issues, although further exploration is required to ensure seamless integration.
> >
> > For the reasons outlined above, while an additional paper that is based on our approach (e.g., directly applying this idea to other metrics) may not qualify as a novel publication, we believe that the potential generalizability of our idea can advance the field of federated learning. We hope the reviewer can consider it from this perspective.
> >
> > Best, Authors of Paper 7239

---

> ### Comment · Reviewer_zKQ7 · 2024-11-24
>
> I thank the authors for their continued engagement.
>
> I agree that, to the best of my knowledge, your work is the first to explicitly weight an auxiliary loss based on similarity. My initial concerns about the novelty of the approach stemmed from its potentially incremental advancement over prior works that directly incorporated model similarity metrics as terms in the local objective to constrain model divergence (e.g., [4]). However, the authors’ recent comments have highlighted an advantage that was not initially apparent to me: while incorporating such model divergence terms directly can be challenging due to their potentially different scale compared to the primary loss (requiring hyperparameters to tune), this work circumvents that issue by leveraging an auxiliary loss that is already well-behaved and designed for compatibility.
>
> Given the generic nature of the proposed method, I recommend broadening the paper's scope by investigating similarity-based weighting for multiple auxiliary losses in many domains. That said, the additional experiments and arguments presented have convinced me that the proposed technique holds value for the community; even if its exploration is currently limited to a specific application domain, I am thus inclined to accept the work and raise my score.
>
> [4] Li, et al., "Federated Optimization in Heterogeneous Networks"

---

> > ### Author Response · Authors · 2024-11-26
> >
> > We sincerely thank Reviewer zKQ7 for raising the score to positive! We agree that investigating similarity-based weighting for other auxiliary losses is a valuable direction, and we have included this as part of the future work in **Section 6** of the revised paper. We appreciate your constructive feedback and thoughtful suggestions.

---

### Official Review · Reviewer_KxQB · 2024-11-03

**Soundness:** 3
**Presentation:** 2
**Contribution:** 2
**Rating:** 5
**Confidence:** 2

**Summary:**

This paper aims to ensure the reliability of FL in real-world applications. It proposes Non-Uniform Calibration for Federated Learning (NUCFL) that integrates FL with model calibration to achieve the goal. NUCFL dynamically adjusts the model calibration objectives by measuring the relationships between the local and the global model. Experiments show the superiority of NUCFL compared with existing FL methods.

**Strengths:**

[1] The paper takes an important step to explore FL from the perspective of model calibration, which is important to various decision-making scenarios.

[2] Extensive experiments are carried out to show that NUCFL can seamlessly integrate with existing FL methods and improve the original performance of those methods.

**Weaknesses:**

[1] It is unclear whether there is a trade-off between accuracy and reliability. If so, please give a detailed illustration of this aspect. If not, the advantage of not sacrificing accuracy seems an irrelevant contribution.

[2] The relation between model calibration and enhancing the confidence of FL outputs is weak. It requires further demonstration and analysis of why model calibration can realize that.

[3] It is better to provide a further investigation into how existing centralized calibration methods can adapt to the FL setting, for example, simply combining them with FedAvg framework. Empirical studies can also involve this part.

**Questions:**

N/A

---

> ### Author Response · Authors · 2024-11-17
> **Response to Reviewer KxQB (1/2)**
>
> ## 1. Response to Weakness 1: trade-off between accuracy and reliability
>
> Thank you for this insightful question. We found two types of trade-offs between accuracy and reliability in our study:
>
> **(a) Calibration Method Trade-Off**: In a centralized setup, there are two types of calibration methods—train-time calibration and post-hoc calibration. Post-hoc calibration, which is often used for centralized models, adjusts the calibration error without impacting accuracy. However, as shown in Table 6, when we examine train-time calibration methods in a centralized context, we observe that in over 25% of cases, these methods improve calibration at the cost of reduced accuracy, unlike the stable post-hoc methods. But given that train-time methods are more practical for FL due to the lack of post-hoc dataset, we take this concern into consideration. We found that existing train-time calibration methods tend to degrade accuracy in FL settings, although they vary in their effectiveness at reducing calibration error. This observation further guided our goal to design a calibration method that could improve reliability without compromising accuracy.
>
> **(b) FL Algorithm Trade-Off**: Another trade-off we identified is between accuracy and reliability across different FL algorithms. While advanced FL algorithms, such as FedDyn and FedNova, can achieve higher accuracy than naive FedAvg, they also tend to introduce greater calibration error. For example, in Table 1, the accuracy (ACC) for FedAvg is 61.34, while FedDyn and FedNova reach 62.39 and 63.01, respectively. However, their calibration error also increases from 10.52 in FedAvg to 12.53 in FedDyn and 11.45 in FedNova. With the help of our method applied to FedDyn and FedNova, we improve their accuracies to 62.84 and 63.17, respectively, while significantly reducing their calibration error to 10.24 and 8.01—both lower than the original FedAvg calibration error of 10.52. This shows our method’s effectiveness in improving reliability without sacrificing accuracy across various FL algorithms.
>
> Overall, although the accuracy and expected calibration error (ECE) are not directly competing, the accuracy tends to get decreased if we solely focus on ECE. For example, if we only consider the ECE loss, the model can be trained in a way that both the confidence and the accuracy are low e.g., 0.3 (which can be viewed as a reliable model because the accuracy aligns well with the confidence, but the accuracy itself is low). Hence, maintaining the accuracy while lowering ECE is one of the primary goals in the model calibration community.
>
> As the first work to consider train-time model calibration for FL, we intend to keep pursuing this goal in future research, as train-time calibration has shown potential to affect the performance/accuracy of FL algorithms. We thank the reviewer for this insightful comment and have added this discussion to our updated manuscript (**Section 5.2 and Appendix B.6**).
>
>
> ## 2. Response to Weakness 2: relation between model calibration and enhancing the confidence
>
> We appreciate the opportunity to clarify these terms and explain their relationship in more detail.
>
> - Model Calibration: The process involves aligning the model’s confidence (predicted probability) with its actual accuracy, minimizing the gap between them.
>
> - Confidence: This refers to the predicted probability assigned by the model to a specific class given an input. High confidence with low accuracy indicates overconfidence, while low confidence with high accuracy indicates underconfidence.
>
> Thus, "enhancing or reducing confidence" refers to adjusting the model’s predicted probabilities, whereas model calibration aims to achieve "reliable" confidence by minimizing the gap between predicted confidence and actual accuracy (in line 45 in the motivation of the manuscript). That is, it is important to note that "increasing confidence" does not necessarily indicate "better model calibration." A model with excessively high confidence can produce overconfident predictions, potentially misleading users to over-rely on the model even when its predictions are incorrect. Model calibration addresses this challenge by ensuring that the model's confidence accurately reflects the probability of being correct, improving reliability and user trust in the model's outputs. This performance is captured by expected calibration error (ECE). Our experimental results show that the proposed scheme leads to enhanced ECE in federated learning settings, thus leading to better calibration.
>
> More detail can be found in Section 3.2. We thank the reviewer for this comment and have included this discussion to **Section 3.2 of the updated manuscript**.

---

> ### Author Response · Authors · 2024-11-17
> **Response to Reviewer KxQB (2/2)**
>
> ## 3. Response to Weakness 3: adapting centralized calibration methods to the FL setting
>
> Thank you for this valuable suggestion. We would like to highlight that we have already implemented such baselines (adaptations of centralized calibration methods to FL) and provided comprehensive empirical results in our manuscript. Specifically, as shown in Tables 1, 2, and 6–22, the baselines termed Focal, LS, BS, MMCE, FLSD, MbLS, DCA, and MDCA are all centralized calibration schemes that have been adapted to various FL algorithms, including FedAvg, FedProx, Scaffold, FedDyn, and FedNova. These results demonstrate the performance of centralized calibration methods when applied to the FL context.
>
> In our manuscript, we describe how these centralized train-time calibration methods are integrated into the FL framework (Section 4.1). For instance, auxiliary-based train-time calibration methods (Eq. 5) incorporate their calibration loss, $\ell_{cal}$, into the original FL optimization process. For other loss-based train-time calibration methods, we modify each client’s loss function accordingly to the specific loss functions. For example, we replace the standard cross-entropy loss $\ell$ in Eq. 4 with a specific calibration loss, such as focal loss or label smoothing loss.  The integration of centralized calibration methods into the FL setting has certain limitations, such as applying calibration penalties to each client solely based on their respective datasets, which can lead to biased calibration toward local heterogeneity. Our method addresses this limitation by incorporating the relationship between local and global models, ensuring calibration reflects the broader global distribution and meets the global calibration needs necessary for optimal performance.
>
> We thank the reviewer for this thoughtful comment and have clarified in **Section 5.1 of the updated manuscript** that the centralized calibration methods have been adapted to the FL setting.
>
>
> Again, we appreciate the reviewer for the helpful comments. In case there are remaining questions/concerns, we hope to be able to have an opportunity to further answer them.

---

> > ### Author Response · Authors · 2024-11-24
> >
> > Dear Reviewer KxQB,
> >
> > Thank you again for your time spent reviewing our paper. As the discussion phase winds down, we just wanted to check and see whether you have any further comments based on our response? If so, we would be happy to address them in them in the remaining time.

---

### Official Review · Reviewer_kU7Z · 2024-11-03

**Soundness:** 2
**Presentation:** 2
**Contribution:** 2
**Rating:** 5
**Confidence:** 5

**Summary:**

This paper presents a model calibration method for federated learning, which adaptively adjusts penalties based on the relationship between local and global models.

**Strengths:**

The motivation makes sense.

The paper is well-written and easy to follow.

**Weaknesses:**

The contribution of this paper is limited. The proposed method seems a simple incremental work of the existing FL model calibration method.

Although I understand the author's motivation, I need the author to explain why the proposed metric is more important than accuracy. The proposed metrics are not intuitive enough. Typically, accuracy and the proposed metric are positively correlated. The author should add some detailed experiments to validate its motivation.

The experimental results show that The proposed method is only slightly optimized for federated learning. In addition, the proposed method may cause a decrease in accuracy in some scenarios.

Lake of comparison with SOTA FL optimization methods [1-5]. These methods achieve better model performance by guiding the model to optimize towards the flat region. I think these methods can still reduce the measures proposed by the author. I wonder if the author can compare these methods or integrate the proposed method into them.

[1] FedSpeed: Larger local interval, less communication round, and higher generalization accuracy ICLR 2023

[2] Is Aggregation the Only Choice? Federated Learning via Layer-wise Model Recombination, KDD 2024

[3] Generalized Federated Learning via Sharpness Aware Minimization, ICML 2022

[4] Dynamic regularized sharpness aware minimization in federated learning: Approaching global consistency and smooth landscape. ICML 2023

[5] FedCross: Towards Accurate Federated Learning via Multi-Model Cross-Aggregation, ICDE 2024

**Questions:**

Please see the weakness.

---

> ### Author Response · Authors · 2024-11-17
> **Response to Reviewer kU7Z (1/2)**
>
> We thank the reviewer for acknowledging the motivation and clarity of our paper. We would like to address any misunderstanding by first providing further background of this work.
>
> - In the centralized setting, model calibration [1] aims to align predicted probabilities with the actual likelihood of correctness, ensuring that the model's confidence reflects true accuracy. The commonly used measurements for calibration are Expected Calibration Error (ECE) [2] and Static Calibration Error (SCE) [3], which quantify the difference between predicted confidence and observed accuracy. The goal of model calibration works [1, 4, 5, 6] in a centralized setting is always to reduce the above-mentioned errors with minimal reduction in accuracy.
>
> - In this work, we pioneer the empirical study of model calibration issues in federated learning. To the best of our knowledge, only one previous work [7] addresses a similar scenario. We propose a new adaptable calibration method that can be applied to various FL algorithms, effectively mitigating calibration error and, in most cases, enhancing the performance of these FL algorithms—a result not achieved by [7]. Our comprehensive comparisons demonstrate that our method outperforms theirs.
>
>
> >[1] On Calibration of Modern Neural Networks, ICML ‘17
> >
> >[2] Obtaining well calibrated probabilities using bayesian binning, AAAI ‘15
> >
> >[3] Measuring Calibration in Deep Learning, CVPR ‘20
> >
> >[4] The Devil is in the Margin: Margin-based Label Smoothing for Network Calibration, CVPR '22
> >
> >[5] A Stitch in Time Saves Nine: A Train-Time Regularizing Loss for Improved Neural Network Calibration, CVPR '22
> >
> >[6] Calibrating Deep Neural Networks using Focal Loss, NeurIPS '20
> >
> >[7] Fedcal: Achieving local and global calibration in federated learning via aggregated parameterized scaler, ICML ‘24
>
>
> ## 1. Response to Weakness 1
> We would like to clarify that our proposed solution is a versatile module designed to adapt seamlessly to any "FL algorithm"—not to "FL model calibration methods." This flexibility allows our method to enhance calibration across diverse FL setups, making it broadly applicable.
>
> Given that only one prior work [7] addresses model calibration in FL, our approach stands out with a fundamentally different methodology. Unlike [7], which involves training post-hoc scalers on each local client and subsequently aggregating those scalers on the server for global calibration, our method explicitly considers the interactions between the global server and local clients during calibration. The scaler used for calibrating the global model in [7] can only reflect the clients' local distributions without considering the dynamic interplay between server and clients, which may lead to bias toward individual clients and fail to capture the global context. In Tables 4, 23, and 24 of our manuscript, our method consistently outperforms [7] in terms of model calibration across various scenarios. We have highlighted this discussion in the blue text in **Section 2 of our manuscript**.
>
> ## 2. Response to Weakness 2
>
> We would like to clarify that the metrics we use are widely recognized and commonly used in the  model calibration research community [4, 5, 6]. In our work, we do not intend to suggest that accuracy is unimportant, nor do we claim that calibration is more significant than accuracy. However, calibration is crucial in federated learning applications where model reliability is paramount, such as in the medical domain. Without proper calibration, models risk producing misleading confidence levels, which can have serious implications for real-world decision-making in sensitive applications.
>
> The focus of this paper is indeed to improve calibration, and therefore we emphasize calibration error in our evaluation. As stated in our research goal (line 70-88), our primary aim is to reduce calibration error without harming accuracy. We recognize accuracy as essential, and our methodology is built on the premise that calibration improvements should not come at the expense of accuracy. We have emphasized this point in **lines 86–88 of the updated manuscript**.
>
> Additionally, we have observed that accuracy and calibration error do not always correlate positively, which is consistent with existing works in centralized model calibration works [4, 5]. For example, when comparing uncalibrated FedAvg with FedNova or FedDyn (FL algorithms known for strong performance in non-IID scenarios, as seen in the first row of Tables 1 and 2), we frequently find that as accuracy improves, calibration error increases—showing a negative correlation where gains in one metric coincide with declines in the other. This highlights the importance of model calibration in federated learning settings, as high accuracy alone may not guarantee reliability in real-world applications.

---

> > ### Comment · Reviewer_kU7Z · 2024-11-22
> >
> > Thanks for the replay. I am still concerned about the effectiveness of calibration errors. The author claims that " high accuracy alone may not guarantee reliability in real-world applications", but I want to know how to evaluate the reliability. Is there a more intuitive experiment to validate this? In addition, what are the advantages of the proposed metric compared to directly using loss as a metric?

---

> ### Author Response · Authors · 2024-11-17
> **Response to Reviewer kU7Z (2/2)**
>
> ## 3. Response to Weakness 3
>
> We clarify that, as stated in our background, the goal of model calibration is not to improve accuracy but rather to reduce calibration error without compromising accuracy. Accuracy improvement is therefore not the primary focus of our paper, which is why we did not highlight any gains in accuracy across different FL algorithms. Instead, our objective is to provide superior calibration for the baseline models while ensuring their performance remains stable. Again, this goal is consistent with the model calibration literature in the centralized setting [1, 4, 5, 6].
>
> We found that applying existing calibration algorithms in federated learning settings do not provide enough calibration error improvements, as they do not consider the inherent data heterogeneity in FL environments. The non-IID data across clients makes model calibration particularly difficult, as it must ensure reliability across diverse data distributions and client conditions. Our key contribution is to tackle this problem by proposing a flexible model calibration scheme tailored for federated learning, which can be seamlessly integrated with various federated learning algorithms. In particular, our NUCFL approach assesses the similarity between local and global model relationships, and controls the penalty term for the calibration loss during client-side local training.
>
> We would like to emphasize that our method also demonstrates significant calibration superiority while maintaining baseline performance. We have thoroughly validated NUCFL across diverse conditions—using four datasets, each with four different data distributions, resulting in 540 FL configurations with our NUCFL calibration module (Tables 1, 2, 7-22). Only in 11.8% of cases did our method lead to a slight accuracy reduction in the uncalibrated baseline FL algorithms, with an average decrease of just 0.00213 points. Importantly, in all cases, our method achieved a lower calibration error than the baseline, showcasing its robustness and reliability. We thank the reviewer for this insightful comment, and we have included this discussion in **Section 5.2 of the updated manuscript**.
>
>
> ## 4. Response to Weakness 4
>
> We appreciate the reviewer’s suggestion. Since the suggested papers do not directly tackle model calibration, they result in high expected calibration error (ECE). We implemented two of the baselines suggested by the reviewer, and also integrated with our scheme. The results are provided below.
>
>
> It is also important to emphasize that our calibration method serves as an adaptable module that can be applied to any FL algorithm. In this study, we have applied it to commonly used FL baselines, such as Scaffold, FedProx, FedDyn and FedNova, which are considered advanced FL algorithms optimized for performance in various FL studies. The primary goal of our paper is to demonstrate that, regardless of how FL algorithms are optimized for performance, our method can be seamlessly integrated into any FL framework to enhance model calibration, provided that the FL structure is maintained. For example, our calibration method effectively reduces calibration error for FedDyn and FedNova without harming their accuracy.
>
>
> To further confirm this using the baseline suggested by the reviewer, we conducted additional experiments using FedSpeed (with $\lambda$ = 0.1) and FedSAM as recommended by the reviewer. Following the same dataset, distribution, and FL setup as in Table 1, we compared uncalibrated FedSpeed and FedSAM results with those obtained using FedCal (current SOTA) and our method (NUCFL). The results are presented in the table below.
>
>
>
> |Calibration method | FedSpeed | FedSAM
> |---|---|---|
> | | ACC, ECE | ACC, ECE
> |Uncalibrated | 65.10, 15.33 | 64.23, 13.95
> |FedCal (current SOTA)  | 65.10, 12.95 | 64.23, 12.08
> Our NUCFL (DCA+L-CKA) | **65.17, 11.15** | **64.52, 10.22**
>
> In the absence of any calibration, we find that these advanced FL algorithms do improve accuracy compared to FedAvg (ACC: 61.34; ECE: 10.52). However, they also introduce higher calibration errors. By applying our calibration method to these advanced FL algorithms, we find that our approach effectively reduces calibration error and even outperforms the SOTA calibration method in terms of calibration results. These results also demonstrate the adaptability of our method, showing that it can be applied to any FL algorithm to improve calibration while maintaining accuracy. We have added this comparison and discussion in **Appendix B.2 and Table 25**.
>
>
>
> Again, thank you for your time and efforts for reviewing our paper. We would appreciate further opportunities to answer any remaining concerns you might have.

---

> > ### Comment · Reviewer_pbym · 2024-11-21
> > **Comment of sharpness-aware approaches**
> >
> > As a side comment, I think Reviewer kU7Z03 raised an interesting point of further discussion in future works, i.e., the relationship between calibration error and the loss landscape. In particular, as I also mentioned in my previous comments, the proposed experiments (both original and new ones) highlight that accuracy and calibration error do not appear to be straightforwardly connected. For instance, in the table comparing FedAvg and FedSAM, the former reaches higher accuracy but the  latter shows lower calibration error. Thus, a few questions come to mind:
> > - SAM and FedSAM are known for improving model generalization, so it makes sense for the calibration error to be lower w.r.t. FedAvg. However, since both SAM and calibration error are linked to model generalization, how does calibration error impact the loss landscape flatness? I.e., are models characterized by high calibration error also found in sharp regions of the loss landscape (or viceversa)?
> > - Several works are currently studying SAM more in-depth to understand whether its generalization properties actually derive from the geometry of the loss landscape or the flatness of the solution is just one of the many side benefits due to its implicit regularization properties. It would be interesting to see if a connection can be built between studies focused on model calibration and works from the sharpness literature, and how that eventually reflects on FL.
> >
> >  I am not looking for empirical answers to these questions, but I would be interested in knowing what the authors think of these points.

---

> > ### Comment · Reviewer_kU7Z · 2024-11-22
> >
> > The author only selected two methods from the five SOTA works I proposed for comparison.
> > I do not know why the author chose them. The author needs to give some explanation.
> > Note that, I give the SOTA methods based on two technical routes, i.e., sharpness awareness optimization (SAM) and multi-model collaborative optimization.
> > However, both of the selected baselines are based on sharpness awareness optimization (SAM).
> > The author ignored the method based on multi-model collaborative optimization [2][5].
> > I hope the author can add relevant comparisons or give a reasonable explanation.

---

> ### Author Response · Authors · 2024-11-24
> **Response to Reviewer kU7Z (1/2)**
>
> ### **Effectiveness of the expected calibration error (ECE) metric**
>
> We appreciate **Reviewer kU7Z** for the follow-up. The statement that "high accuracy alone may not guarantee reliability in real-world applications" arises because accuracy focuses solely on correctness, ignoring the confidence of predictions. Reliability, on the other hand, reflects the alignment between the predicted probabilities and actual outcomes. The reliability metric is a useful supplement as it is unrealistic for models to achieve 100% accuracy.
>
>
> There are two commonly used ways to evaluate reliability, as introduced by Guo et al. [1]:
>
> 1. **Numerical Metrics:** Reliability metrics such as Expected Calibration Error (ECE) [2], Maximum Calibration Error (MCE) [2], and Static Calibration Error (SCE) [3] have been introduced in the model calibration community [4,5,6,7] to tackle this issue and capture the reliability of the model. They directly measure the error between the confidence of the prediction (i.e., prediction probabilities) and the real accuracy (i.e., probability of being correct). Our evaluations report both the ECE and SCE metrics.
>
>
> 2. **Visualizations:** Reliability Diagrams (as shown in Figure 1) offer an intuitive representation of calibration by plotting predicted confidence against observed accuracy. A well-calibrated model exhibits points close to the diagonal, giving a convenient way of visually assessing reliability.
>
>
> These existing calibration metrics have distinct advantages over directly using loss as a reliability metric. While the conventional "loss" measures prediction quality by penalizing incorrect predictions, it does not account for how well confidence aligns with accuracy. Calibration metrics, in contrast, explicitly evaluate this alignment. In scenarios where incorrect predictions can lead to high-risk consequences (e.g., disease diagnosis, stock trading), having a high-accuracy model is necessary. However, unless the model achieves 100% accuracy (which is rarely feasible), it is also critical to determine whether to rely on the ML model’s predictions or to rely on other factors, such as human judgment (e.g., doctor’s  opinion, financial advice).
>
> Consider an example where a model's predicted probability for the true label is 0.8 (a value that can be computed for any test sample based on the softmax output), while the real accuracy of the model probability is 0.5 (a value unavailable during inference). If this mismatch exists, it becomes difficult to decide whether the prediction is trustworthy by only looking at the prediction probability value. Once the model is trained such that its confidence (or predicted probability) aligns closely with its accuracy, users can interpret the “prediction probability” for each test sample as the “probability of being correct.” Hence, based on a specific rule, a user can decide whether to rely on the model or not: If the prediction probability is 0.95, the user may be inclined to rely on the model since it indicates that the prediction is correct with probability of 0.95. If the prediction probability is 0.3, the user may not want to use the model’s prediction because the prediction is correct with only 0.3 probability.
>
> Importantly, it is well known that neural networks produce overconfident predictions, where the predicted probabilities tend to be much higher than the actual accuracy (i.e., resulting in a high ECE or SCE). In such cases, it is difficult for users to decide whether to trust the model’s prediction by only looking at the confidence, potentially leading to significant risks. Hence, a model with a “high ECE” is referred to as an “unreliable” model, whereas a model with a “low ECE” is often considered a “reliable” model. This is why we report the ECE metric as a key supplement to the accuracy metric, as in existing model calibration literature [4,5,6,7].
>
>
> > [1] On Calibration of Modern Neural Networks, Guo et al., ICML '17
> >
> > [2] Obtaining well calibrated probabilities using bayesian binning, Naeini et al., AAAI '15
> >
> > [3] Measuring Calibration in Deep Learning, Nixon et al., 2019
> >
> > [4] Calibrating Deep Neural Networks using Focal Loss, Mukhoti et al., NeurIPS '20
> >
> > [5] The devil is in the margin: Margin-based label smoothing for network calibration, Liu et al., CVPR '21
> >
> > [6] Improved Trainable Calibration Method for Neural Networks on Medical Imaging Classification, Liang et al., BMVC '20
> >
> > [7] A stitch in time saves nine: A train-time regularizing loss for improved neural network calibration, Hebbalaguppe et al., CVPR '22

---

> ### Author Response · Authors · 2024-11-24
> **Response to Reviewer kU7Z (2/2)**
>
> ### **Additional baselines**
>
> We also thank **Reviewer kU7Z** for this additional comment. The reason we initially chose two baselines was simply to demonstrate our efforts as quickly as possible during the rebuttal period. Now we further report the results using FedMR [2] and FedCross ($\alpha = 0.99$) [5]. We follow the same dataset, distribution, and FL setup as in Table 1. We compared uncalibrated FedMR and FedCross with those calibrated using FedCal and our proposed NUCFL by incorporating the calibration loss during local training, prior to collaborative model selection or model recombination. The results are presented in the table below.
>
> |Calibration method | FedMR | FedCross
> |---|---|---|
> | | ACC; ECE | ACC; ECE
> |Uncalibrated |  64.95; 13.18 | 65.30; 15.81
> |FedCal (current SOTA)  |  64.95; 12.85 | 65.30; 14.39
> Our NUCFL (DCA+L-CKA) |  64.95; 12.11   | 65.33; 13.01
>
> Aligning with the results we introduced for previous response, we observe that these methods can also be interpreted as improved optimization techniques for enhancing the accuracy of FL. With the integration of our calibration method, we see that these algorithms can achieve even lower calibration error. We thank the reviewer for this comment and have included this comparison in the revised manuscript, making edits to **Appendix B.2** and **Table 25** accordingly.
>
> Please let us know if you have any further concerns. We would be happy to address them.

---

> > ### Comment · Reviewer_kU7Z · 2024-11-25
> >
> > Thanks for your response. The author addressed some of my concerns. However, I still believe that the contribution of this paper is incremental. Since ICLR is a top conference, I cannot give a positive score.
> > However, I really appreciate the author's attitude and efforts in the rebuttal phase, so I improved my score to 5 (marginally below the acceptance threshold).
> > Different reviewers have different evaluation criteria. I hope the author can understand.
> > Good luck!

---

> > > ### Author Response · Authors · 2024-11-25
> > >
> > > We are glad that we were able to address reviewer’s questions and sincerely thank you for raising the score. We greatly appreciate your thoughtful feedback. Thank you again for your time and consideration.

---

> ### Author Response · Authors · 2024-11-24
> **Response to Reviewer pbym**
>
> ### **Comments on sharpness-aware approaches**
>
> We sincerely thank **Reviewer pbym** for this insightful side comment.
>
> To clarify, the table we provided includes additional results comparing FedSpeed and FedSAM. When comparing FedAvg (in Table 1, ACC: 61.34; ECE: 10.52) with both of these baselines, we observed that they introduce higher accuracy, accompanied by higher calibration error.
>
> We believe that the generalization capability of FedSAM, as highlighted in its original paper, directly contributes to its improved accuracy, as also shown in our additional results: FedSAM (ACC: 64.23) vs. FedAvg (ACC: 61.34).  However, it seems that the changes in the loss landscape brought about by SAM do not necessarily ensure alignment between model confidence and actual accuracy, which could result in a higher ECE, as observed in our results: FedSAM (ECE: 13.95) vs. FedAvg (ECE: 10.52). This suggests that while SAM's perturbation improves accuracy through changing loss landscape sharpness, it does not inherently guarantee an alignment between accuracy and confidence.
>
> Although this is currently an unexplored area, we agree with the reviewer that investigating the relationship between calibration error and loss landscape sharpness is an interesting direction for future work. Understanding how the loss landscape correlate with calibration error could open new avenues for bridging calibration studies with sharpness literature in FL.
>
> Once again, we appreciate the reviewer for raising these thought-provoking points.

---

### Official Review · Reviewer_pbym · 2024-11-04

**Soundness:** 3
**Presentation:** 3
**Contribution:** 3
**Rating:** 8
**Confidence:** 4

**Summary:**

The paper addresses the issue of model calibration in federated learning (FL), with a specific focus on heterogeneous scenarios, characterized by distribution shifts across clients. The paper argues that most works have focused on improving the accuracy of the proposed methods so far, but almost no attention has been given to the model's confidence in its predictions. This highlights a gap in the current literature: FL models should hava reliable confidence in their predictions to be deployed in real-world use cases. To fill this gap, this work introduces Non-Uniform Calibration for Federated Learning (NUCFL), a novel method to integrate FL with model calibration, that can be easily combined with existing approaches.

The work discusses the various benefits of NUCFL, studying its application with several calibration approaches and FL methods, showing its effectiveness in improving the model's confidence in its predictions, according to multiple metrics.

**Strengths:**

- The paper underlines a gap in the current FL research and proposes an effective method to address it
-  NUCFL can be easily added on top of existing methods
- The paper extensively shows the efficacy of NUCFL across various settings, model architectures and methods
- NUCFL is extensively studied through in-depth and well-carried analyses
- NUCFL does not increase communication costs

**Weaknesses:**

- Difficulties in proving NUCFL's convergence and theoretical properties, as also pointed out by the authors
- The paper argues that works like Zhang et al. [2022] and Luo et al. [2021] misuse the term "calibration" and do not further analyze the comparison with those methods. However, I believe it would still be relevant to the FL community to understand how NUCFL compares to them.
- The application of NUCFL seems limited to supervised scenarios

Comments (did not affect my rating):
- I believe the paper would benefit from the outlined Algorithm of NUCFL for better understanding
- Please modify the citation style to improve the paper's readibility (the citation are not enclosed in brackets)
- Line 136: typo ".."

**Questions:**

- Could the authors provide a comparison between NUCFL and Zhang et al. [2022] and Luo et al. [2021], according to the introduced metrics?
- Does NUCFL increase computational costs on the client-side?
- How could NUCFL be adapted to unsupervised/semi-supervised settings?
- How sensible if NUCFL to the hyperparam $\beta$?

---

> ### Author Response · Authors · 2024-11-20
> **Response to Reviewer pbym (1/2)**
>
> We sincerely thank the reviewer for the thoughtful and positive feedback on our work. We are delighted that the reviewer found value in our efforts to address a critical gap in FL.
>
> ## 1. Response to Weakness 1
>
> We thank the reviewer for this comment. Weacknowledge that our paper is primarily an empirical study exploring the practical impacts of model calibration for federated learning. As highlighted in Remark 1, similar to existing work on model calibration in centralized settings (e.g., [1]-[5]), this area remains challenging in providing theoretical guarantees or convergence proofs for calibration methods, even in the centralized case. Thus, it was also natural for these centralized model calibration methods to focus primarily on algorithmic approaches.
>
> We tried to substantiate the effectiveness of our approach through extensive experiments. Specifically, we demonstrate that our approach:
>
> (a) significantly reduces calibration error across various datasets and FL settings, and
>
> (b) adapts effectively to multiple FL algorithms, further validating its robustness.
>
> We hope these contributions highlight the practical value of our work despite the theoretical challenges associated with this area.
>
> > [1] Focal Loss for Dense Object Detection, Lin et al.,  ICCV ‘17
> >
> > [2] When Does Label Smoothing Help?, Muller et al., 2019
> >
> > [3] Trainable Calibration Measures For Neural Networks From Kernel Mean Embeddings, Kumar et al., ICML ‘18
> >
> > [4] Calibrating Deep Neural Networks using Focal Loss, Mukhoti et al., 2020
> >
> > [5] The Devil is in the Margin: Margin-based Label Smoothing for Network Calibration, Liu et al., CVPR ‘22
>
>
> ## 2. Response to Weakness 2 and Question 1
>
> We thank the reviewer for this insightful comment. To address this feedback, we have conducted comparisons with the mentioned works, FedLC [Zhang et al., 2022] and CCVR [Luo et al., 2021]. While these methods do not explicitly consider "model calibration" for FL, we recognize them as advanced FL optimization approaches. Specifically, we evaluate both the uncalibrated versions of FedLC (with $\tau = 1$) and CCVR (on FedAvg) as well as their calibrated counterparts by integrating our proposed NUCFL. We follow the same experimental setup as described in Table 1, utilizing the CIFAR-100 dataset under a non-IID data distribution scenario. The results are provided below.
>
>
>
> | |FedAvg  | CCVR | FedLC
> |--|--|--|--|
> |  | ACC, ECE | ACC, ECE | ACC, ECE
> | Uncal. | 61.34 10.52 | 63.01, 13.69| 62.73, 14.58
> | + NUCFL (DCA + L-CKA)| **62.05 6.14** | **63.05, 9.28** | **62.77, 10.98** |
>
>
>
> Comparing uncalibrated CCVR and FedLC with FedAvg, we observe an increase in accuracy alongside a rise in calibration error. This indicates that these algorithms are advanced FL optimization methods designed primarily to improve accuracy rather than address model calibration.  Comparing uncalibrated FedAvg with NUCFL-calibrated FedAvg, uncalibrated CCVR, and uncalibrated FedLC, we observe that the latter two methods improve accuracy while significantly increasing the calibration error compared to the uncalibrated FedAvg. However, NUCFL enables FedAvg to achieve both an improvement in accuracy and a reduction in calibration error.
>
> Since we view CCVR and FedLC as advanced FL optimization methods, we can apply NUCFL to them. Comparing the original versions with their NUCFL-calibrated counterparts, we observe improvements in both accuracy and calibration error. This demonstrates the adaptability of our method and its compatibility with any FL algorithm.
>
> We thank the reviewer for this comment and have included the comparison and discussion in **Appendix B.7 and Table 27** of the updated manuscript.
>
>
>
> > [Zhang et al.] Federated Learning with Label Distribution Skew via Logits Calibration, 2022
> >
> > [Luo et al.] No Fear of Heterogeneity: Classifier Calibration for Federated Learning with Non-IID Data, 2021

---

> ### Author Response · Authors · 2024-11-20
> **Response to Reviewer pbym (2/2)**
>
> ## 3. Response to Weakness 3 and Question 3:
>
> We thank the reviewer for pointing out this important aspect. We acknowledge that model calibration for unsupervised learning is an interesting and relatively new area, even in centralized settings. Most existing works [1,2] in this domain focus on calibration under scenarios such as domain adaptation or domain shifts, or on estimating model accuracy in the absence of labels. Extending these approaches to unsupervised FL is indeed an intriguing avenue for future research.
>
> Some possible directions include integrating calibration methods [1,2] designed for centralized and unsupervised learning into the local update processes of FL. Additionally, NUCFL’s auxiliary penalties could be adjusted to reflect domain distributions, such as enhancing inter-domain style consistency in out-of-domain scenarios. We believe that these scenarios are inherently more complex than supervised settings due to the added challenges of the lack of labeled data, necessitating more careful and domain-specific design choices.
>
> We appreciate the reviewer bringing up this point and agree that exploring calibration methods in unsupervised FL is an exciting direction for future work. We have included this discussion in **Section 6** of our updated manuscript. Thank you again for your thoughtful feedback.
>
> > [1] Unsupervised Calibration under Covariate Shift
> >
> > [2] Calibration of Network Confidence for Unsupervised Domain Adaptation Using Estimated Accuracy
>
>
>
> ## 4. Response to Question 2 :
>
> We thank the reviewer for this query. NUCFL indeed introduces additional computational costs on the client-side for calculating the calibration loss and measuring the similarity between the local model and the received global model. However, these costs are negligible compared to the computational demands of the client’s local training process. We have included the actual time spent (in seconds) for each component in below table, which demonstrates that the additional overhead is minimal and does not significantly impact the overall efficiency.
>
>
> | |Local Training | Calibration Loss | Similarity Calculation
> |--|--|--|--|
> |time (s)|174 | 31 | 25 |
>
>
>
> ## 4. Response to Question 4 :
>
> In Section 4.2, we discuss the challenges associated with using a fixed value for $\beta$. A fixed $\beta$ may result in local models being calibrated solely based on their respective datasets. Uniformly small $\beta$ bias the calibration toward local heterogeneity, while uniformly large $\beta$ may overlook accuracy improvements derived from classification.
>
> However, tuning optimal $\beta$ values for each local client is also challenging. This motivated the design of our method, which employs similarity-based functions to dynamically determine $\beta$ for each client without requiring manual parameter tuning (as discussed in lines 310–311). The $\beta$ values are automatically determined by the nature of the distribution between the global model and the local model.
>
> In our manuscript, we also examine the effects of different beta values. The results, extracted from Table 1 and Table 3, are summarized below. For DCA, a fixed $\beta = 1$ is used for all clients. NUCFL assigns dynamic penalties to each client based on similarity scores, with higher similarity between the global and local models corresponding to higher beta values. Additionally, "Reversed NUCFL" serves as an ablation study, where the calibration logic is reversed, assigning lower beta values to higher similarity scores.
>
>
>
> | |ACC| ECE|
> |--|--|--|
> |DCA (fixed $\beta$)| 61.24 |7.69
> |NUCFL (DCA + L-CKA)| 62.05 | 6.14
> |Reversed NUCFL (DCA+L-CKA) | 60.36 | 13.89
>
>
>
> The improvement of calibration error from fixed beta (DCA) to dynamic beta (NUCFL) shows the effectiveness and robustness of our method. By dynamically adjusting beta based on the similarity between local and global models, NUCFL effectively reduces calibration error without the need for extensive hyperparameter tuning, highlighting its practicality and adaptability.
>
>
>
> We also observe from the ablation study, comparing NUCFL with Reversed NUCFL, that this comparison highlights the soundness of our design logic in using similarity to control beta. The reversed logic, which assigns lower beta values to higher similarity scores, negatively impacts the performance, further validating our approach and its effectiveness in improving calibration error.
>
>
>
>
>
> ## 5. Response to other comments
>
> We thank the reviewer for other comments. In response, we have included an algorithm table **(Algorithm 2) in Appendix A.2** to provide a clearer understanding of our NUCFL. Additionally, we have modified the citation style to improve the paper's readability and corrected the typo.
>
>
> Again, we thank the reviewer for the positive feedback on our work and for recognizing the value of our efforts in addressing model calibration for FL. We are happy to address any further questions if needed.

---

> ### Comment · Reviewer_pbym · 2024-11-21
> **Official reply**
>
> I thank the authors for addressing my questions.
>
> **W1**: First, I would like to underline that I found this work valuable even if not supported by strong theoretical claims. I think it opens a new interesting direction for studying the behavior of models in FL with other metrics other than their reached accuracy. This claim is supported by several empirical evidences, providing a significant contribution. To foster future research, which are the main challenges the authors found in providing theoretical analyses? And are any of those challenges proper of the FL scenario?
>
> **W2**: I thank the authors for providing this additional comparison. It is interesting to see how the use of model calibration achieves roughly the same accuracy of the baselines, but with significant lower calibration errors. This suggests calibration error and model accuracy may not be straightforwardly connected. Do the authors have any intuitions on this aspect? If accuracy is not a strong indicator of high/low calibration error, how can the presence of high calibration errors be spotted in practice (e.g., noise in the learning trends, etc)? To clarify my question, I am wondering how one could distinguish a badly vs a better calibrated model by simply looking at their behavior during training or at test time,

---

> ### Author Response · Authors · 2024-11-23
> **Response to Reviewer pbym**
>
> ## 1. Response to W1:
>
> We thank the reviewer for their kind words and for recognizing the value of our work.
>
> When considering only accuracy (i.e., conventional loss), the client’s local objective function is clearly defined, and the global objective function can be derived accordingly. However, in our work, we incorporate calibration by introducing auxiliary losses and dynamically adjusting the weight for each client’s auxiliary loss. This makes it challenging to define a well-structured “local objective function” for clients and a corresponding “global objective function” for the system to demonstrate convergence. This challenge is not unique to our work; it is common in empirical studies that modify the objective function with dynamic weighting mechanisms, particularly in scenarios like FL where client heterogeneity further complicates theoretical analysis. Future studies can consider how to mathematically characterize the local and global objectives for these scenarios to enable convergence analysis.
>
> ## 2. Response to W2:
> Thank you again for your time. We agree that these metrics are not straightforwardly connected. The reason is that calibration error and accuracy measure different aspects of a model's performance. While accuracy reflects the proportion of correct predictions, calibration error measures the alignment between predicted probabilities and actual correctness. These metrics are not necessarily aligned because:
>
> -  A model can be highly accurate but poorly calibrated if it systematically over- or underestimates its confidence.
>
> - Conversely, a well-calibrated model may not achieve high accuracy, especially in scenarios where the underlying model struggles to learn discriminative features for classification.
>
>  In fact, it is well known that neural networks often produce overconfident predictions [1,2,3], meaning that their confidence is higher than the actual accuracy. Hence, even when the confidence (or the prediction probability) is 0.99 for a specific test sample, the probability of being correct could be much lower, e.g., 0.5. This indicates that the models are unreliable (unless the accuracy is 100%), which motivated researchers to study metrics for reliability such as expected calibration error (ECE).
>
> To assess calibration during the training procedure, we can compute the ECE and/or SCE metrics as training progresses, applying the formulas in lines 367 and/or 375 over time. These calculations can be conducted on the client side both using their local models as well as intermediate global models received, to assess the progress towards improving calibration. They are computed by using the model predictions and the available ground truth labels from the training dataset. At test time, it becomes more challenging to assess calibration (similar to accuracy) since we typically lack true labels in practical scenarios. Users will need access to an additional validation set with true labels to assess model reliability in this case.
>
>
> > [1] On Calibration of Modern Neural Networks, ICML '17
> >
> > [2] Why ReLU networks yield high-confidence predictions far away from the training data and how to mitigate the problem, CVPR '19
> >
> > [3] Calibrating Deep Neural Networks using Focal Loss, NeurIPS '20

---

> > ### Comment · Reviewer_pbym · 2024-11-24
> > **Official reply**
> >
> > I thank the authors for their further discussion and for clarifying my points. **I am confident on the quality of this work and my assigned score.**
> >
> > Assuming there is a dataset available for computing ECE and/or SCE metrics, do the authors think these metrics could be used to weight the server-side aggregation (e.g., instead of the standard weighing applied by FedAvg) to prefer better calibrated solutions?

---

> > > ### Author Response · Authors · 2024-11-25
> > > **Response to Reviewer pbym**
> > >
> > > We thank the reviewer for your thoughtful feedback and continuous support!
> > >
> > > Regarding your question, prior works [1,2] have designed server-side aggregation strategies for gradients or models of local clients. Specifically, [1] proposed a principal gradient-based server aggregation method to reduce the aggregation loss, while [2] proposed a geometric median based aggregation approach for robust aggregation. Now by integrating calibration losses such as the DCA or MDCA losses described in Equations (6) and (7) of the manuscript, the gradients produced at each client would differ from those without calibration losses. This adjustment ensures better alignment between confidence and accuracy. The new gradient will directly impact (i) the aggregation weight used in the principal gradient-based server aggregation described in [1] and (ii) the outcome of the geometric median-based aggregation outlined in [2]. This approach has the potential to produce better-calibrated global solutions by prioritizing contributions from locally better-calibrated models.
> > >
> > > On the other hand, [3] controls the aggregation weights using a server-side proxy dataset. If a proxy dataset for computing ECE and/or SCE metrics is available on the server-side, it could similarly guide aggregation by evaluating the calibration performance of each client model against this proxy dataset. Specifically, instead of  only considering the cross-entropy loss at the server using the proxy dataset, one can consider the weighted sum of the cross-entropy loss and the calibration losses (in Equations (6) and (7)) to design the aggregation weights. This would allow the server to adjust the aggregation weights dynamically to favor models that perform better in terms of calibration, potentially resulting in a more reliable and well-calibrated global model.
> > >
> > > Again, we thank the reviewer for this insightful comment. We believe that exploring these ideas as part of future work could open up exciting new directions for designing aggregation strategies that effectively prioritize both accuracy and model calibration in federated learning systems.
> > >
> > > > [1] Zeng et al., “Tackling Data Heterogeneity in Federated Learning via Loss Decomposition”, MICCAI 2024.
> > > >
> > > > [2] Pillutla et al., “Robust Aggregation for Federated Learning,” IEEE Transactions on Signal Processing, 2022
> > > >
> > > > [3] Li et al., “Revisiting Weighted Aggregation in Federated Learning with Neural Networks,” ICML 2023

---

### Meta-Review · Area_Chair_S1bF · 2024-12-22

**Metareview:**

The paper introduces Non-Uniform Calibration for Federated Learning (NUCFL), a novel framework addressing model calibration in FL environments. It highlights a critical gap in FL research—ensuring reliable model predictions (calibration) across heterogeneous data distributions. The key innovation lies in dynamically adjusting calibration penalties based on the similarity between local and global models, thus aligning local training with global calibration needs. NUCFL is versatile and compatible with various FL algorithms. The authors provided extensive emprical validation in the manuscripts and during rebuttal, demonstrating the effectiveness of NUCFL on wide range of FL algorithms and datasets. Despite some theoretical and exploratory gaps, I lean to recommend acceptance.

**Additional Comments On Reviewer Discussion:**

All the reviewers agreed the problem studied in this paper was well motivated. The proposed method is versatile and compatible with various FL algorithms, indicating good practical values. The concerns are around theoritical foundations, the possibility to extend beyond supervised setting, experiment setting (e.g., hyperparameter sensitivity, comparison with SOTA FL methods)  The authors provided additional experiments and revisions during the rebuttal phase, demonstrating their understanding and engagement with reviewer concerns. In the reviewer discussion statge, one reviewer championed this paper. Despite some theoretical and exploratory gaps, the contribution is sufficiently novel and impactful to merit acceptance at ICLR.

---

### Decision · Program_Chairs · 2025-01-22

Accept (Poster)